# A $CO_2$ - $\Delta^{14}CO_2$ inversion setup for estimating European fossil $CO_2$ emissions

Carlos Gómez-Ortiz[1], Guillaume Monteil[1,2], Sourish Basu[3,4], and Marko Scholze[1]

[1]Department of Physical Geography and Ecosystem Science, Lund University, Lund, Sweden
[2]Barcelona Supercomputing Center, Barcelona, Spain
[3]Global Modeling and Assimilation Office, NASA Goddard Space Flight Center, Greenbelt, MD, USA
[4]Earth System Science Interdisciplinary Center, University of Maryland, College Park, MD, USA

**Correspondence:** Carlos Gómez-Ortiz (carlos.gomez@nateko.lu.se)

**Abstract.** Independent estimation and verification of fossil $CO_2$ emissions on a regional and national scale are crucial for evaluating the fossil $CO_2$ emissions and reductions reported by countries as part of their Nationally Determined Contributions (NDCs). Top-down methods, such as the assimilation of *in situ* and satellite observations of different tracers (e.g., $CO_2$, CO, $\Delta^{14}CO_2$, $XCO_2$), have been increasingly used for this purpose. In this paper, we use the Lund University Modular Inversion Algorithm (LUMIA) to estimate fossil $CO_2$ emissions and natural fluxes by simultaneously inverting *in situ* synthetic observations of $CO_2$ and $\Delta^{14}CO_2$ over Europe. We evaluate the inversion system by conducting a series of Observing System Simulation Experiments (OSSEs). We find that in regions with a dense sampling network, such as Western/Central Europe, adding $\Delta^{14}CO_2$ observations in an experiment where the prior fossil $CO_2$ and biosphere fluxes are set to zero allows LUMIA to recover the time series of both categories. This reduces the prior-to-truth root mean square error (RMSE) from 1.26 TgC day$^{-1}$ to 0.12 TgC day$^{-1}$ in fossil $CO_2$ and from 0.97 TgC day$^{-1}$ to 0.17 TgC day$^{-1}$ in biosphere fluxes, reflecting the true total $CO_2$ budget by 91%. In a second set of experiments using realistic prior fluxes, we find that in addition to retrieving the time series of the optimized fluxes, we are able to recover the true regional fossil $CO_2$ budget in Western/Central Europe by 95% and in Germany by 97%. In all experiments, regions with low sampling coverage, such as Southern Europe and the British Isles, show poorly resolved posterior fossil $CO_2$ emissions. Although the posterior biosphere fluxes in these regions follow the seasonal patterns of the true fluxes, a significant bias remains, making it impossible to close the total $CO_2$ budget. We find that the prior uncertainty of fossil $CO_2$ emissions does not significantly impact the posterior estimates, showing similar results in regions with good sampling coverage like Western/Central and Northern Europe. Finally, having a good prior estimate of the terrestrial isotopic disequilibrium is important to avoid introducing additional noise into the posterior fossil $CO_2$ fluxes.

## 1 Introduction

Carbon dioxide ($CO_2$) from fossil fuels and cement production has become the dominant source of anthropogenic emissions to the atmosphere from around 1950, leading to a mixing ratio of $CO_2$ in the atmosphere of 419.70 ppm on September 16$^{th}$, 2023, 49% above pre-industrial levels (https://gml.noaa.gov/ccgg/trends/gl_trend.html, accessed September 18$^{th}$, 2023). Although land and ocean sinks of $CO_2$ have increased over the past six decades, the fraction of emissions removed from the atmosphere

is expected to decline as the $CO_2$ mixing ratio increases; therefore, a higher proportion of emitted $CO_2$ will remain in the atmosphere (Eyring et al., 2021). Monitoring $CO_2$ emissions and removals is important to follow compliance with international treaties such as the Paris Agreement (UNFCCC, 2016). In the Agreement, the Parties have committed to report their emissions and removals of $CO_2$ and other greenhouse gases (GHGs) to the United Nations Framework Convention on Climate Change (UNFCCC) through the annual GHG inventory. In the case of fossil $CO_2$ emissions, these inventories have been reported to have uncertainties between $5\%$ and $10\%$ in developed countries. These annual inventories and other national-level data are used to spatially and temporally distribute $CO_2$ emissions at sub-national and sub-annual scales. These spatially distributed products help us to better understand the sources of $CO_2$ emissions and to implement more effective policies toward emission reduction (Han et al., 2020). Commonly available emission products, such as the Carbon Dioxide Information and Analysis Center (CDIAC) $FFCO_2$ emission maps (Andres et al., 2011), the Open-source Data Inventory for Anthropogenic $CO_2$ (ODIAC) emission data product (Oda and Maksyutov, 2011; Oda et al., 2018), and the Emissions Database for Global Atmospheric Research (EDGAR) (Janssens-Maenhout et al., 2019), use national energy statistics, power plant emission data, and spatial proxies such as nighttime light observations, population data, and road transport networks to spatially and temporally distribute the emissions. This additional information introduces new uncertainties that, in EDGAR, for instance, can reach a global uncertainty of approximately $11\%$ (Solazzo et al., 2021), or can be as high as $120\%$ in the case of CDIAC (Andres et al., 2016). These uncertainties can be more significant and challenging to characterize at sub-annual and sub-national scales, even in developed countries (Basu et al., 2016; Miller et al., 2012).

These emission products can be used alongside atmospheric observations of $CO_2$ and other tracers in inverse modeling systems to reduce their uncertainty, enhance our understanding of fossil $CO_2$ emissions and natural fluxes, and improve the accuracy of national carbon budgets. So far, atmospheric $CO_2$ inversion frameworks have predominantly been used to constrain terrestrial sources and sinks of $CO_2$ (Basu et al., 2013; Chevallier et al., 2007; Monteil et al., 2020; Monteil and Scholze, 2021). To constrain the terrestrial carbon cycle, inverse modelers typically prescribe fossil $CO_2$ fluxes from emission data products, like those mentioned previously, assuming them to be perfectly well-known (Turnbull et al., 2009). The atmospheric $CO_2$ represent a mixture of all sources, with the natural signal being predominant during most of the year (the growing season covers spring to fall), masking the contribution of fossil $CO_2$ emissions (Shiga et al., 2014). Consequently, additional information is necessary to segregate the fossil contribution from the natural signal in atmospheric $CO_2$ observations to constrain the fossil $CO_2$ fluxes. Some strategies have included sampling approaches where observations are taken close to major fossil $CO_2$ sources (e.g., cities and power plants) (Bréon et al., 2015), or satellite observations of large point sources such as column-integrated atmospheric $CO_2$ mixing ratio ($XCO_2$) (Kaminski et al., 2022; Wang et al., 2020). A more commonly employed method is to combine these $CO_2$-only observations (either $CO_2$ or $XCO_2$) with additional tracers such as $NO_2$ and the $NO_x$:$CO_2$ ratio (Kuhlmann et al., 2021), or ground observations of CO (Newman et al., 2013; Brioude et al., 2013), APO (Atmospheric Potential Oxygen) (Pickers et al., 2022), and, more widely, the radiocarbon ($\Delta^{14}CO_2$) content of carbon dioxide (Turnbull et al., 2009; Basu et al., 2016; Wang et al., 2018), which we use in this study.

Radiocarbon is the radioactive isotope of carbon with a half-life of approximately 5730 years and is produced naturally in the upper atmosphere by cosmic-ray-induced reactions with nitrogen (Turnbull et al., 2009). Fossil $CO_2$ does not contain

radiocarbon (it has already decayed), and adding its $^{14}$C-free emissions to the atmosphere causes a depletion of $\Delta^{14}CO_2$ (Suess, 1955). Meanwhile, radiocarbon is being absorbed and released by the ocean and the biosphere, making it an effective tracer of the natural carbon cycle and, therefore, a tool to distinguish fossil emissions from the natural cycle signal in atmospheric $CO_2$ observations (Turnbull et al., 2009, 2022; Zazzeri et al., 2023). Radiocarbon is also produced as a by-product of nuclear facilities (e.g., nuclear power plants) and atmospheric nuclear weapon tests, the latter occurring mostly between 1945 and 1980, with the highest intensity in 1961-1962 (Naegler and Levin, 2006). These bomb tests caused a significant disturbance in the radiocarbon cycle, resulting in an isotopic disequilibrium in the biosphere and ocean (Hesshaimer et al., 1994). Isotopic disequilibrium is the difference between the isotopic signatures or radiocarbon content of carbon entering and leaving a pool. Despite its similar meaning, this occurs differently in the ocean and the biosphere. In the ocean, the disequilibrium results from $\Delta^{14}$C-depleted $CO_2$ from water that has returned to the surface and was out of contact with the atmosphere, allowing the radiocarbon to decay significantly. In the biosphere, the disequilibrium is a result of the heterotrophic respiration of $\Delta^{14}$C-enriched $CO_2$ assimilated a couple of decades ago when the atmospheric $\Delta^{14}$C was higher due to the bomb spike (Lehman et al., 2013). Therefore, the ocean disequilibrium flux tends to dilute the atmospheric $\Delta^{14}$C content, whereas the biosphere disequilibrium flux tends to enrich it.

The usefulness of atmospheric $\Delta^{14}CO_2$ observations in estimating the fossil $CO_2$ content in the atmosphere as a fraction of the total atmospheric $CO_2$ mixing ratio has been demonstrated in various modeling studies. For instance, Levin and Karstens (2007) present an observational approach to estimate hourly regional fossil fuel $CO_2$ offsets at a continental site (Heidelberg, Germany), using weekly mean $^{14}CO_2$-based fossil fuel $CO_2$ mixing ratios and CO observations. On a larger scale, Levin et al. (2008) examine monthly mean $^{14}CO_2$ observations from two German stations (Schauinsland and Heidelberg), compared against background measurements from Jungfraujoch, to assess the regional fossil fuel $CO_2$ surplus and emphasize the importance of high-precision radiocarbon measurements for quantifying fossil fuel $CO_2$ contributions at a regional scale in Europe. The study by Miller et al. (2012) explores the relationship between fossil fuel $CO_2$ emissions and enhancements in atmospheric mixing ratios of $^{14}CO_2$ and other anthropogenic trace gases. Utilizing a six-year dataset from vertical profiles in the northeast U.S., they separate the fossil and natural components of atmospheric $CO_2$ using apparent emission ratios of various gases to fossil fuel $CO_2$, offering observation-based estimates of national emissions and comparing these with inventory-based estimates. Turnbull et al. (2015) use measurements of $CO_2$, $^{14}CO_2$, and CO from multiple sampling towers around Indianapolis, U.S., to differentiate fossil fuel $CO_2$ from background levels in an urban environment and evaluate the consistency of a bottom-up emission product. More recently, by using radiocarbon observations in $CH_4$ ($\Delta^{14}CH_4$) and $CO_2$ ($\Delta^{14}CO_2$) over London, Zazzeri et al. (2023) reveal that fossil fractions of $CH_4$ and atmospheric mixing ratios of fossil $CO_2$ are consistently higher than those predicted by simulations using emission products such as EDGAR. This discrepancy highlights the potential of $^{14}CO_2$ measurements to refine our understanding of fossil and biospheric $CO_2$ and $CH_4$ partitioning in urban settings, especially when the influence of nuclear power plants is minimal.

Nevertheless, large-scale four-dimensional inversion systems have only recently begun to include $\Delta^{14}CO_2$ as an additional tracer to constrain fossil $CO_2$ emissions. Basu et al. (2016) introduced a novel dual-tracer atmospheric inversion technique that differentiates between biospheric and fossil fuel $CO_2$ fluxes using atmospheric $CO_2$ and $\Delta^{14}CO_2$ measurements over the U.S.

This method not only allows for the estimation of monthly regional fossil fuel $CO_2$ fluxes but also addresses biases in bio-
spheric flux estimates that occur when using traditional $CO_2$-only inversion methods with fixed fossil fuel flux assumptions.
Their approach represents a significant advancement in quantifying regional and national fossil fuel emissions from atmo-
spheric observations. Building upon this study, Basu et al. (2020) presented a more focused analysis in providing national and
sub-national-scale estimates of fossil fuel $CO_2$ emissions, using an extensive observation database of both $CO_2$ and $\Delta^{14}CO_2$.
Graven et al. (2018) conducted an in-depth analysis of fossil fuel $CO_2$ emissions in California, utilizing atmospheric obser-
vations from nine sites and employing the Weather Research and Forecasting model along with the Stochastic Time-Inverted
Lagrangian Transport model (WRF-STILT). The research integrates measurements of $CO_2$ mixing ratio and $\Delta^{14}CO_2$, uniquely
combining these observations with high-resolution emission data from Vulcan v2.2 and EDGARv4.2, aiming to refine estimates
of regional fossil fuel $CO_2$ emissions and explore the impact of various factors such as nuclear industry emissions and air-sea
exchanges on atmospheric $CO_2$ levels. In Europe, Wang et al. (2018) evaluated the potential of a $\Delta^{14}CO_2$ observation network
for estimating regional fossil fuel $CO_2$ emissions through atmospheric inversions. They examined the effectiveness of differ-
ent network configurations, from minimal to very dense setups, in reducing uncertainties in fossil $CO_2$ emissions estimation.
The study used synthetic observations and the LMDZv4 global transport model, paying special attention to representation and
aggregation errors. Establishing a network of both $CO_2$ and $\Delta^{14}CO_2$ measurement stations requires significant investments to
ensure long monitoring periods that allow the identification of sub-annual and sub-national scale variations in fossil $CO_2$ emis-
sions. The Integrated Carbon Observation System (ICOS) atmospheric network includes 39 stations in 14 European countries
and overseas territories. Hourly $CO_2$ atmospheric observations are available for 26 stations, with the earliest data from 2015
when the network was established. However, some of these stations already existed by then, and there is information from
previous years. Fourteen stations measure $\Delta^{14}CO_2$ in 2-week integrated samples analyzed by the ICOS Central Radiocarbon
Laboratory. The ICOS network is expanding to include more stations, and new sampling strategies are being developed to
increase the number of $\Delta^{14}CO_2$ measurements.

In this work, we present the new capabilities of the Lund University Modular Inversion Algorithm (LUMIA) system (Monteil
and Scholze, 2021) to perform simultaneous inversions of atmospheric $CO_2$ and $\Delta^{14}CO_2$ observations as a first attempt to
develop a model capable of supporting the monitoring and verification of fossil $CO_2$ emissions across Europe. Such emissions
monitoring and verification support capacities are essential for assessing compliance with international agreements, such as
the Paris Agreement (UNFCCC, 2016), and for guiding policy decisions aimed at reducing carbon emissions, as described by
Janssens-Maenhout et al. (2020). We perform Observing System Simulation Experiments (OSSEs), recreating the current state
of the ICOS network and its sampling strategy, and using different flux products (as priors and true values) to demonstrate
the performance of the inversion scheme and show its capabilities. We begin by assessing the impact of oceanic fluxes on the
total mixing ratios of $CO_2$ and $\Delta^{14}CO_2$. Then, we evaluate the impact of adding $\Delta^{14}CO_2$ observations on the estimation of
fossil $CO_2$ emissions by comparing the model's ability to recover true fluxes starting from a prior flux set to zero. Finally, with
a more realistic setup, i.e., prior, we evaluate the impact of the prescribed fossil $CO_2$ flux uncertainty and the impact of the
terrestrial isotopic disequilibrium product.

## 2 Theoretical background

The depletion of radiocarbon in the atmosphere due to fossil $CO_2$ emissions has been demonstrated in various studies since the 1950s, primarily through the $\Delta^{14}C$ content of tree rings (Suess, 1955; Tans et al., 1979). Anthropogenic disturbances in atmospheric radiocarbon content, such as those from nuclear bomb tests and nuclear power facilities (Hesshaimer and Levin, 2000), have led to a deeper understanding of the radiocarbon exchange processes between the atmosphere, the biosphere (Hahn et al., 2006), and the ocean (Hesshaimer et al., 1994).

With subsequent advances in measurement and modeling techniques, $\Delta^{14}CO_2$ observations have been used to estimate the fossil $CO_2$ offset within atmospheric $CO_2$ mixing ratios (Levin and Hesshaimer, 2000; Kuc et al., 2003; Naegler and Levin, 2006; Levin and Karstens, 2007; Levin et al., 2008). This is achieved by comparing observations from free troposphere background stations with those from regionally polluted stations, establishing an essential foundation for estimating fossil $CO_2$ emissions using inverse modeling, as will be discussed in the following sections.

### 2.1 Regional transport model

We are using the LUMIA (Lund University Modular Inversion Algorithm) system as described by Monteil and Scholze (2021), modifying the way background mixing ratios ($y^b$ in Equation 1) are calculated by computing a smoothed and detrended average of real observations from the ICOS network for each sampling site. Originally, the LUMIA system was developed to optimize regional Net Ecosystem Exchange (NEE) fluxes over Europe using in situ $CO_2$ observations from the ICOS (Integrated Carbon Observation System) Atmosphere network. In this study, we have extended LUMIA to additionally assimilate $\Delta^{14}CO_2$ observations from the same network and optimize multiple flux categories. This extension introduces a new step in the mass balance of atmospheric transport as follows:

$$y_{CO_2} = y^b_{CO_2} + \sum_c K(\boldsymbol{F}_c) \tag{1a}$$

$$y_{C\Delta^{14}C} = y^b_{C\Delta^{14}C} + \sum_c K(\boldsymbol{\Delta}_c \boldsymbol{F}_c) \tag{1b}$$

where $y_{CO_2}$ and $y_{C\Delta^{14}C}$ represent the modeled mixing ratios of $CO_2$ and $C\Delta^{14}C$, respectively, and $y^b_{CO_2}$ and $y^b_{C\Delta^{14}C}$ denote their background mixing ratios (i.e., the boundary condition), derived from smoothed real observations (see Section 3.3). Since the values of $\Delta^{14}CO_2$ in ‰ (permil) units are not additive (as it represents the change of the $^{14}C : {}^{12}C$ atmospheric ratio relative to an absolute standard of $^{14}C$ from 1950 (Stuiver and Polach, 1977)), we convert all $\Delta^{14}CO_2$ values to values of $CO_2 \times \Delta^{14}CO_2$ (or $C\Delta^{14}C$ for simplification) (Basu et al., 2016). In terms of units, for mixing ratios this would be $C\Delta^{14}C$ ppm ‰, and for fluxes PgC ‰ yr$^{-1}$. Since ‰ only means multiplication by 1000, we drop that factor from $\Delta^{14}C$ into the quantity $C\Delta^{14}C$, expressing it in ppm for mole fractions and PgC yr$^{-1}$ for fluxes to maintain the same order of magnitude and units for $CO_2$ and $C\Delta^{14}C$. For example, a sample with a $CO_2$ mole fraction of 400 ppm and a $\Delta^{14}C$ value of 45‰ would have $C\Delta^{14}C = 18$ppm. Expressed in this way, $C\Delta^{14}C$ becomes additive and can be transported by a model.

The operator $K$ represents the regional transport model (pre-computed footprints, see Section 3.2), which is used to calculate the contribution of surface fluxes $\boldsymbol{F}$ (in each category $c$) to the change of $CO_2$ and $C\Delta^{14}C$ in the atmosphere. $\boldsymbol{F}_c$ in this study corresponds to gridded fluxes with a resolution of $0.5° \times 0.5°$ and 1 hour. In Eq. 1b, the term $\boldsymbol{\Delta}_c$ represents the fraction of $^{14}C$ in the accompanying flux category $\boldsymbol{F}_c$ (Tans et al., 1979; Turnbull et al., 2016).

Expanding the foreground (regional) part of Equation 1 to include the flux categories explicitly yields:

$$\sum_c K(\boldsymbol{F}_c) = K(\boldsymbol{F}_{\mathrm{ff}}) + K(\boldsymbol{F}_{\mathrm{bio}}) + K(\boldsymbol{F}_{\mathrm{oce}}) \tag{2a}$$

where $\boldsymbol{F}_{\mathrm{ff}}$ is the fossil $CO_2$ emissions, $\boldsymbol{F}_{\mathrm{bio}}$ is the net $CO_2$ flux between the atmosphere and terrestrial ecosystems (Net Ecosystem Exchange, NEE, hereafter also called biosphere flux), and $\boldsymbol{F}_{\mathrm{oce}}$ is the atmosphere-ocean $CO_2$ exchanges. Calculating each $K(\boldsymbol{F}_c)$ separately tracks the influence of each category, not just the total. For radiocarbon, the equation looks similar but includes an additional term for the radiocarbon from nuclear facilities:

$$\sum_c K(\boldsymbol{\Delta}_c \boldsymbol{F}_c) = K(\boldsymbol{\Delta}_{\mathrm{ff}} \boldsymbol{F}_{\mathrm{ff}}) + K(\boldsymbol{\Delta}_{\mathrm{atm}}(\boldsymbol{F}_{\mathrm{bio}} + \boldsymbol{F}_{\mathrm{oce}})) + K((\boldsymbol{\Delta}_{\mathrm{bio}} - \boldsymbol{\Delta}_{\mathrm{atm}})\boldsymbol{F}_{\mathrm{bio2atm}}) \tag{2b}$$

$$+ K((\boldsymbol{\Delta}_{\mathrm{oce}} - \boldsymbol{\Delta}_{\mathrm{atm}})\boldsymbol{F}_{\mathrm{oce2atm}}) + K(\boldsymbol{\Delta}_{\mathrm{nuc}} \boldsymbol{F}_{\mathrm{nuc}})$$

$$= K(\boldsymbol{\Delta}_{\mathrm{ff}} \boldsymbol{F}_{\mathrm{ff}}) + K(\boldsymbol{\Delta}_{atm}(\boldsymbol{F}_{\mathrm{bio}} + \boldsymbol{F}_{\mathrm{oce}})) + K(\boldsymbol{F}_{\mathrm{biodis}}) + K(\boldsymbol{F}_{\mathrm{ocedis}}) + K(\boldsymbol{\Delta}_{\mathrm{nuc}} \boldsymbol{F}_{\mathrm{nuc}}) \tag{2c}$$

where $\boldsymbol{\Delta}_{\mathrm{ff}}$ is set equal to -1000‰, indicating that the fossil $CO_2$ does not contain any $\Delta^{14}CO_2$ and therefore dilutes the atmospheric $\Delta^{14}CO_2$ content. $\boldsymbol{\Delta}_{\mathrm{atm}}\boldsymbol{F}_{\mathrm{bio}}$ and $\boldsymbol{\Delta}_{\mathrm{atm}}\boldsymbol{F}_{\mathrm{oce}}$ refer to the exchange of "modern" $C\Delta^{14}C$ between terrestrial ecosystems and the ocean, respectively, with the atmosphere, since $\Delta^{14}C$ in new biomass and the top layer of the ocean would almost match atmospheric $\Delta^{14}C$ ($\Delta_{\mathrm{atm}}$) (Graven et al., 2020). $\boldsymbol{F}_{\mathrm{biodis}}$ and $\boldsymbol{F}_{\mathrm{ocedis}}$ represent the isotopic disequilibrium, or the isotopic difference between the source (ocean or biosphere) and the atmosphere. $\boldsymbol{F}_{\mathrm{biodis}}$ is "old-captured" and $\Delta^{14}C$-enriched $C\Delta^{14}C$ released by heterotrophic respiration ($\boldsymbol{F}_{\mathrm{bio2atm}}$). $\boldsymbol{F}_{\mathrm{ocedis}}$ is "old-captured" and $\Delta^{14}C$-depleted $C\Delta^{14}C$ released through vertical transport of water masses ($\boldsymbol{F}_{\mathrm{oce2atm}}$) (Lehman et al., 2013; Basu et al., 2016). $\boldsymbol{F}_{\mathrm{nuc}}$ is the radiocarbon production due to nuclear activities, mainly from nuclear facilities, since radiocarbon production from nuclear bomb tests has stopped largely (Hesshaimer and Levin, 2000). Converting $\boldsymbol{\Delta}_{\mathrm{nuc}}\boldsymbol{F}_{\mathrm{nuc}}$ to $C\Delta^{14}C$ notation, for modeling purposes, as mentioned above, is achieved through:

$$\boldsymbol{\Delta}_{\mathrm{nuc}} \boldsymbol{F}_{\mathrm{nuc}} = \frac{N}{r_{\mathrm{std}}} \boldsymbol{F}_{\mathrm{nuc}} \tag{3}$$

where $r_{\mathrm{std}}$ is the standard $^{14}C : C$ ratio ($1.176 \times 10^{-12}$), and $N = (975/(\delta^{13}C + 1000))^2$ is the isotope fractionation correction (Stuiver and Polach, 1977). As the $\delta^{13}C$ value, we use the global atmospheric yearly average of $-8$‰ (Basu et al., 2016). Combining equations 1 through 3 for the modeled $CO_2$ and $\Delta^{14}CO_2$ mixing ratios, yields the following:

$$y_{\mathrm{CO}_2} = y_{\mathrm{CO}_2}^{\mathrm{b}} + K(\boldsymbol{F}_{\mathrm{ff}}) + K(\boldsymbol{F}_{\mathrm{bio}}) + K(\boldsymbol{F}_{\mathrm{oce}}) \tag{4a}$$

$$y_{C\Delta^{14}C} = y^{b}_{C\Delta^{14}C} + K(\boldsymbol{\Delta}_{ff}\boldsymbol{F}_{ff}) + K(\boldsymbol{F}_{biodis}) + K(\boldsymbol{F}_{ocedis}) + K(\boldsymbol{\Delta}_{atm}(\boldsymbol{F}_{bio} + \boldsymbol{F}_{oce})) + K(\frac{N}{r_{std}}\boldsymbol{F}_{nuc}) \tag{4b}$$

An additional source of radiocarbon, the cosmogenic production, occurs naturally in the upper atmosphere as a result of cosmic-ray-induced reactions with nitrogen. This term is implicitly included in the background $y^{b}_{C\Delta^{14}C}$.

## 2.2 Observations

We perform inversions for a regional domain ranging from 15°W, 33°N to 35°E, 73°N, as shown in Figure 1. This domain is consistent with those used in previous studies, such as Monteil et al. (2020) and Thompson et al. (2020). The sampling stations depicted in Figure 1 represent the ICOS Atmosphere network for the years 2018-2020, noting that new sampling stations have been added since that period. The ICOS Atmosphere network is a component of ICOS, a European research infrastructure designed to provide long-term, high-quality, and harmonized observations of carbon dynamics. The network includes 33 stations across Europe, all measuring $CO_2$, with 15 of these stations also measuring $\Delta^{14}CO_2$.

There are two sampling strategies used at the ICOS stations: continuous sampling and periodic sampling. Continuous sampling is performed at almost every available sampling height at the station, using commercially available automatic samplers for hourly measurements of, for example, $CO_2$. Periodical sampling, on the other hand, is conducted only at the highest sampling height using flask samplers. These flasks are later analyzed in various ICOS laboratories. Hourly integrated flask samples, collected every three days, serve both as quality control for continuous sampling and for measuring other gases not continuously monitored (e.g. $SF_6$, $H_2$, stable isotopes of $CO_2$), in addition to $\Delta^{14}CO_2$ for the determination of the atmospheric fossil $CO_2$ component through inverse modeling (Levin et al., 2020). Furthermore, a 2-week integrated flask sample is designed to pass air over a NaOH solution, specifically for $\Delta^{14}CO_2$ sampling.

In this paper, we use the continuous 1-hour $CO_2$ and integrated 2-week $\Delta^{14}CO_2$ periodic sampling strategies for the evaluation of LUMIA. A summary of the stations, including their location, sampling height, number and average of measurements, and integration days, is presented in Table 1.

## 2.3 Inverse modeling problem

Atmospheric inverse modeling can be used for a variety of purposes, including the establishment of the initial conditions of a model, the identification of sources and sinks, and the evaluation and improvement of prior emissions (Bocquet et al., 2015). The goal is to estimate the best set of variables (fluxes) consistent with atmospheric measurements of a tracer (e.g. $CO_2$ and $\Delta^{14}CO_2$) in the study domain (observations), given the atmospheric transport that relates the two. In its most basic form, this can be formulated as

$$\boldsymbol{y} = H(\boldsymbol{x}, \boldsymbol{b}) + \boldsymbol{\varepsilon} \tag{5a}$$

**Table 1.** Observation stations used in this study. Included is a summary of the number of observations ($N_{obs}$), the average observations $\pm$ one standard deviation, and the integration time of $\Delta^{14}CO2$ samples for the year 2018, based on data accessible through the ICOS Python API (https://pypi.org/project/icoscp/, accessed February 2023). Stations with zero $N_{obs}$ did not measure or report observations of the corresponding tracer in 2018 to ICOS, but are incorporated into this study for comprehensive analysis.

| Code | Name | Country | Lat (°E) | Lon (°N) | Altitude (m.a.s.l) | Max. samp. height (m.a.g.l) | $N_{obs}$ $CO_2$ | $N_{obs}$ $\Delta^{14}C$ | Avg. $CO_2$ (ppm) | Avg. $\Delta^{14}C$ (‰) | Integration time (days) |
|------|------|---------|-----|-----|----------|-------------|------|------|-------------|-------------|-------------|
| BIR | Birkenes | NO | 58.39 | 8.25 | 219 | 75 | 2616 | – | 421.9 ± 8.0 | – | – |
| CMN | Monte Cimone | IT | 44.19 | 10.70 | 2165 | 8 | 5832 | – | 406.3 ± 6.0 | – | – |
| GAT | Gartow | DE | 53.07 | 11.44 | 70 | 341 | 8784 | 0 | 419.5 ± 10.0 | – | – |
| HEL | Helgoland | DE | 54.18 | 7.88 | 43 | 110 | 1080 | – | 430.4 ± 10.1 | – | – |
| HPB | Hohenpeissenberg | DE | 47.8 | 11.02 | 934 | 131 | 8784 | 17 | 415.6 ± 6.8 | -4.1 ± 2.8 | 13.4 ± 0.5 |
| HTM | Hyltemossa | SE | 56.1 | 13.42 | 115 | 150 | 8784 | 21 | 417.1 ± 8.4 | -3.2 ± 3.2 | 14.0 ± 1.6 |
| IPR | Ispra | IT | 45.81 | 8.64 | 210 | 100 | 8784 | – | 430.0 ± 15.9 | – | – |
| JFJ | Jungfraujoch | CH | 46.55 | 7.99 | 3580 | 5 | 8784 | 15 | 413.1 ± 3.6 | -1.0 ± 3.5 | 14.0 ± 0.0 |
| JUE | Jülich | DE | 50.91 | 6.41 | 98 | 120 | 8784 | – | 423.0 ± 11.2 | – | – |
| KIT | Karlsruhe | DE | 49.09 | 8.42 | 110 | 200 | 8784 | 21 | 428.7 ± 17.5 | -14.1 ± 10.4 | 6.2 ± 0.7 |
| KRE | Křešín u Pacova | CZ | 49.57 | 15.08 | 534 | 250 | 8784 | 13 | 422.0 ± 11.5 | -4.1 ± 3.0 | 13.2 ± 0.6 |
| LIN | Lindenberg | DE | 52.17 | 14.12 | 73 | 98 | 8784 | 5 | 426.0 ± 13.1 | -8.6 ± 6.3 | 14.0 ± 0.0 |
| LMP | Lampedusa | IT | 35.52 | 12.63 | 45 | 8 | 8088 | – | 414.7 ± 4.2 | – | – |
| LUT | Lutjewad | NL | 53.4 | 6.35 | 1 | 60 | 8784 | – | 422.3 ± 12.2 | – | – |
| NOR | Norunda | SE | 60.09 | 17.48 | 46 | 100 | 8784 | 19 | 417.8 ± 8.2 | -0.7 ± 4.2 | 13.3 ± 0.5 |
| OPE | Observatoire pérenne de l'environnement | FR | 48.56 | 5.5 | 390 | 120 | 8784 | 17 | 420.2 ± 9.5 | -3.3 ± 3.5 | 13.5 ± 0.5 |
| OXK | Ochsenkopf | DE | 50.03 | 11.81 | 1022 | 163 | 8784 | 0 | 416.8 ± 6.4 | – | – |
| PAL | Pallas | FI | 67.97 | 24.12 | 565 | 12 | 8784 | 17 | 416.2 ± 7.7 | -1.5 ± 3.5 | 12.9 ± 1.9 |
| PRS | Plateau Rosa | IT | 45.93 | 7.70 | 3480 | 10 | 0 | – | – | – | – |
| PUI | Puijo | FI | 62.91 | 27.65 | 232 | 84 | 1248 | – | 426.6 ± 4.5 | – | – |
| PUY | Puy de Dôme | FR | 45.77 | 2.97 | 1465 | 10 | 8784 | – | 414.0 ± 5.4 | – | – |
| RGL | Ridge Hill | GB | 52.0 | -2.54 | 199 | 90 | 8784 | – | 413.4 ± 6.1 | – | – |
| SAC | Saclay | FR | 48.72 | 2.14 | 160 | 100 | 8784 | 12 | 420.5 ± 10.5 | -2.7 ± 6.4 | 16.5 ± 4.3 |
| SMR | Hyytiälä | FI | 61.85 | 24.29 | 181 | 125 | 8784 | – | 416.8 ± 8.5 | – | – |
| SSL | Schauinsland | DE | 47.92 | 7.92 | 1205 | 35 | 0 | – | – | – | – |
| STE | Steinkimmen | DE | 53.04 | 8.46 | 29 | 252 | 8784 | 13 | 421.9 ± 11.5 | -6.7 ± 4.2 | 13.5 ± 1.6 |
| SVB | Svartberget | SE | 64.26 | 19.77 | 269 | 150 | 8784 | 13 | 416.1 ± 8.0 | -0.9 ± 3.1 | 15.6 ± 1.9 |
| TOH | Torfhaus | DE | 51.81 | 10.54 | 801 | 147 | 8784 | – | 417.6 ± 7.8 | – | – |
| TRN | Trainou | FR | 47.96 | 2.11 | 131 | 180 | 8784 | 11 | 419.4 ± 8.6 | -4.7 ± 4.8 | 14.7 ± 3.3 |
| UTO | Utö - Baltic sea | FI | 59.78 | 21.37 | 8 | 57 | 8784 | – | 416.2 ± 8.0 | – | – |
| WAO | Weybourne | GB | 52.95 | 1.12 | 31 | 10 | 8784 | – | 413.4 ± 6.1 | – | – |
| WES | Westerland | DE | 54.92 | 8.31 | 12 | 14 | 8784 | – | 416.2 ± 2.7 | – | – |
| ZSF | Zugspitze | DE | 47.42 | 10.98 | 2666 | 3 | 0 | – | – | – | – |

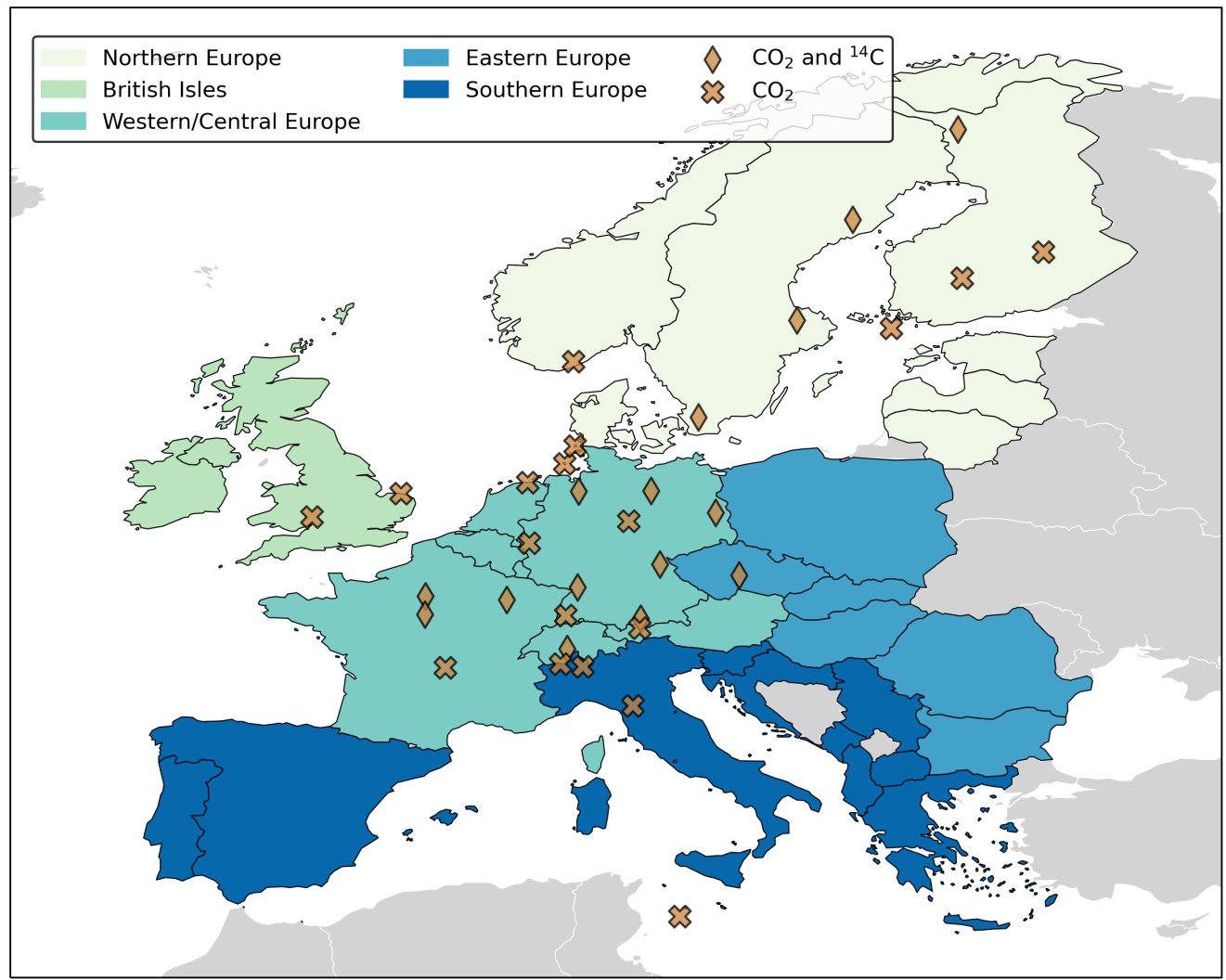

**Figure 1.** Study domain and location of the ICOS Atmosphere network sampling stations used in this paper. The regions will be used for the analysis and discussion of the results.

where the control vector $x$ contains the variables to be estimated and the observation vector $y$ contains the observations (atmospheric mixing ratios). $H$ is the observation operator, which includes the transport model and any additional processing of observations, such as accounting for the boundary conditions and variables $b$, which we will not optimize. $\varepsilon$ is the error vector that includes the errors in the observations, the transport model, and the control vector. By pre-subtracting the prior model estimate for the observations, $y^{\text{apri}}$, Equation 5a can be rewritten as $y = y^{\text{apri}} + H(x) + \varepsilon$, with $y^{\text{apri}}$ the *a priori* model estimate for the observations, computed following Equation 4. In this case, Equation 5a can be simplified as:

$$\delta_{\boldsymbol{y}} = \mathbf{H}\boldsymbol{x} + \boldsymbol{\varepsilon} \tag{5b}$$

where $\delta_{\boldsymbol{y}} = \boldsymbol{y} - \boldsymbol{y}^{\mathrm{apri}}$ is the vector storing the prior model-data mismatches. In our case, the atmospheric transport is linear thus $H(\boldsymbol{x})$ can be rewritten as $\mathbf{H}\boldsymbol{x}$, where $\mathbf{H}$ is the Jacobian of $H$ (Monteil and Scholze, 2021).

There are multiple approaches to solving the inverse modeling problem. In this paper, and in general in LUMIA, we use the variational approach, in which the control vector $\boldsymbol{x}$ that minimizes the cost function in Eq. 6 is sought iteratively by minimizing the mismatch between the model outputs and the observations (Chatterjee and Michalak, 2013; Rayner et al., 2019; Scholze et al., 2017):

$$J(\boldsymbol{x}) = \frac{1}{2}\left(\boldsymbol{x} - \boldsymbol{x}^{\mathrm{b}}\right)^{T}\mathbf{B}^{-1}\left(\boldsymbol{x} - \boldsymbol{x}^{\mathrm{b}}\right) + \frac{1}{2}\left(\mathbf{H}\boldsymbol{x} - \delta_{\boldsymbol{y}}\right)^{T}\mathbf{R}^{-1}\left(\mathbf{H}\boldsymbol{x} - \delta_{\boldsymbol{y}}\right) \tag{6}$$

where $\mathbf{B}$ is the prior uncertainty covariance matrix, and $\mathbf{R}$ is the observational uncertainty covariance matrix (defined as a diagonal matrix) controlling the weight of each observation in the model-data mismatch $\delta_{\boldsymbol{y}}$ and the target variable ($\boldsymbol{x}^{\mathrm{b}}$) in the optimization. The iterative procedure searches for the value of $\boldsymbol{x}$ that minimizes $J(\boldsymbol{x})$, that is, the value of $\boldsymbol{x}$ for which the gradient $(\nabla_x J)$ is equal to zero. To reduce the number of iterations and large matrix multiplications, the optimization is performed on a preconditioned control vector $\boldsymbol{\omega} = \mathbf{B}^{-1/2}(\boldsymbol{x} - \boldsymbol{x}^{\mathrm{b}})$. More information on preconditioning can be found in Monteil and Scholze (2021).

### 2.3.1 Construction of the control vector ($\boldsymbol{x}$)

The control vector $\boldsymbol{x}$ contains the set of parameters adjustable by the inversion, which are offsets to the different sources and sinks of $CO_2$ and $\Delta^{14}CO_2$ that we want to estimate. From Equation 4, our main interest is to optimize the fossil $CO_2$ flux $\boldsymbol{F}_{\mathrm{ff}}$. However, since through the radiocarbon cycle, we can separate fossil and biogenic $CO_2$, we also need to optimize the fluxes from the biosphere ($\boldsymbol{F}_{\mathrm{bio}}$), as well as the isotopic disequilibrium $\boldsymbol{F}_{\mathrm{biodis}}$, to reduce the uncertainty of these two terms, which can have an important impact on the inversion result. The remaining fluxes ($\boldsymbol{F}_{\mathrm{nuc}}$, $\boldsymbol{F}_{\mathrm{oce}}$, and $\boldsymbol{F}_{\mathrm{ocedis}}$) are prescribed.

To limit the computational requirements, we do not solve directly for the high-resolution fluxes (e.g. $0.5° \times 0.5°$ and 1-hourly) used in the transport model, but for weekly offsets for 2500 clusters of grid cells. Appendix B describes the clustering algorithm in more detail and the script can be found at *lumia/Tools/optimization_tools.py* of the LUMIA source code provided as an asset. In short, it groups contiguous grid cells, depending on how sensitive the observation network is to their emissions: grid cells directly upwind of the sampling stations are optimized at the native resolution of $0.5°$, but in parts of the domain not well sampled by the observations (e.g., North Africa, Turkey), the resolution drops down to $5° \times 3.5°$ (see Figure 2).

Equation 5b can be rewritten as:

$$\delta_{\boldsymbol{y}} = \sum_{\mathrm{c}}\mathbf{H}\boldsymbol{x}_{\mathrm{c}} \tag{7}$$

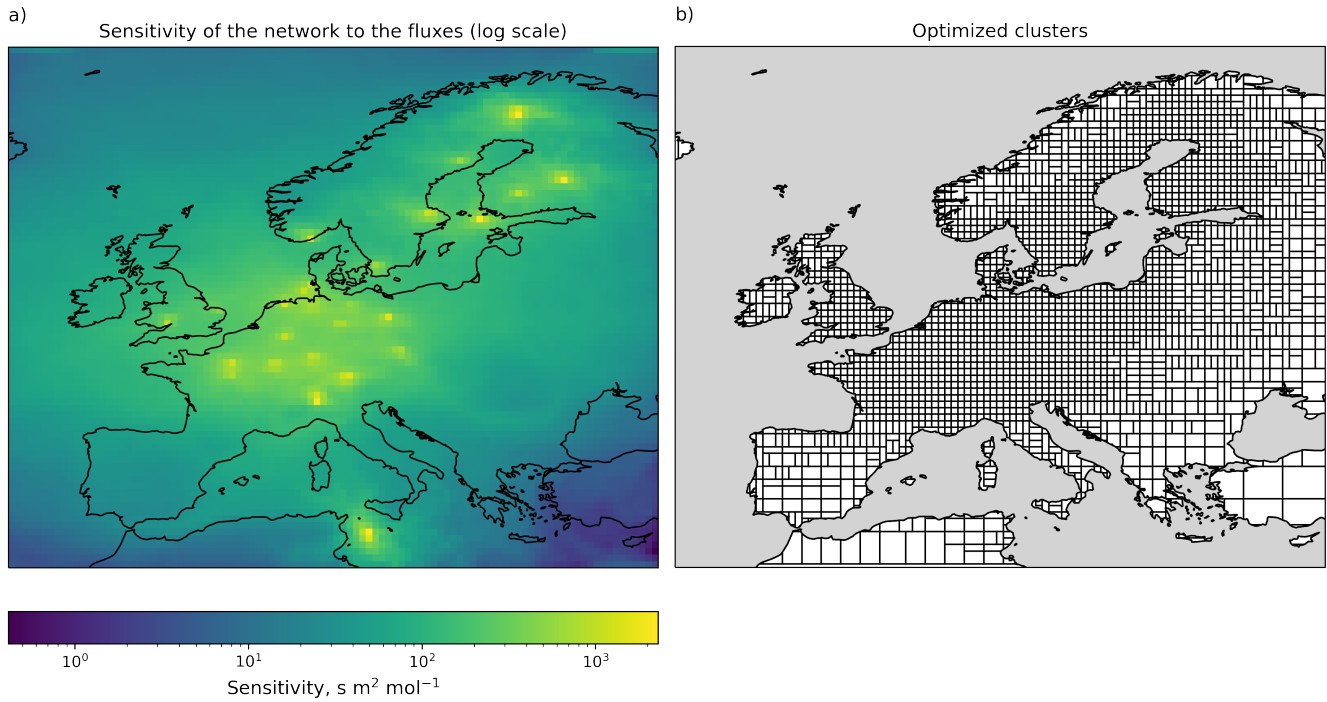

**Figure 2.** Visual representation of a) the sensitivity of the observation network to each grid-cell (in logarithmic scale) and b) the optimized clusters and their variable spatial resolution.

where $\mathbf{H}$ is the Jacobian matrix of the observation operator with dimensions $(n_{obs}, n_{p_{opt}} * n_{t_{opt}})$, and $\boldsymbol{x}_{\mathrm{c}}$, with dimensions $(n_{p_{opt}} * n_{t_{opt}})$, represents the portion of the control vector $\boldsymbol{x}$ that contains offsets for the optimized categories c. Thus, $\boldsymbol{x}_{\mathrm{c}}$ is built from the relative contribution of each model time step $t_{mod}$ (1 hour) and of each grid cell $p_{mod}$ ($0.5° \times 0.5°$) to each optimized time step $t_{opt}$ and cluster $p_{mod}$. Here, $n_{t_{opt}}$ and $n_{p_{opt}}$ represent the number of optimized intervals (weekly) and grid cell clusters (e.g. 2500 for biosphere), respectively.

### 2.3.2 Construction of the prior error covariance matrix ($\mathbf{B}$)

Our matrix $\mathbf{B}$ is constructed such that we first determine the spatio-temporal structure of the uncertainties, which is then scaled to match the reported uncertainties. Since we optimize for offsets, the prior control vector $\boldsymbol{x}^{\mathrm{b}}$ contains only zeros (so $\mathbf{F}_c = \mathbf{F}_c^0$). The uncertainties in $\boldsymbol{x}^{\mathrm{b}}$ are given by the error covariance matrix $\mathbf{B}$. We assume no correlation between different categories and different tracers. Therefore, sections of $\mathbf{B}$ specific to each tracer/category can be constructed independently. We do this in three steps:

1. Construct a vector of variances (diagonals of $\mathbf{B}$), which contain the spatio-temporal pattern of the uncertainties.

2. Construct the covariances based on spatial and temporal correlation functions. Specifically, the covariances are set following $cov(\boldsymbol{x}_i, \boldsymbol{x}_j) = \sigma_{\boldsymbol{x}_i}\sigma_{\boldsymbol{x}_j} e^{-(d(p_i,p_j)/L_h)^2} e^{-|t_i-t_j|/L_t}$, with $d(p_i, p_j)$ the great circle distance between the center of the clusters (area-weighted average of the center-coordinates of the grid-cells in the cluster), and $|t_j - t_i|$ the temporal distance between $\boldsymbol{x}_i$ and $\boldsymbol{x}_j$. $i$ and $j$ refer to the position in the control vector with assigned space/time coordinates, i.e. $\boldsymbol{x}_i$ has coordinates $(p_i, t_i)$, and $\boldsymbol{x}_j$ has coordinates $(p_j, t_j)$. $L_h$ and $L_t$ represent the horizontal and temporal correlation lengths, respectively.

3. Scale the entire (section of the) **B** matrix by a uniform scaling factor to match a prescribed category-specific annual uncertainty value $\delta \boldsymbol{F}_c$.

The values of the correlation lengths $L_h$ and $L_t$, as well as the scaling factors $\delta \boldsymbol{F}_c$ are provided in Section 3.3.1. For constructing the vector of variances ($\sigma_x^2$), two approaches were used:

– For fossil $CO_2$ emissions $\boldsymbol{F}_{\text{ff}}$, the variance is set to $\sigma_{p,t,c}^2 = |\sum_{i,j,t_{mod}} \boldsymbol{F}_{i,j,t_{mod}}^c|^2$, where $\sigma_{p,t,c}^2$ is the variance corresponding to the control vector elements for the interval $t$ and spatial cluster $p$ of category $c$. The spatial coordinates $i$ and $j$ are the ensemble of grid cells that are within the cluster $p$, and the temporal coordinate $t_{mod}$ is the ensemble of 1-hourly model time steps that are within the (weekly) optimization interval $t$. For instance, if the cluster $p$ groups four model grid cells, the variance $\sigma_{p,t,c}^2$ will be calculated over 672 flux components (4 grid cells, seven days with 24 hourly time steps).

– For the other fluxes, the procedure is similar, but the formula is $\sigma_{p,t,c}^2 = \sqrt{|\sum_{i,j,t_{mod}} \boldsymbol{F}_{i,j,t_{mod}}^c|}$.

The rationale behind these formulas is to scale the uncertainties to the prior estimate of the fluxes (assuming that very low prior fluxes should imply low prior uncertainties) but avoid artificially low errors in instances where negative and positive fluxes compensate each other (i.e., NEE, in the spring and autumn times). Furthermore, the location of fossil $CO_2$ emissions is relatively better known. Therefore, the formula used for fossil $CO_2$ emissions concentrates the uncertainties more at the location of prior emissions than that used for the other categories. Regardless of the formula used for determining the variance, it is scaled afterward to match the target uncertainty reported in Table 2.

## 3 Observing System Simulation Experiments (OSSEs)

To assess the performance of the inversion system, we designed and performed a series of perfect transport Observing System Simulation Experiments (hereafter called OSSEs). In OSSEs, the impact of new observing systems, existing system configurations, observing strategies, and new data optimization on the quantification of the target variables is evaluated (Hoffman and Atlas, 2016). This is done by generating a set of simulated observations, called synthetic observations, from a set of reasonable but arbitrary fluxes, $\boldsymbol{F}_c^{\text{t}}$, considered 'true' fluxes in the OSSE. Then, by using fluxes from different models or products as prior fluxes ($\boldsymbol{F}_c$), we investigate the ability of an inverse modeling system to reconstruct the true fluxes consistent with the model setup (e.g., prescribed uncertainties, error structure), making assumptions such as perfect transport and perfect boundary

condition. Since they totally neglect the systematic model and representation errors, which should be accounted for in a real inversion, perfect transport OSSEs lead to overly optimistic results and should be interpreted with care. They are, however, well suited to our objective here, which is to test the robustness of our implementation of the dual-tracer inversion. In the following sections, we describe the different data sets, model setups, assumptions, and experiments used in this study.

### 3.1 True and prior fluxes

We use a set of fluxes commonly used in this kind of inverse problem with a high horizontal and temporal resolution ($0.5°$ $\times\ 0.5°$ and 1 hour, respectively) to generate our synthetic observations. For the $CO_2$ fluxes, we use EDGARv4.3 emission database (Janssens-Maenhout et al., 2019) distributed spatially and temporally based on fuel type, category, and country-specific emissions, using the COFFEE approach (Steinbach et al., 2011) (EDGAR in Table 3, see (Gerbig and Koch, 2021b)) as $F_{ff}^t$ for the base year 2018. For $F_{bio}^t$, we use a simulation of the LPJ-GUESS vegetation model (Smith et al., 2014) (LPJ-GUESS in Table 3, see (Wu, 2023)), and for $F_{oce}^t$, we use the Jena Carbo-Scope v1.5 product (Rödenbeck et al., 2013). We use the terrestrial and ocean disequilibrium and nuclear fluxes from Basu et al. (2020) as our $F_{biodis}^t$ (BASU in Table 3), $F_{ocedis}^t$ and $F_{nuc}^t$, respectively.

As prior fluxes, we use products that followed different methodologies and schemes, with different spatial and temporal structures than the true fluxes, to make the implementation more realistic. For $F_{ff}$, we use a version of ODIAC (Open-source Data Inventory for Anthropogenic $CO_2$) (ODIAC in Table 3, see (Oda and Maksyutov, 2020)) with a 1km $\times$ 1km spatial and monthly temporal resolution. Thus, our prior fossil $CO_2$ fluxes include monthly variability but do not resolve the daily cycle (Oda et al., 2018). We also prepare a flat-year average version of this product (FlatODIAC in Table 3). For $F_{bio}$, we use a product from simulations of the VPRM vegetation model (Mahadevan et al., 2008; Thompson et al., 2020) (VPRM in Table 3, see (Gerbig and Koch, 2021a)). Due to the lack of an alternative product for the $F_{biodis}$, we generate our own prior by calculating a series of randomly perturbed versions of the true flux following their prescribed uncertainties and their horizontal and temporal correlations (RndBASU in Table 3). This perturbation is done by adding a random perturbation to the control vector and transforming such vector to the flux space. All fluxes are gridded to $0.5° \times 0.5°$ and 1-hour resolution by the nearest neighbor interpolation.

### 3.2 Observation footprints (FLEXPART)

Similar to Monteil and Scholze (2021), we compute the regional transport (e.g. operator $H$ in Equation 4) using the FLEX-PART 10.4 Lagrangian transport model (Pisso et al., 2019). For each observation, FLEXPART computes a "footprint", i.e. a vector containing the sensitivity of the observation to changes in the surface fluxes. The footprints are pre-computed and then used throughout the subsequent steps of the inversion (see Monteil and Scholze (2021) for further details). The FLEXPART simulations were driven by ERA5 reanalysis data at a horizontal resolution of $0.5° \times 0.5°$ and 1-hourly temporal resolution (Hersbach et al., 2018). The footprints were computed differently for the $CO_2$ and $\Delta^{14}CO_2$ observations. For $CO_2$, we computed a set of footprints for each observation up to 14 days backward in time, following the approach from Monteil and Scholze (2021). Integrated $\Delta^{14}CO_2$ observations (Section 2.2) quantify the $\Delta^{14}C$ value of atmospheric $CO_2$ throughout 1 to 3 weeks

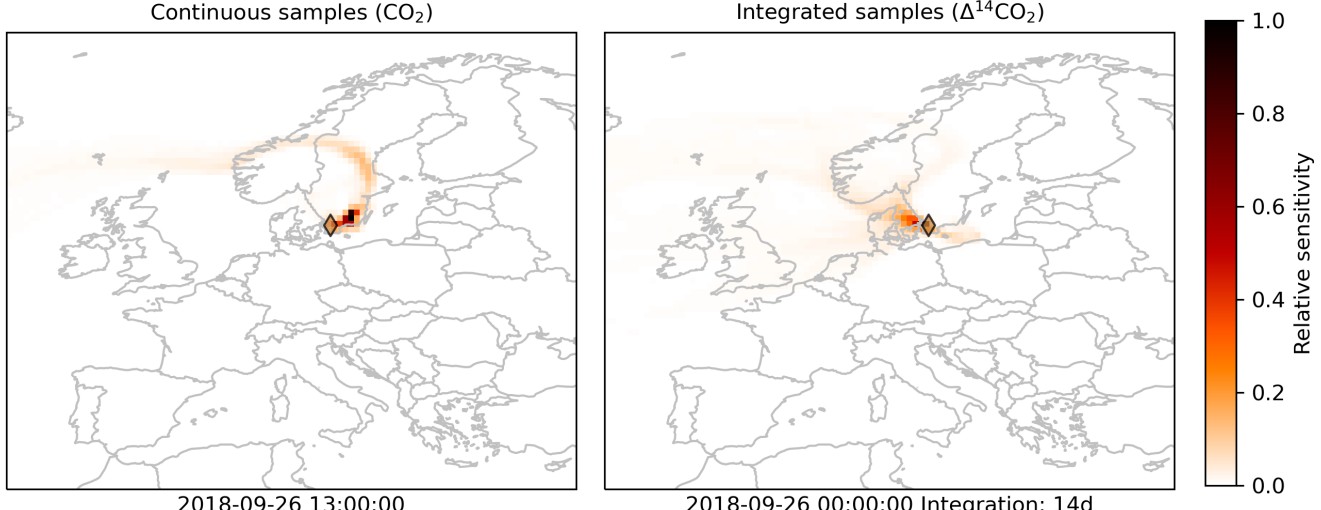

**Figure 3.** Examples of so-called (pre-calculated) footprints for $CO_2$ (left) and $\Delta^{14}CO_2$ (right) at the Hyltemossa ICOS station. The maps show the sensitivity of the respective atmospheric tracer at the sampling site to the surface fluxes over the regional domain up to two weeks before the observation. The left panel displays the sensitivity of $CO_2$ at the indicated sampling time and shows influences by surface fluxes from the North Atlantic through Scandinavia, while the right panel demonstrates the dispersed sensitivity of a 14-day integrated $\Delta^{14}CO_2$ sample across Northwestern Europe and the Baltic region. The two maps illustrate the distinct spatial integration of the two tracers over time.

(see Table 1). We account for this in FLEXPART by distributing the FLEXPART particles released over the whole integration period of the observations. The simulations are then carried on for (up to) 14 days backward in time from the start of the integration period. A Python code was developed to run FLEXPART and post-process the footprints for being used in LUMIA (https://github.com/lumia-dev/runflex). In Figure 3, we show an example of an observation footprint for $CO_2$ and $\Delta^{14}CO_2$ at the Hyltemossa ICOS station in southern Sweden. The $CO_2$ footprint (left panel of Fig. 3) shows how the observation of

June 26th, 2018 at 13:00 LT is sensitive to fluxes from the North Atlantic, passing through Norway, Sweden, and finally from Sweden's East and South coasts close to the Baltic Sea. The $\Delta^{14}CO_2$ aggregated footprint, on the other hand, shows a more spread sensitivity due to the long integration time, collecting fluxes from Southern Norway, Northwestern Europe, and the Baltic.

### 3.3 Synthetic observations and background mixing ratios

We generate mixing ratio time series for one year for each of the stations according to the current setup of the ICOS Atmosphere network as described in Section 2.2. For replicating the most realistic conditions of the sampling frequency, we use real sampling and integration times (in the case of radiocarbon) from the stations, taking for each one the sampling times for 2018. In this way, we account for the sampling gaps and the differences in integration times commonly produced due to maintenance,

and general operational eventualities. For stations with the number of observations, $N_{obs}$, equal to zero in Table 1, we set fixed sampling and integration times (14 days). Most of these stations were already in operation in 2018, but some were not yet labeled as ICOS stations (e.g. Schauinsland) or had not implemented and or started the tracer measurement (e.g. $\Delta^{14}CO_2$ at Ochsenkopf).

Following Monteil and Scholze (2021), we select the $CO_2$ observation times according to the sampling station's elevation to guarantee the model's best representation. For sampling stations located under 1000 m.a.s.l, we select the times when the boundary layer is most likely well developed, from 11:00 to 15:00 LT. For the contrary case, we take the times around midnight, from 22:00 to 2:00 LT, where the boundary layer is most likely below the sampling intake, or in other words, is sampling the free troposphere. For our OSSEs, we use the same transport model (i.e. the pre-computed observation footprints from FLEXPART) to generate the synthetic observations and perform the inversions. Therefore, this data selection is not strictly necessary for this study, but we want to replicate the conditions of a real inversion. Since we are using the same background mixing ratio for the synthetic observations and the simulated prior and posterior observations (i.e. we are assuming a perfect boundary condition), we simplify the calculation of it by computing a smoothed and detrended weekly (for $CO_2$) and monthly (for $\Delta^{14}CO_2$) average of the real observations (ICOS et al., 2023) for each sampling site. For sampling sites, for which there are for some reason no real observations for the year 2018 in the ICOS database (e.g. $\Delta^{14}CO_2$ measurements were not yet implemented or were not yet part of ICOS), we took the observations from the nearest year available to calculate the background.

We then perform a forward run of the model using the true fluxes mentioned in Section 3.1 to calculate the corresponding "true" $CO_2$ and $\Delta^{14}CO_2$ mixing ratio time series and add the background value corresponding to each site, observation time and tracer. To weaken the assumption of a perfect transport and boundary condition, we add a random perturbation to the synthetic observations. This random perturbation is equal to $\boldsymbol{y}^{so*} = \boldsymbol{y}^{so} + \varepsilon \times \xi$, where $\boldsymbol{y}$ is the synthetic observation, $\varepsilon$ is the observation error (both the instrumental and representativity errors, see Section 3.3.1 below), and $\xi$ is a standard normal random vector. In this way, the added perturbation is based on the observation error. Figure 4 shows the synthetic $CO_2$ and $\Delta^{14}CO_2$ observation time-series and the components of each flux at Hyltemossa station. As mentioned in Section 2.1, we convert all radiocarbon values to $C\Delta^{14}C$ values. On the side of the observation, we do this by applying the following equation:

$$[C\Delta^{14}C] = \frac{[\Delta^{14}CO_2] \times [CO_2]}{1000} \tag{8}$$

In a real setup, this would imply having paired $CO_2$ and $\Delta^{14}CO_2$ observations, and in the case of the integrated samples, this would mean having an average of $CO_2$ observations along the integration period of the $\Delta^{14}CO_2$ sample. However, since we are using synthetic observations, we transported the $CO_2$ fluxes using the $\Delta^{14}CO_2$ footprints and stored the values to convert back and forth between $\Delta^{14}CO_2$ and $C\Delta^{14}C$ units, ‰ and ppm, respectively.

As can be seen from Figure 4, both $\boldsymbol{F}_{oce}^{t}$ and $\boldsymbol{F}_{ocedis}^{t}$ have virtually no impact on the mixing ratios at the Hyltemossa station (and all other stations used in our setup, not shown), hence we decided not to include these components in the control vector, i.e. we transport them but do not optimize them further.

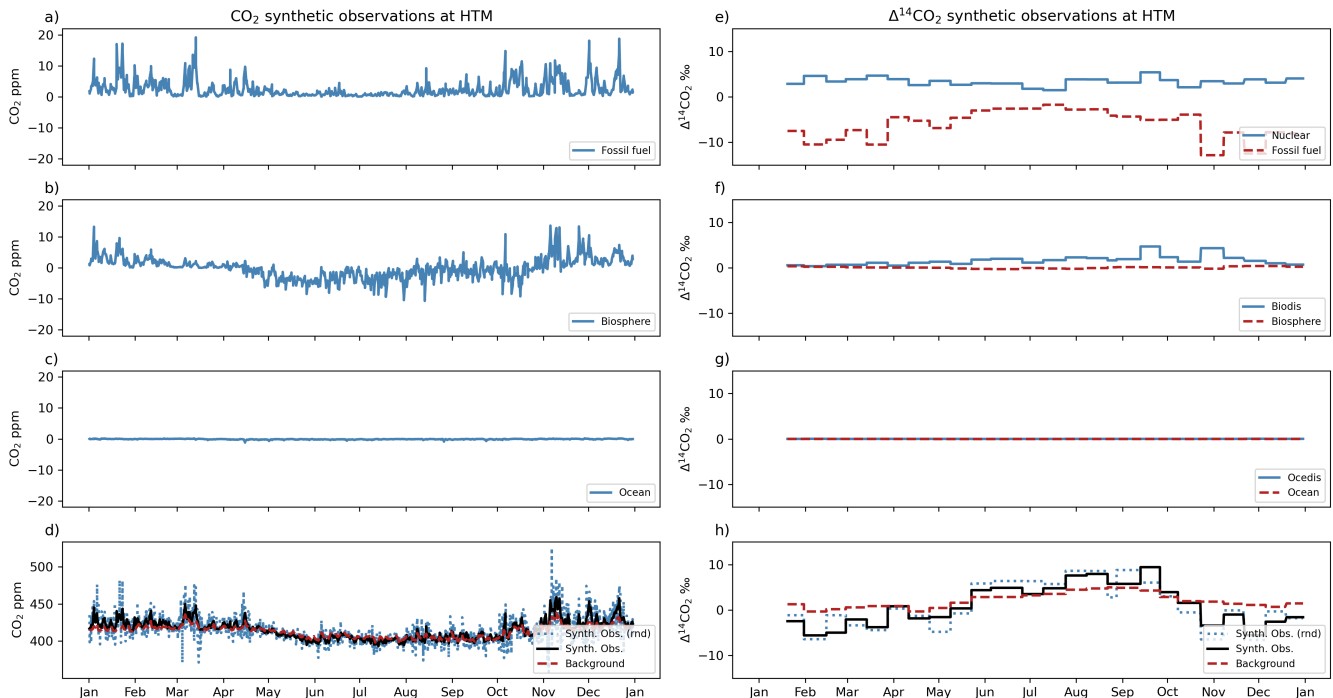

**Figure 4.** Synthetic observations of $CO_2$ and $\Delta^{14}CO_2$ at the HTM station over a one-year period. Panels a) to d) display $CO_2$ mixing ratio variations due to different sources: a) fossil fuel, b) biosphere, c) ocean, and d) combined synthetic observations with random perturbation (blue dotted line) against the background mixing ratios (red dashed line). Panels e) to h) illustrate $\Delta^{14}CO_2$ variations: e) nuclear and fossil fuel, f) biospheric disequilibrium and biosphere, g) ocean disequilibrium and ocean, and h) total synthetic observations with random perturbation (blue dotted line) compared to the background (red dashed line). The blue solid and dashed lines represent the synthetic observations without and with random noise added, respectively.

### 3.3.1 Experiments and inversion setup

To make the inversions comparable, we keep the same inversion setup for all the experiments. Table 2 summarizes the main model parameter values. We choose a Gaussian horizontal correlation and an exponential temporal correlation for the prior flux uncertainties (See Section 2.3.1). Since our main purpose in this study is to demonstrate that our multi-tracer inversion

system is capable of estimating both the fossil $CO_2$ emissions and natural $CO_2$ fluxes, we choose prior uncertainty values that are reasonable and consistent with other studies. The prior uncertainties are assigned as follows: for $\boldsymbol{F}_{\text{ff}}$, we use the difference between the annual budgets for the whole study domain from ODIAC (1.26 PgC yr$^{-1}$) (Oda and Maksyutov, 2020) and from an emissions product based on EDGARv4.3 (1.47 PgC yr$^{-1}$) (Gerbig and Koch, 2021b) as a reference to define its uncertainty (Basu et al., 2016). We use $150\%$ (0.3 PgC yr$^{-1}$) of the difference as the base uncertainty for all the experiments, and we select

two extra values to evaluate the impact of the prescribed uncertainty on the inversion: $50\%$ of the difference (0.1 PgC yr$^{-1}$) and the exact difference of 0.21 PgC yr$^{-1}$ ($100\%$). For $\boldsymbol{F}_{\text{bio}}$ we choose the $25\%$ (0.37 PgC yr$^{-1}$) of the monthly prior (Monteil and

**Table 2.** Parameter setup used in all the inversions performed in this study.

| Fluxes | | | | | | |
|---|---|---|---|---|---|---|
| Flux category | Horizontal correlation | Temporal correlation | Prior uncertainty (PgC yr$^{-1}$) | Error structure | Optimization interval (days) | Grid points |
| $F_{ff}$ | 200-g | 1-e-monthly | 0.30 | log | 7 | 2500 |
| $F_{bio}$ | 500-g | 1-e-monthly | 0.37 | sqrt | 7 | 2500 |
| $F_{biodis}$ | 1000-g | 2-e-monthly | 0.22 | sqrt | 7 | 500 |
| Observations | | | | | | |
| | Tracer | Type of sampling | Prior uncertainty | | | |
| | $CO_2$ | Continuous 1-hour | Weekly moving standard deviation | | | |
| | $\Delta^{14}CO_2$ | Integrated 2-weekly | Constant 0.8 ppm ‰ | | | |

Scholze, 2021), and 30% (0.22 PgC yr$^{-1}$) of the annual budget for $F_{biodis}$ (Basu et al., 2020). We optimize all the categories at the same temporal resolution but at a higher horizontal resolution for $F_{ff}$ and $F_{bio}$ (2500 points) than for $F_{biodis}$ (500 points). To set up the observation error, which includes the instrumental and the representativity errors, we use different methods for

the $CO_2$ and the $\Delta^{14}CO_2$. For $CO_2$, where the error of representativity is usually larger than the instrumental error, we apply a weekly moving standard deviation to each observation i.e. the prior error of each observation is equal to the standard deviation of the observations in a time window of $\pm 3.5$ days around that observation. In this way, we account for the changes in the $CO_2$ mixing ratios according to the local site conditions. For instance, at a background station such as Jungfraujoch (JFJ) on the top of the Swiss Alps, the observation error ranges from 0.9 to 29.2ppm (mean value of $9.3 \pm 4.0$ppm), while at polluted

sites such as Saclay (SAC) just outside Paris, the $CO_2$ mixing ratios change rapidly and the error ranges from 5.9 to 215.5ppm (mean value of $55.8 \pm 40.7$ppm). For $\Delta^{14}CO_2$ on the other hand, the instrumental error is larger than the representativity error, we use a constant value of 0.8ppm in C$\Delta^{14}$C units or $1.91 \pm 0.05$‰ in $\Delta^{14}CO_2$ units, calculated using Equation 8.

We perform one forward run and six inversions, summarized in Table 3. We generate the synthetic observations and evaluate the impact of $F_{oce}^{t}$ and $F_{ocedis}^{t}$ on the total synthetic observations as described in Section 3.3 with the forward run (SYNTH).

Starting with the inversions, we perform two experiments to test the impact of having $\Delta^{14}$C observations (ZBASE and ZCO2Only). We use the prior $F_{ff}$ and $F_{bio}$ set to zero (both in the spatial and temporal domain) with a prior uncertainty setup based on ODIAC and VPRM, respectively. The reason for using prior fluxes set to zero is that the flux products for both categories can have spatial and temporal distributions similar to their respective true values, making it easier for the model to retrieve the true fluxes. Instead, we set the values to zero but give the model some information through the prior uncertainty

setup. The remaining fluxes are prescribed and set to their true values. We assimilate both $CO_2$ and $\Delta^{14}$C observations for

**Table 3.** Inversions performed in this work.

| Simulation | $F_{ff}$ | $F_{bio}$ | $F_{biodis}$ | Optimized fluxes | Tracers | Run |
|---|---|---|---|---|---|---|
| SYNTH | EDGAR | LPJ-GUESS | BASU | None | $CO_2$, $\Delta^{14}CO_2$ | Forward |
| ZBASE | ZEROFossil | ZEROBio | BASU | $F_{ff}$, $F_{bio}$ | $CO_2$, $\Delta^{14}CO_2$ | Inversion |
| ZCO2Only | ZEROFossil | ZEROBio | BASU | $F_{ff}$, $F_{bio}$ | $CO_2$ | Inversion |
| BASE0.1 | ODIAC | VPRM | RndBASU | $F_{ff}$, $F_{bio}$, $F_{biodis}$ | $CO_2$, $\Delta^{14}CO_2$ | Inversion |
| BASE0.21 | ODIAC | VPRM | RndBASU | $F_{ff}$, $F_{bio}$, $F_{biodis}$ | $CO_2$, $\Delta^{14}CO_2$ | Inversion |
| BASE(0.3) | ODIAC | VPRM | RndBASU | $F_{ff}$, $F_{bio}$, $F_{biodis}$ | $CO_2$, $\Delta^{14}CO_2$ | Inversion |
| BASENoBD | ODIAC | VPRM | BASU | $F_{ff}$, $F_{bio}$ | $CO_2$, $\Delta^{14}CO_2$ | Inversion |

ZBASE and only $CO_2$ observations for ZCO2Only. In the second set of inversions, we use a more realistic setup. In the first, BASE, we simulate a complete and realistic inversion setup, assimilating $CO_2$ and $\Delta^{14}C$ observations and optimizing $F_{ff}$, $F_{bio}$, and $F_{biodis}$. In the BASE experiments, we change the prescribed prior uncertainty of $F_{ff}$ (0.1, 0.21 and 0.3 PgC yr$^{-1}$) to evaluate its impact on the optimization.

In the final inversion, BASENoBD, we prescribe $F_{biodis}$ (i.e., the true value in this context) instead of optimizing it. The terrestrial disequilibrium term ($F_{biodis}$) is challenging to estimate due to the large uncertainties associated with heterotrophic respiration fluxes and the age of respired carbon (Basu et al., 2016). These uncertainties can vary significantly depending on the vegetation model or methodology used. We compare the posterior $F_{ff}$ of this experiment with the one of the BASE experiment (in which $F_{biodis}$ is optimized), to evaluate the impact of the prior $F_{biodis}$ product on the posterior $F_{ff}$. By keeping $F_{biodis}$ fixed

in BASENoBD, we can assess how much of the error in the posterior $F_{ff}$ of BASE comes from the additional optimization of $F_{biodis}$.

## 4    Results

### 4.1    Impact of $F_{oce}$ and $F_{ocedis}$

We start by testing the impact of ocean-related fluxes ($F_{oce}^t$ and $F_{ocedis}^t$) in the total synthetic observations by performing a

forward simulation (SYNTH in Table 3). Figure 4, shows the results from this forward simulation and the contribution of each flux category to the mixing ratios of both tracers for the Hyltemossa (HTM) station. The results show that the contribution of the ocean and ocean disequilibrium fluxes to the total mixing ratio is below the error assigned to the synthetic observations. For $CO_2$, the average contribution is -0.07 $\pm$ 0.12 ppm (for an average observation error of 10.0 $\pm$ 5.7 ppm) at HTM, -0.07 $\pm$ 0.15 ppm (average obs. error 9.8 $\pm$ 9.0 $CO_2$ ppm) at all stations. For $\Delta^{14}CO_2$, the average contribution due to $F_{oce}$ is -0.009

$\pm$ 0.009‰ (average obs. error 1.9 $\pm$ 0.04‰) at HTM and -0.007 $\pm$ 0.007‰ (average obs. error 1.9 $\pm$ 0.05‰) at all stations. Similarly, the contribution due to $F_{ocedis}$ is 0.016 $\pm$ 0.009‰ at HTM and 0.02 $\pm$ 0.017‰ at all stations. Due to the low impact

of ocean-related fluxes, we prescribe them in the inversions along with $F_{nuc}$ and optimize only $F_{ff}$, $F_{bio}$, and $F_{biodis}$. A summary for each station can be found in Appendix A.

## 4.2 Impact of adding $\Delta^{14}CO_2$ observations

In this section, we present the results from the ZBASE and ZCO2Only experiments. We start by analyzing the retrieval of truth fossil $CO_2$ ($F_{ff}^t$) and biosphere ($F_{bio}^t$) time series. We divide the results into the regions shown in Figure 1, where Northern Europe represents Scandinavia, Finland, and the Baltic States, Western/Central Europe represents Benelux, France, Germany, Switzerland, Liechtenstein, and Austria, Southern Europe represents the Iberian Peninsula, Italy, and the Balkans (except for Romania and Bulgaria), Eastern Europe represents Poland, Slovakia, Hungary Romania, and Bulgaria, and the British Isles

represents Ireland and the United Kingdom. The study domain includes all the land shown in Figure 1, even the countries not mentioned in the definition of the regions (countries in gray in Figure 1).

### 4.2.1 Retrieval of the monthly and regional time series

In general, there is a closer agreement between the posterior and the truth for the biosphere fluxes ($F_{bio}$) than for the fossil $CO_2$ emissions ($F_{ff}$) in both the ZBASE and ZCO2Only experiments. This means that the model performs better at recovering

$F_{bio}$ from the observations compared to $F_{ff}$, as shown in Figure 6 for $F_{bio}$ and Figure 5 for $F_{ff}$. In the study domain, the inclusion of $\Delta^{14}CO_2$ observations in the ZBASE experiment yields better performance than ZCO2Only for both flux categories. Specifically, the posterior $F_{ff}$ ZBASE exhibits closer alignment to the truth than ZCO2Only with a lower RMSE (see Table 4), indicating a better fit of the seasonality for $F_{ff}$. Similarly, the posterior biosphere fluxes more closely follow the true time series than the fossil $CO_2$ emissions in both experiments, with ZBASE outperforming ZCO2Only in terms of RMSE and BIAS

values.

The regional analysis reflects the influence of the coverage by sampling stations on the inversion outcomes. Western/Central Europe, benefiting from the highest number of stations (18 out of 33 stations considered in this study, 10 of them measuring both tracers), shows the best alignment between the posterior and true time series for $F_{ff}$, especially in the ZBASE experiment (Figure 5b), while ZCO2Only shows pronounced RMSE and BIAS values (Table 4). Conversely, regions like Eastern Europe

(one station measuring both tracers) and the British Isles (two stations measuring only $CO_2$), despite their lower station coverage, exhibit posterior ZBASE $F_{ff}$ time series that closely approximate the truth, with Eastern Europe showing consistent performance throughout the year (panels d and f in Figure 5). However, the posterior ZBASE biosphere fluxes in these regions do not align as closely with the true values as observed in e.g. Western/Central Europe (panels d and f in Figure 6). In Eastern Europe, the posterior ZBASE shows big differences with the truth during May, June (maximum difference of 0.42 TgC day$^{-1}$),

and later in September, while ZCO2Only shows a better fit during these months but a positive bias the rest of the year (Figure 6d). In contrast, the posterior biosphere flux from the ZCO2Only experiment shows a better fit to the truth than the ZBASE one in the British Isles (Table 4).

Lastly, Southern and Northern Europe show similar results despite their differences: Northern Europe has better coverage of sampling stations, and its annual truth fossil $CO_2$ emissions are lower (an average of 0.20 TgC day$^{-1}$ against 0.59 TgC day$^{-1}$).

**Table 4.** RMSE and BIAS values for $F_{ff}$ and $F_{bio}$ from the ZBASE and ZCO2Only experiments in all the regions.

| Region | Fossil fuel ($F_{ff}$) | | | | | | Biosphere ($F_{bio}$) | | | | | |
|---|---|---|---|---|---|---|---|---|---|---|---|---|
| | RMSE (TgC day^-1) | | | BIAS | | | RMSE (TgC day^-1) | | | BIAS | | |
| | Prior | ZBASE | ZCO2Only | Prior | ZBASE | ZCO2Only | Prior | ZBASE | ZCO2Only | Prior | ZBASE | ZCO2Only |
| Study Domain | 4.07 | 1.51 | 2.75 | -4.03 | -1.51 | -2.74 | 4.66 | 1.12 | 2.12 | 1.18 | 0.74 | 1.90 |
| Western/Central Europe | 1.26 | 0.12 | 0.53 | -1.25 | -0.06 | -0.52 | 0.97 | 0.17 | 0.46 | 0.15 | -0.04 | 0.43 |
| Southern Europe | 0.60 | 0.42 | 0.51 | -0.59 | -0.41 | -0.50 | 0.89 | 0.35 | 0.41 | 0.35 | 0.29 | 0.35 |
| Eastern Europe | 0.55 | 0.07 | 0.33 | -0.54 | -0.02 | -0.33 | 0.61 | 0.22 | 0.34 | 0.15 | -0.04 | 0.26 |
| Northern Europe | 0.20 | 0.19 | 0.20 | -0.20 | -0.19 | -0.20 | 0.76 | 0.21 | 0.25 | 0.00 | 0.16 | 0.22 |
| British Isles | 0.28 | 0.14 | 0.15 | -0.28 | 0.07 | -0.21 | 0.30 | 0.16 | 0.09 | 0.02 | -0.13 | -0.02 |

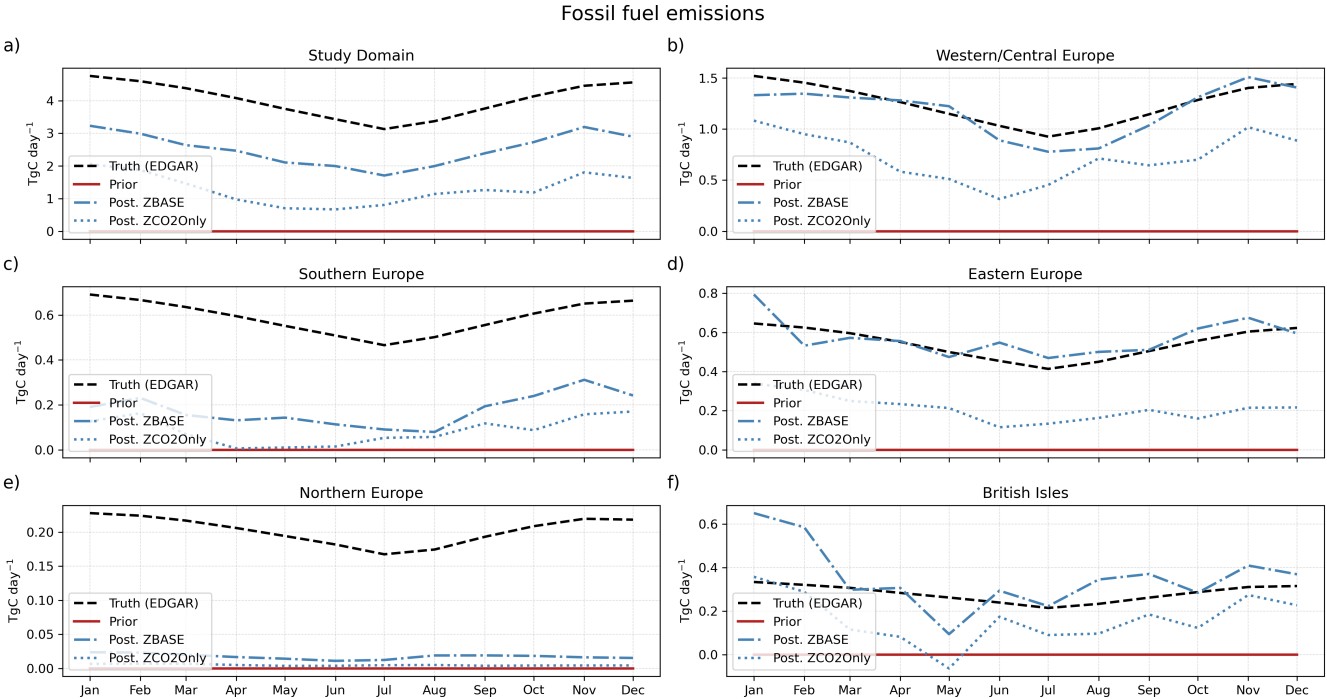

**Figure 5.** Monthly fossil $CO_2$ truth (dashed lines), prior (solid lines), and posterior fluxes from the ZBASE (dashed-dotted lines) and ZCO2Only (dotted lines) experiments for a) the study domain and the 5 sub-regions defined: b) Western/Central Europe, c) Southern Europe, d) Eastern Europe, e) Northern Europe, and f) British Isles.

In both regions, the posterior $F_{ff}$ of the two experiments is far from the truth (Figures 5c and 5e), while the posterior $F_{bio}$ of both regions and experiments is close to each other, with Northern Europe showing a better fit to the truth than Southern Europe, in which the posterior shows a more pronounced positive bias along the year (Figures 6c and 6e).

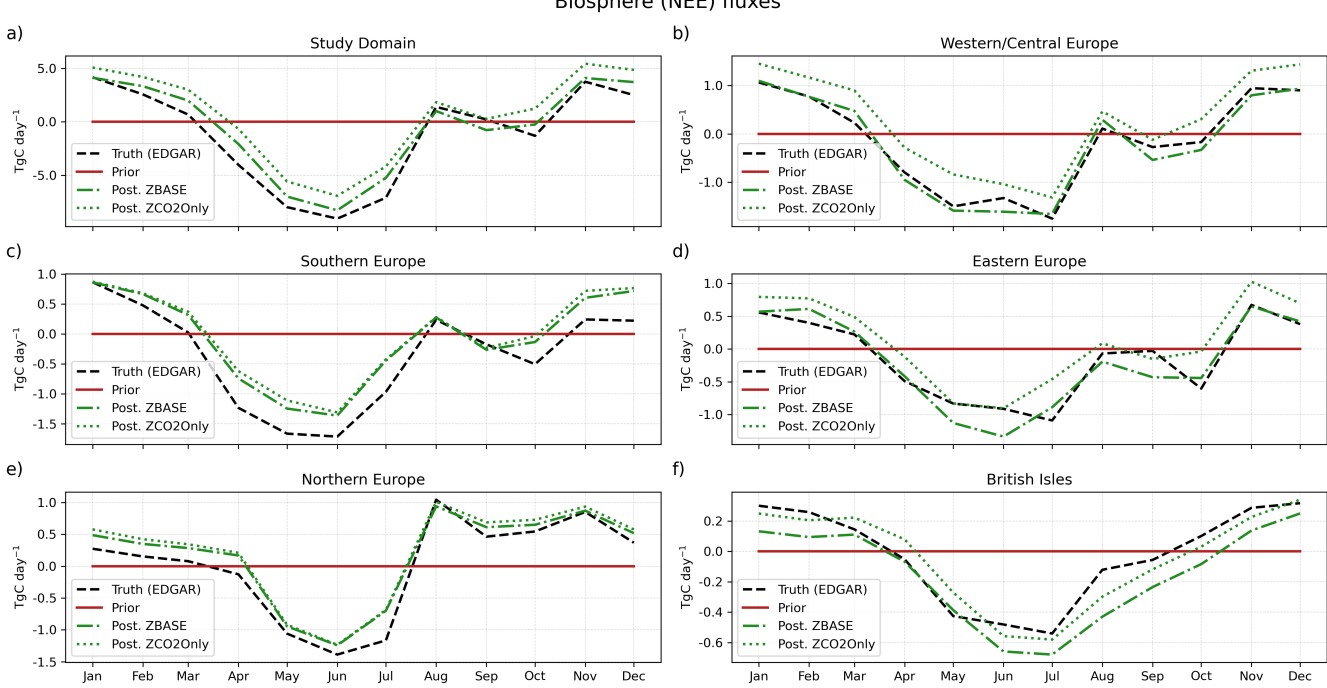

**Figure 6.** Monthly biosphere (NEE) truth (dashed lines), prior (solid lines), and posterior fluxes from the ZBASE (dashed-dotted lines) and ZCO2Only (dotted lines) experiments for a) the study domain and the 5 sub-regions defined: b) Western/Central Europe, c) Southern Europe, d) Eastern Europe, e) Northern Europe, and f) British Isles.

### 4.2.2 Analysis of the spatial error reduction

We set up the ZBASE and ZCO2Only experiments with prior uncertainties and error structures as in Table 2 based on the values of ODIAC for $F_{\text{ff}}$ and VPRM for $F_{\text{bio}}$. Therefore, the model had some information about the spatial and temporal error structure of the prior fluxes. To evaluate the spatial performance of LUMIA, we first aggregate the hourly values (both truth and posterior) to the optimization interval of one week. After this, we calculate the posterior RMSE of each experiment and flux category at the grid cell level, and finally, we calculate the RMSE reduction by subtracting the posterior RMSE of ZBASE from the posterior RMSE of ZCO2Only as follows:

$$RMSE_{\text{reduction}} = RMSE^{apos}_{\text{ZCO2Only}} - RMSE^{apos}_{\text{ZBASE}} \tag{9}$$

Here, positive values of $RMSE_{\text{reduction}}$ indicate posterior $RMSE^{apos}_{\text{ZBASE}}$ values that are lower than $RMSE^{apos}_{\text{ZCO2Only}}$, i.e. grid cells where when adding $\Delta^{14}$C observations (ZBASE) shows values closer to the truth (better performance, lower RMSE) than when only having $CO_2$ observations (ZCO2Only). For fossil fuel, we find larger prior RMSE values in Western/Central

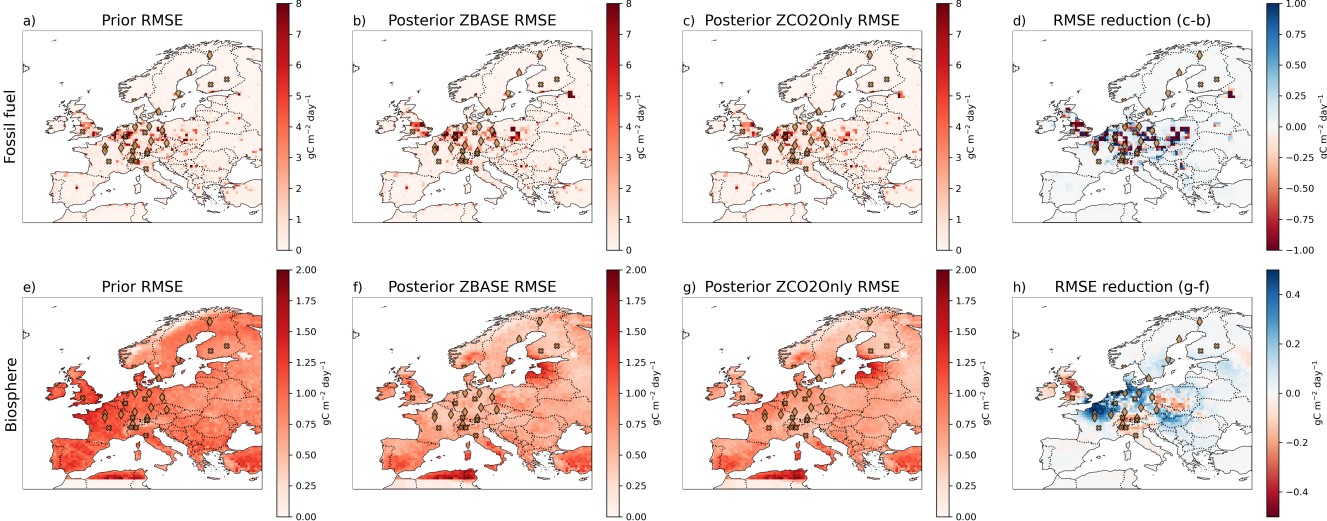

**Figure 7.** Spatial error of fossil $CO_2$ (a to d) and biosphere (e to h) for the ZBASE and ZCO2Only experiments. a) and e) show the prior RMSE for $F_{\mathrm{ff}}$ and $F_{\mathrm{bio}}$, respectively, b) and f) show the posterior RMSE for ZBASE, c) and g) show the posterior RMSE for ZCO2Only, and d) and h) show the RMSE reduction (see Equation 9) for fossil and biosphere. In Figures d) and h), positive values (in blue) show the pixels where ZBASE performs better than ZCO2Only (i.e. adding $\Delta^{14}CO_2$ observations improves the posterior estimates), and negative values (in red) where ZCO2Only performs better than ZBASE. Crosses and diamonds represent stations that only measure $CO_2$ and those that additionally measure $\Delta^{14}CO_2$, respectively.

Europe, but as well some grid cells show the location of larger cities like in southern England, Poland, and Spain (Figure 7a).
For the biosphere fluxes, we find the larger prior RMSE values in Western/Central Europe and the British Isles (Figures 7e).

     The largest positive RMSE reductions (where ZBASE performs better than ZCO2Only) (Figures 7d and 7h) occur around the sampling stations in Western/Central Europe and the British Isles for both flux categories. For fossil $CO_2$, most of the study domain has positive values (92%), although a large part of these values (around 75%) is close to zero, representing the values in Southern and Northern Europe where there is a low adjustment of the fluxes when adding $\Delta^{14}C$ observations (Figure 7d).
For the biosphere fluxes, the posterior RMSE maps (Figures 7f and 7g) show the regions that are poorly constrained due to the absence of observations such as the southern part of the domain and the Baltic States. Despite a lower portion of the study domain (40%) (Figure 7h) is showing an improvement in the posterior estimation when adding $\Delta^{14}C$ observations compared with fossil fuel, this presents a clearer pattern in areas such as southeast England, the northern part of Western/Central Europe, Denmark, and southern Sweden, as well as some areas in Eastern Europe.

**4.2.3   Recovery of the annual budget**

Next, we assess how accurately the model can estimate the annual budget for fossil fuel, biosphere (NEE), and the total $CO_2$. Figure 8 shows the annual budget of the study domain, the sub-regions (right), and some of the largest European countries by

area (left). We include the ODIAC emission data product and the VPRM product for the biosphere in Figure 8 as references since we base the prior uncertainty and error structure on the spatial and temporal distribution of these two products. As we find in the temporal distribution (Figure 5), in the study domain, the posterior fossil $CO_2$ from both experiments does not fit the truth, but ZBASE shows a lower bias from the truth than ZCO2Only. This result is reflected in the annual budget, where ZBASE recovers 63% from $F_{ff}^t$ while ZCO2Only recovers only 32% (Figure 8a). Likewise, the posterior $F_{bio}$ of ZBASE that closely fits $F_{bio}^t$, recovers 38% of the biosphere budget (Figure 8b), while ZCO2Only, which shows a larger positive bias in the temporal distribution, returns a positive annual budget, contrary to the negative annual budget of the true biosphere fluxes. This behavior is repeated in most of the regions and countries shown in Figure 8, where ZCO2Only strongly underestimates the annual fossil $CO_2$ emissions, with the lowest estimates in Southern (15%) and Northern Europe (2%), the latter with a strong underestimation from ZBASE as well (9%), France (33%), and Spain ($\sim 0\%$), which has a similar situation as Northern Europe (5% recovery for ZBASE), and returns an annual biosphere budget that compensates for the total $CO_2$ budget which is close to ZBASE in most of the cases.

Western and Eastern Europe show the best posterior $F_{ff}$ ZBASE values, 95%, and 105% of the truth, respectively. However, while some countries in these regions with good sampling coverage, such as the Benelux Union, show good recovery of $F_{ff}^t$ (96%), some others with fewer neighboring sampling stations, such as France and Poland, show results far from the annual fossil $CO_2$ emissions: 71% and 166%, respectively. Germany, which has the best coverage in the study domain, shows some overestimation (111%). On the other hand, the biosphere annual budget compensates in most cases for the total $CO_2$ budget, returning values that over and underestimate the truth, where the only regions with closer values are Western/Central Europe (126%) and Eastern Europe (128%) for the ZBASE experiment (Figure 8c). Finally, we find better estimates of the total $CO_2$ budget in most cases for the ZBASE experiment, with the largest recovery in Western/Central Europe (91%), Eastern Europe (96%), and Northern Europe (89%) (Figure 8e), and in the country level in Germany (99%) and France (94%) (Figure 8f).

## 4.3 A realistic setup

The most realistic approach we can take to perform OSSEs is to use a realistic set of prior fluxes that differ substantially from the true fluxes used to generate the synthetic observations. In this section, we perform a series of experiments using the prior $F_{ff}$, $F_{bio}$, and $F_{biodis}$ fluxes described in Section 3.1 to evaluate the impact of prescribing different prior fossil $CO_2$ uncertainty values as well as the impact of the prior $F_{biodis}$ flux product (RndBASU) in the optimization of $F_{ff}$.

### 4.3.1 Impact of the prior fossil $CO_2$ uncertainty

Figure 9 shows the weekly $F_{ff}$ time series for the three experiments using different prior uncertainties (BASE0.1, BASE0.21, and BASE0.3). The EDGAR ($F_{ff}^t$) and ODIAC (prior) products have different temporal distributions along the year, with ODIAC being flatter than EDGAR, but both with a minimum during summer, for EDGAR in July (3.13 TgC day$^{-1}$), and for ODIAC in August (3.05 TgC day$^{-1}$). In the study domain (Figure 9a), the posterior $F_{ff}$ for the three experiments is very close to each other and approaches the truth from January to February and later from August to December. From May to August, there is an increment in the posterior fluxes that depart from $F_{ff}^t$ with the maximum difference in July that we find in

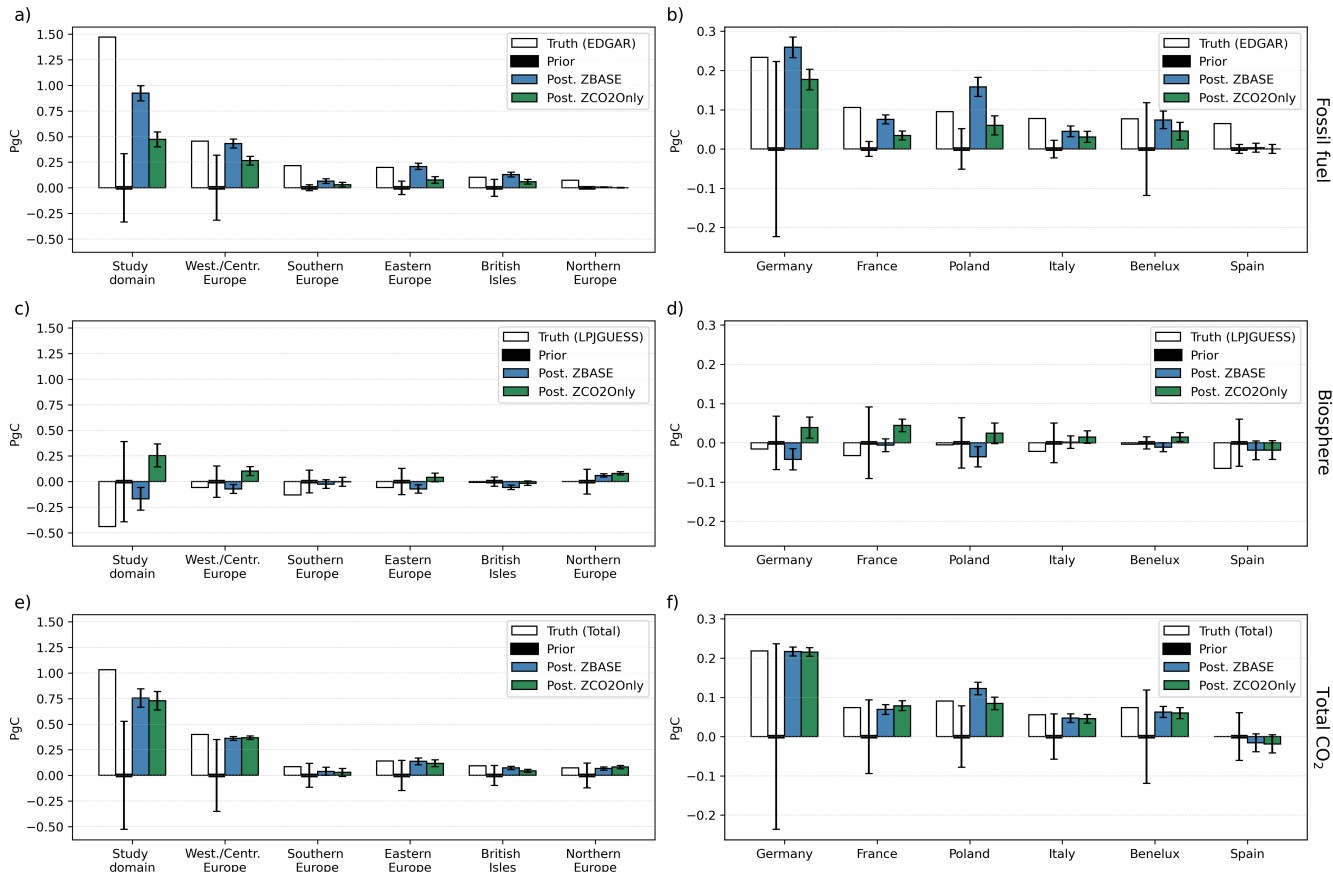

**Figure 8.** True, prior and posterior annual budgets of fossil (a-b), biosphere (c-d) and total $CO_2$ (e-f) for the study domain, the sub-regions (right), and some of the largest European countries by area (left). The white bars show the true annual budgets based on EDGAR and LPJ-GUESS flux products. The black bars represent the prior value, 0 PgC. The blue and green bars show the posterior budgets of ZBASE and ZCO2Only, respectively. The error bars represent the prior and posterior uncertainty calculated with a Monte Carlo ensemble of 100 members.

Western/Central Europe (ranging from 0.10 for BASE0.1 to 0.17 TgC day$^{-1}$ for BASE0.3) and in Eastern Europe (0.08 to 0.26 TgC day$^{-1}$) (Figure 9). The posterior time series from the three experiments have the same RMSE with respect to the truth, 0.48 TgC day$^{-1}$, which is lower than the prior RMSE of 0.65 TgC day$^{-1}$. The posterior $F_{ff}$ time series in Western/Central Europe shows the best results, with the estimates being close to truth, except for June and July. The three experiments show
the same performance, reducing the RMSE by $50\%$ (RMSE$_{prior}$ = 0.26 TgC day$^{-1}$, RMSE$_{BASE0.1}$ = 0.13 TgC day$^{-1}$), but BASE0.21 and BASE0.3 show the values farther from the truth in June and July. Northern Europe (Figure 9c), on the other hand, shows priors that are already very close to the truth, with a posterior RMSE equal to the truth of 0.07 TgC day$^{-1}$. Finally, in Eastern Europe, with the lowest sampling coverage, the three posterior time series degrade the prior estimate.

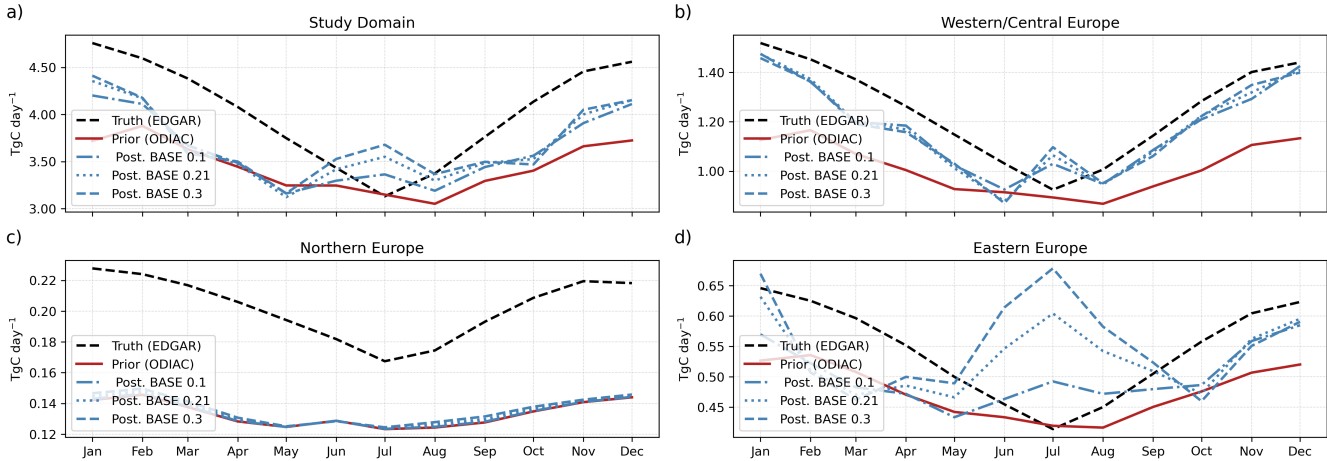

**Figure 9.** Monthly fossil $CO_2$ truth (black dashed lines), prior (red solid lines), and posterior fluxes from the BASE0.1 (blue dashed-dotted lines), BASE0.21 (blue dotted lines), and BASE0.3 (blue dashed lines) experiments for a) the study domain and the 3 sub-regions: b) Western/Central Europe, c) Northern Europe, and d) Eastern Europe (note the different scales on the y-axis).

The difference in the annual budget of EDGAR and ODIAC for the study domain is 0.21 PgC for the year 2018, which is as large as the emission of the country with the largest emission in the study domain for the same year, Germany, with 0.23 PgC according to EDGAR, and 0.19 PgC according to ODIAC (Figure 10). This difference in the study domain is nearly recovered by all three experiments, with a recovery ranging from $30\%$ for BASE0.1 to $45\%$ for BASE0.3. In Western/Central Europe, the three experiments recover $96\%$ of the truth (around $71\%$ of the difference between true and prior), similar to Germany, where the recovery ranges from $94\%$ for BASE0.1 to $97\%$ for BASE0.3 ($68\%$ to $82\%$ of the difference). As we find in the time series (Figure 9d), the prior annual budget is very close to the truth both in Eastern Europe, where the difference is 0.02 PgC, and in Poland, 0.01 PgC. In both cases, the posterior recovers the annual budget, with overestimation from BASE0.3 for the whole sub-region and from BASE0.21 and BASE0.3 in Poland, which are as big as $120\%$. Finally, and as expected from Figure 9c and the prior uncertainty for the sub-region, there is no recovery of the annual budget in Northern Europe further than the prior estimate.

### 4.3.2 Impact of the terrestrial isotopic disequilibrium product

The prior $F_{\text{bio}}$ and $F_{\text{biodis}}$ are very different in magnitude from the true values, with differences as large as 13.4 TgC day$^{-1}$ and 7.6 TgC day$^{-1}$, respectively, during summer for the whole study domain (Figures 11d and 11g). This gap is well resolved for $F_{\text{bio}}$ in the study domain and Western/Central Europe (Figures 11d and 11f), and with some underestimation in Eastern Europe between June and September (Figure 11e). However, for the posterior $F_{\text{biodis}}$ we find some larger differences from the truth during June and September in the study domain and the two sub-regions. When we prescribe $F_{\text{biodis}}$ (BASENoBD), the posterior $F_{\text{ff}}$ values from June to August in the study domain and Western/Central Europe (Figures 11a and 11b) get closer

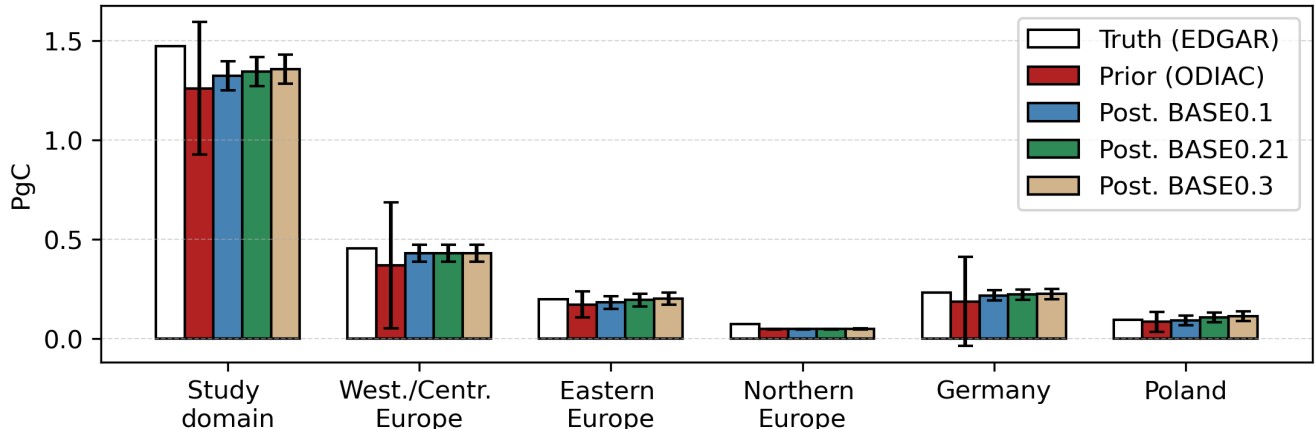

**Figure 10.** Total annual fossil $CO_2$ emissions for the study domain, Western/Central Europe, Eastern Europe, Northern Europe, Germany, and Poland. The white bars show the true emissions based on the EDGAR emission database. The red bars show the prior fluxes based on the ODIAC emission data product. The blue, green and tan bars show the posterior fossil $CO_2$ emissions for the BASE0.1, BASE0.21, and BASE0.3 experiments, respectively. The error bars represent the prior and posterior uncertainty calculated with a Monte Carlo ensemble of 100 members.

to $\boldsymbol{F}_{\mathrm{ff}}^{\mathrm{t}}$, and after the summer in the study domain. This can also be seen in an improvement in the RMSE values with 0.32 TgC day$^{-1}$ for the study domain and 0.10 TgC day$^{-1}$ for Western/Central Europe. In Eastern Europe, the posterior $\boldsymbol{F}_{\mathrm{ff}}$ for BASENoBD experiments does not show a significant improvement and, on the contrary, further degrades the prior estimate during the summer and the autumn.

### 4.3.3 The observational space

Finally, we analyze the model's performance in the observational space at all sampling stations aggregated together, one polluted station (Saclay, SAC) (Figure 12) and one background station (Jungfraujoch, JFJ) (Figures 12 and 13) for the BASE experiment. We calculate two performance metrics: the correlation coefficient (R) between the synthetic observations and the prior and posterior simulated mixing ratios for all the sites and individually for the two sites selected, and the reduced chi-square $\chi_\nu^2$ for the overall simulation as measure to of the improvement upon the initial state and to guarantee that we are not under or over-fitting the model (Table 5). We calculate the reduced chi-square as:

$$\chi_\nu^2 = \frac{1}{\nu} \sum_{i=1}^{N} \left( \frac{\boldsymbol{y}_i^{\mathrm{so}} - \boldsymbol{y}_i^{\mathrm{b,a}}}{\epsilon_i} \right)^2 \tag{10}$$

Where $\boldsymbol{y}_i^{\mathrm{so}}$ is the synthetic observation $i$, $\boldsymbol{y}_i^{\mathrm{b,a}}$ is either the prior (b) or the posterior (a) mixing ratio $i$, $\epsilon_i$ is the error of the synthetic observation $i$, $N$ is the number of observations, and $\nu$ are the degrees of freedom calculated as $\nu = N - p$, being $p$

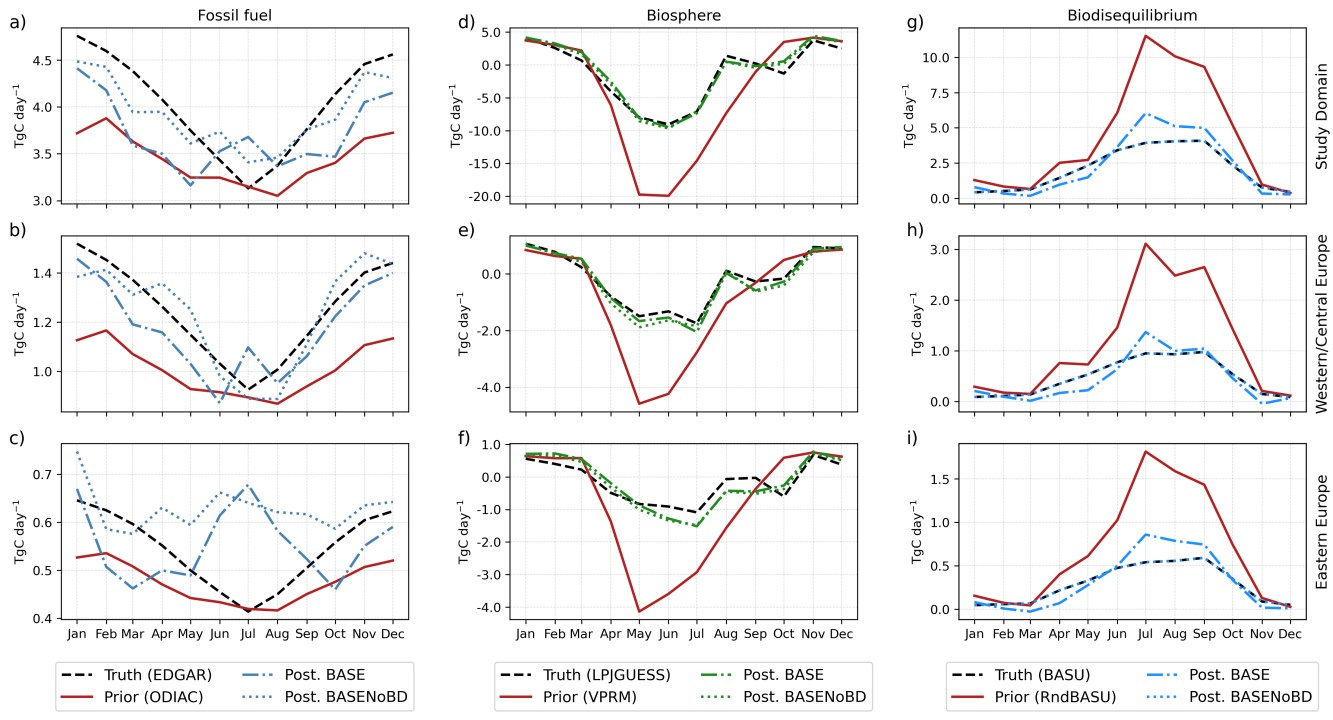

**Figure 11.** Monthly time series of $F_{ff}$ (a) to c)), $F_{bio}$ (d) to f)), and $F_{biodis}$ (g) to i)), for the study domain (top panel), Western/Central Europe (center panel), and Eastern Europe (bottom panel). The truth is represented in black dashed lines, prior in red solid lines, posterior fluxes from the BASE0.1 in blue dashed-dotted lines, and BASENoBD in blue dotted lines.

**Table 5.** Performance metrics (correlation coefficient R, standard deviation and the reduced chi-square $\chi_\nu^2$) for all sites, Saclay (SAC), and Jungfraujoch (JFJ).

|  |  | Prior | | Posterior | |
|---|---|---|---|---|---|
|  |  | R | $\sigma$ | R | $\sigma$ |
| All sites | $CO_2$ | 0.64 | 14.2 | 0.68 | 13.4 |
|  | $\Delta^{14}CO_2$ | 0.72 | 6.4 | 0.99 | 1.2 |
| SAC | $CO_2$ | 0.56 | 31.9 | 0.59 | 31.1 |
|  | $\Delta^{14}CO_2$ | 0.63 | 6.8 | 0.99 | 0.5 |
| JFJ | $CO_2$ | 0.65 | 5.5 | 0.74 | 4.5 |
|  | $\Delta^{14}CO_2$ | 0.75 | 4.2 | 0.84 | 1.5 |
| $\chi_\nu^2$ |  | 1.77 | | 1.06 | |

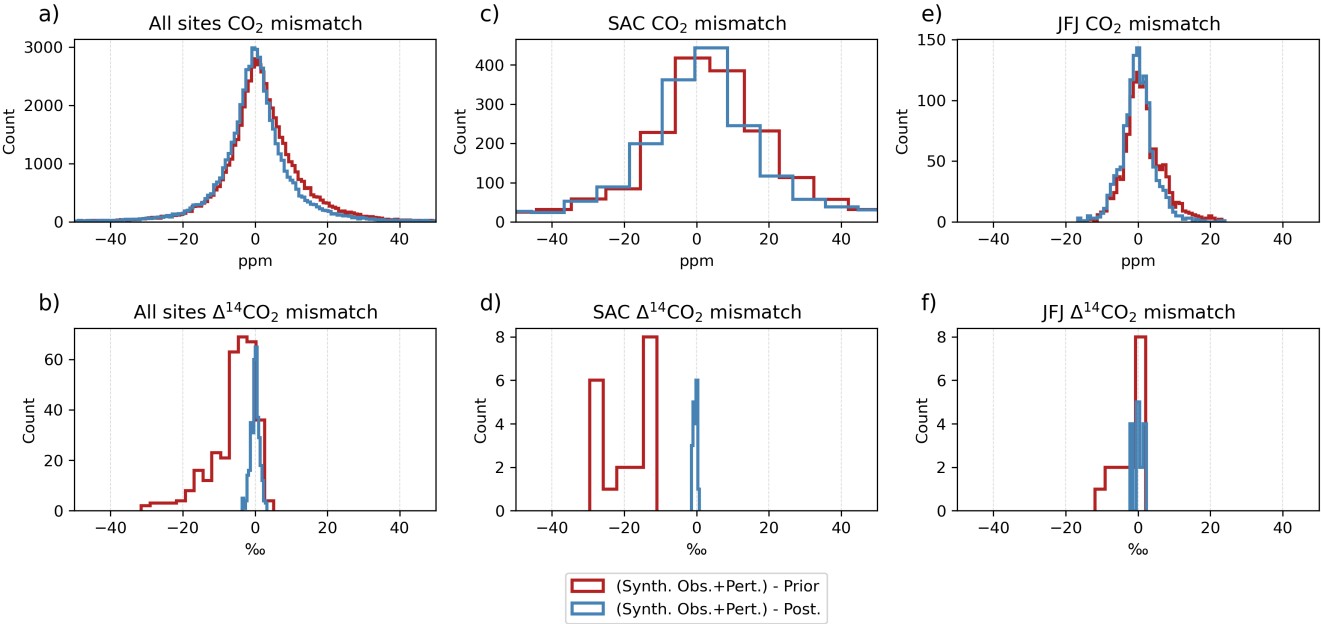

**Figure 12.** Mismatches between the synthetic observations and the prior (red) and posterior (blue) mixing ratios for all the sampling stations, Saclay (SAC) and Jungfraujoch (JFJ) for $CO_2$ (a, c and e) and for $\Delta^{14}CO_2$ (b, d and f). All prior and posterior mixing ratios correspond to the BASE experiment.

the number of fitted parameters in the model. Since $p$ is difficult to calculate due to the different time and space clusters, we keep the number of observations as the degrees of freedom ($\nu = N$).

The histograms in Figure 12 show the mismatches between the synthetic observations and the prior and posterior mixing ratios. For the $CO_2$ mixing ratios at all sites (Figure 12a), the histogram shows a distribution centered around zero for both prior
and posterior mismatches with a standard deviation of 14.2 and 13.4, respectively (see Table 5), indicating systematic deviations from the observed values. The posterior mismatch has a slightly tighter distribution, suggesting a small improvement in the model after adjustments as reflected in the correlation coefficient (Table 5). At Saclay (Figure 12c), the mismatch distribution is wider than the aggregate of all sites, which could suggest greater variability or larger errors at this particular site. The posterior adjustment has not significantly tightened the distribution, indicating that the model adjustments did not perform as
well at this site as they did on average across all sites. On the other hand, the distribution in Jungfraujoch (Figure 12e) is much tighter than in all sites and SAC, with the posterior mismatch displaying a slight improvement in precision as evidenced by the narrower spread. However, when comparing the posterior time series with the synthetic observations before adding the random perturbation (Figures 13a and 13c), there is a better agreement between them than with the prior values, especially during periods of higher variability (April to July at SAC, and April to September at JFJ).
The $\Delta^{14}CO_2$ synthetic observations are in general better fitted by the posterior than $CO_2$ at all sites, SAC and JFJ (Table 5). In all cases, the prior distribution is displaced to the negative values, indicating that the prior simulated values are in

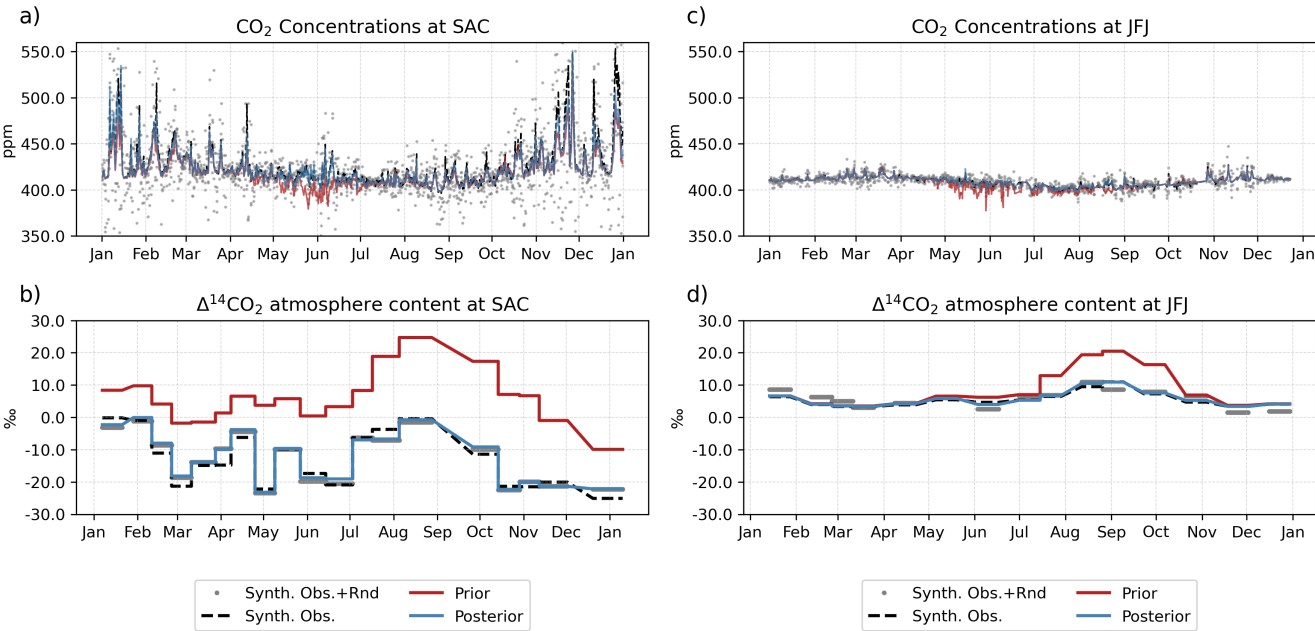

**Figure 13.** mixing ratio time series of $CO_2$ (a and c) and $\Delta^{14}CO_2$ (b and d) at Saclay (SAC) and Jungfraujoch (JFJ), respectively. All prior and posterior mixing ratios correspond to the BASE experiment.

general higher than the synthetic observations as shown for the whole period at SAC (Figure 12d) and from July to November at JFJ (Figure 12f). These larger prior mixing ratios are mainly caused by the prior terrestrial disequilibrium flux from July to November, and by the nuclear production fluxes throughout the year, which is significantly larger at Saclay (Figure 14).
However, the posterior mismatches showed a much narrower spread around zero at all sites (Figure 12b), Saclay (Figure 12d), and Jungfraujoch (Figure 12f) that is evident in the time series at both sites where the posterior closely follows the synthetic observations, and supported by the correlation coefficients (Table 5).

The reported $\chi_\nu^2$ values of 1.77 for the prior and 1.06 for the posterior across all sites and samples suggest a substantial improvement in the model's performance in adjusting the prior mixing ratios to the synthetic observations. A $\chi_\nu^2$ of 1.77 for
the prior indicates that there were significant discrepancies between the prior and the synthetic observations. This is consistent with the broader spread of mismatches in the histograms for both SAC and JFJ sites, as well as the apparent overestimation of $\Delta^{14}CO_2$ content in the time series. The improvement to a $\chi_\nu^2$ of 1.06 for the posterior indicates a better fit to the synthetic observations that are likely to be reflective of the underlying data patterns while still maintaining some degree of generalizability without overfitting the data.

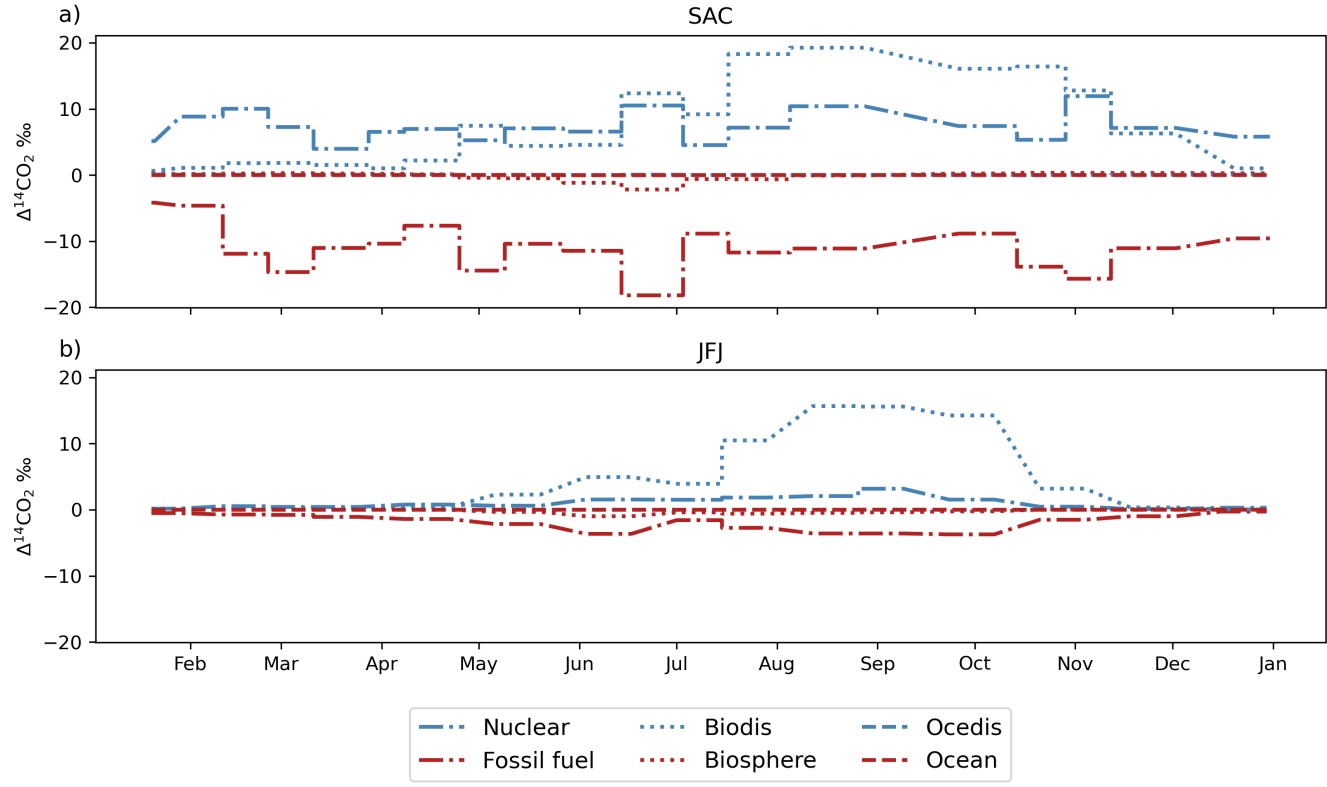

**Figure 14.** Contribution of each category to the prior $\Delta^{14}CO_2$ simulated mixing ratios at Saclay (a) and Jungfraujoch (b).

## 5  Discussion

Under the current sampling strategy and observation network, we demonstrate through OSSEs that adding $\Delta^{14}CO_2$ observations can help us constrain fossil $CO_2$ emissions over Europe using the LUMIA system. We start with two simulation experiments in which we set the prior fossil $CO_2$ and biosphere (Net Ecosystem Exchange, NEE) fluxes to zero: ZBASE and ZCO2Only. Under an OSSE setup, even when using completely different truth and prior flux products (e.g. different spatio-temporal distributions and annual budgets), due to assumptions such as a perfect transport model and background mixing ratios, it is easy for the model to retrieve the true values even without adding $\Delta^{14}CO_2$ observations. For this reason, we set up these two more challenging experiments to assess the capabilities of the inversion system to constrain the fossil $CO_2$ emissions and biosphere fluxes using $CO_2$ and $\Delta^{14}CO_2$. The ZBASE and ZCO2Only experiments show us that in regions with a dense sampling network, such as Western/Central Europe, when adding $\Delta^{14}CO_2$ observations, LUMIA is capable of recovering the seasonality of $F_{ff}$ and $F_{bio}$, as well as the total annual $CO_2$ budget of the whole region and some of the larger countries (also in terms of fossil $CO_2$ emissions) such as Germany and France. On the other hand, the results in Northern Europe, which has a relatively good network coverage, are not as good as in Western/Central Europe regarding fossil $CO_2$. Comparing the ranges

of the true fossil $CO_2$ and biosphere fluxes in Northern and Western/Central Europe, we find that, while $F_{bio}^t$ has a similar range in both regions, $F_{ff}^t$ differs by one order of magnitude. Using the concept of signal-to-noise ratio, if we consider the fossil $CO_2$ as the signal (the variable in which we are more interested) and the biosphere as the noise, this difference of one order of magnitude between them in Northern Europe makes it easier for the model to recover the biosphere fluxes than the fossil $CO_2$ emissions, even with additional information about $\Delta^{14}CO_2$. In addition, the prior uncertainty which is proportional to the fluxes, is close to zero for the fossil fluxes, while it is two orders of magnitude larger for the biosphere, making it for the inversion more costly to constrain the fossil $CO_2$ emissions.

The inversions are able to resolve the NEE both at the continental level and in the sub-regions but struggle more with fossil emissions in some regions with few observations (e.g. Southern Europe and the British Isles). This is similar to what was found by Wang et al. (2018) despite the differences between their inversion implementation and our LUMIA system. The main differences lie in the transport model and the inversion approach. They use a global transport model at a resolution of $3.75° \times 2.5°$ (Laboratoire de Météorologie Dynamique's LMDZv4) and a pre-calculated fossil $CO_2$ tracer (product of the mass balance), while we use a Lagrangian regional transport model at a higher horizontal resolution ($0.5° \times 0.5°$) and optimize both the fossil and the natural fluxes using as tracers $CO_2$ and $\Delta^{14}CO_2$. Wang et al. (2018) found the largest error reductions around Germany, Benelux, and eastern France, where most sampling stations are located. Northern Europe was also poorly constrained in their inversions, similar to what we find. Wang et al. (2018) attributed the results in Northern Europe to the coarse spatial resolution of the transport model. But even with a higher resolution transport model as employed in LUMIA, we still can not resolve the true fossil $CO_2$ emissions in an OSSE setup given the current $CO_2$ and $\Delta^{14}CO_2$ observation networks. We think that a more likely explanation is the difference in the magnitude of the fossil $CO_2$ emissions in this region against the natural fluxes. This can be seen by the differences in the seasonal amplitude of the fluxes. In Western/Central Europe $F_{bio}$ and $F_{ff}$ are of a similar order of magnitude (2.81 TgC day$^{-1}$ for $F_{bio}$ and 0.6 TgC day$^{-1}$ for $F_{ff}$) (see Figures 5 and 6). In contrast, in Northern Europe, there is a tenfold difference in the seasonal amplitude of the two fluxes: 2.44 TgC day$^{-1}$ for $F_{bio}$ and 0.06 TgC day$^{-1}$ for $F_{ff}$. In addition, the prior uncertainty for $F_{ff}$ (0.002 PgC year$^{-1}$) in this region is much lower compared to $F_{bio}$ (0.12 PgC year$^{-1}$) in Northern Europe.

The BASE experiments, in which we use realistic prior fluxes, show that the posterior fossil $CO_2$ emissions are not very sensitive to the prescribed prior uncertainty in regions with a dense sampling network, even when using a low prior $F_{ff}$ uncertainty in which case it is more difficult for the inversion algorithm to recover the true fluxes. As we have observed in previous studies using LUMIA (Monteil et al., 2020; Monteil and Scholze, 2021), the cost of fitting the observations dominates the total cost function value. In this sense, the relative value of the prior uncertainty of $F_{bio}$ against $F_{ff}$ is going to significantly impact the spatio-temporal distribution of flux adjustments, but the total uncertainty of the fluxes is of lesser importance since the model has enough freedom to adjust the data. In other words, the error structure and how is it set up for the different flux categories, is going to have more impact than the total prior uncertainty. Both Basu et al. (2016) and Wang et al. (2018) highlight the importance of a regional horizontal correlation and error structure for fossil $CO_2$ emissions. In our study, we use the same horizontal correlation and error structures developed by Monteil et al. (2020) originally for NEE. We are aware of the necessity of defining specific structures for fossil $CO_2$ within LUMIA due to the low improvement in spatial terms that we find in Figure

7 when adding $\Delta^{14}CO_2$ observations. However, it is important to mention that given the sparse observation network, we can expect spatial misattributions (flux corrections that should happen in one place but are instead made elsewhere), and therefore, we should interpret the results aggregated at the scale that is relevant given the model setup, as we demonstrate through the time series and annual budget results. Such spatial misattribution is illustrated in the spatial RMSE reduction results for the biosphere fluxes. We can clearly identify the formation of dipoles (clusters of larger RMSE values) in regions with no observations such as the southern part of the study domain and the Baltic States indicating that these areas are underconstrained.

We also find the prior terrestrial disequilibrium product to have an important impact on the posterior fossil $CO_2$ emissions (Figure 11). The prior terrestrial isotopic disequilibrium flux in our experiments is on purpose incorrect with the aim of showing the impact that it can have in the estimation of fossil $CO_2$ emissions. As shown in Figure 11, the maximum difference between the prior and the true $F_{biodis}$ is of the same order of magnitude for Western/Central Europe (2.1 TgC day$^{-1}$) and Eastern Europe (1.3 TgC day$^{-1}$) in July. For $F_{ff}$, however, the difference between the prior and truth is about one order of magnitude larger for Western/Central Europe compared to Eastern Europe (0.03 vs 0.005 TgC day$^{-1}$). This larger difference causes a stronger dilution of the fossil emissions in Eastern Europe, and therefore essentially lowers the signal-to-noise ratio of the $\Delta^{14}CO_2$ measurements, and added to the lower network coverage compared to Western/Central Europe, a poorer constrain of the fossil $CO_2$ emissions. As seen also in Figure 9, this is particularly evident in Eastern Europe during the summer months, where the fossil $CO_2$ signal is further convoluted by the large biospheric uptake, making it more difficult to accurately constrain fossil emissions in this region. According to Turnbull et al. (2009), one of the main contributors to atmospheric $\Delta^{14}CO_2$ is heterotrophic respiration in natural environments. Therefore, having a good prior $F_{biodis}$ estimate is crucial in estimating posterior $F_{ff}$. The impact of $F_{biodis}$ and the other $\Delta^{14}CO_2$ flux terms is not negligible, particularly, the emissions from nuclear facilities that can have a larger impact than the terrestrial disequilibrium (Graven and Gruber, 2011) as was evident when analyzing the individual impacts of the flux categories, showing that at sampling sites heavily influenced by emissions from nuclear facilities such as Saclay, these emissions can be as large as the terrestrial isotopic disequilibrium fluxes. In this study, we fixed the $F_{nuc}$ term (i.e. we use the same fluxes for calculating the synthetic observations and in the inversions), and hence, its impact is not considered here. In previous studies (Wang et al., 2018; Basu et al., 2016, 2020) the $F_{nuc}$ is usually prescribed and assumed as an annual value at each nuclear facility location (Graven and Gruber, 2011; Zazzeri et al., 2018) due to a lack of knowledge on the temporal distribution of these emissions. This variability in nuclear emissions has been only studied by measuring the atmospheric content of $\Delta^{14}CO_2$ in the surrounding areas of single nuclear facilities (Turnbull et al., 2014; Vogel et al., 2013; Lehmuskoski et al., 2021), but not yet in a large regional setup, and therefore it needs further investigation.

The Observing System Simulation Experiment (OSSE) framework used in this study assumes a perfect realization of atmospheric transport and mixing processes by employing the same transport model across the simulations. This assumption simplifies the complex nature of atmospheric dynamics and is a common approach to limit the scope of variability in such studies. However, it is crucial to acknowledge that this simplification overlooks one of the largest sources of uncertainty in atmospheric inverse modeling: the accurate representation of atmospheric transport and mixing processes. The variability and uncertainty in atmospheric transport can significantly impact the estimation of greenhouse gas sources and sinks. As demonstrated by Schuh et al. (2019), inconsistencies in transport simulations can introduce systematic biases in surface flux estimations, which can

be as substantial as 1.7 PgC year$^{-1}$ for large zonal bands. In a study by Munassar et al. (2023), in which multiple combinations of global and regional models were tested using two different inversion frameworks (LUMIA and CarboScope-Regional (CSR)), they found that using a different regional transport (FLEXPART and STILT (Stochastic Time-Inverted Lagrangian Transport)) model can cause differences in the posterior NEE annual budget of 0.51 PgC year$^{-1}$. This highlights the sensitivity of inversion-derived emission estimates to the accuracy of the transport model used and emphasizes the critical role that transport uncertainty plays across global flux inversion systems.

Furthermore, the assumption of perfect boundary conditions in the model presents another significant simplification. Boundary conditions in atmospheric modeling can greatly influence the mixing ratio gradients and flux estimates, and their mischaracterization can propagate errors throughout the model domain. Coming back to the study by Munassar et al. (2023), the use of a different global transport model (TM3 and TM5) for the estimation of the boundary condition can cause discrepancies in the posterior annual budget as large as 0.23 PgC year$^{-1}$. Errors in these aspects of the transport model could lead to skewed emission estimates. Given these considerations, the presented results should be interpreted with caution, understanding that the true uncertainty in atmospheric inverse modeling is likely understated in these OSSEs. It underscores the need for more comprehensive approaches that account for transport model uncertainties, such as employing ensemble modeling techniques that incorporate multiple transport models and boundary conditions to better capture the inherent uncertainties in atmospheric dynamics (Locatelli et al., 2015; Aleksankina et al., 2018).

## 6 Conclusions and future perspectives

We have expanded the LUMIA system to be capable of simultaneously inverting atmospheric observations of $CO_2$ and $\Delta^{14}CO_2$ to estimate fossil $CO_2$ emissions and net terrestrial biosphere $CO_2$ fluxes over Europe. We performed the first observing system simulation experiments to test the performance of the $\Delta^{14}$C-enhanced LUMIA version. In the first set of experiments, we show the impact of adding $\Delta^{14}$C observations in a scenario with prior estimates of $\boldsymbol{F}_{ff}$ and $\boldsymbol{F}_{bio}$ set to zero. In regions with good sampling network coverage, assimilating both $CO_2$ and $\Delta^{14}$C observations allows recovering the seasonality of $\boldsymbol{F}_{ff}$ and $\boldsymbol{F}_{bio}$ and the annual $\boldsymbol{F}_{ff}$ budget, while when assimilating only $CO_2$ observations, the posterior $\boldsymbol{F}_{ff}$ is degraded. In the second set of experiments, we performed OSSEs using more realistic priors to test the impact of the prescribed $\boldsymbol{F}_{ff}$ uncertainty and the impact of the prior $\boldsymbol{F}_{biodis}$ product. The prescribed prior uncertainty has no significant impact on the posterior $\boldsymbol{F}_{ff}$. On the other hand, the prior $\boldsymbol{F}_{biodis}$ product can significantly impact the posterior $\boldsymbol{F}_{ff}$.

The purpose of this study is to describe the multi-tracer, more specifically $CO_2$ and $\Delta^{14}CO_2$, version of LUMIA and illustrate its application to estimate both fossil $CO_2$ emissions natural $CO_2$ fluxes simultaneously. Future work should analyze in more detail the impact of various aspects of our inversion set-up here, such as the assumption of a perfect transport model, the specification of the boundary conditions as well as different spatiotemporal error structures, on the posterior fossil $CO_2$ emissions and natural $CO_2$ fluxes. Particular emphasis should be placed on the analysis of the impact of the prior $\boldsymbol{F}_{biodis}$ product using simulated terrestrial biosphere disequilibrium estimates by, e.g. the LPJ model following the methodology by Scholze et al. (2003) because our study here illustrated the importance of this flux term in the $CO_2$ and $\Delta^{14}CO_2$ inversion. In addition,

the impact of the prior $F_{nuc}$, the sampling strategy, and the network density of the $\Delta^{14}C$ observations on the capability to estimate fossil $CO_2$ emissions needs to be evaluated. The current 2-weekly integrated sampling strategy allows us to get a reasonable estimate of the annual budget over the whole domain. But the inversion can neither recover the correct temporal behavior nor the spatial distribution of the fossil $CO_2$ emissions when using $C\Delta^{14}C$ observations provided by the current 2-weekly integrated sampling strategy. Additionally, converting $\Delta^{14}CO_2$ values to $C\Delta^{14}C$ implies calculating the average of the $CO_2$ observations during the 2-week integration period that can introduce additional errors that we did not account for in this study. We will evaluate the use of hourly flask samples under different strategies as described by Levin et al. (2020), such as a "smart" sampling based on pollution episodes of $CO_2$ and CO. This will be in preparation for the intensive $\Delta^{14}CO_2$ sampling campaign (hourly samples taken every third day) planned within the EC's Horizon Europe CORSO (CO2MVS Research on Supplementary Observations) project (https://corso-project.eu/) during 2024 at 10 ICOS stations located in Western Europe.

*Code availability.* The LUMIA source code used in this paper has been published on Zenodo and can be accessed at https://doi.org/10.5281/zenodo.8426217.

*Data availability.* The revised EDGARv4.3 https://doi.org/10.18160/GFNT-5Y47, LPJ-GUESS https://doi.org/10.18160/p52c-1qjm, and VPRM https://doi.org/10.18160/VX78-HVA1 datasets are availabale from the ICOS-Carbon Portal. ODIAC data is available at https://doi.org/10.17595/20170411.001. The input data has been uploaded on Figshare and is available at https://doi.org/10.6084/m9.figshare.24307162.

# Appendix A: Summary of ocean and ocean disequilibrium-derived synthetic observations

**Table A1.**

| Station | Ocean $CO_2$ ppm | Ocean $\Delta^{14}C$ ‰ | Ocedis $\Delta^{14}C$ ‰ | Synth. Obs. $CO_2$ ppm | Synth. Obs. (rnd) $CO_2$ ppm | Obs. Error $CO_2$ ppm | Synth. Obs. $\Delta^{14}C$ ‰ | Synth. Obs. (rnd) $\Delta^{14}C$ ‰ | Obs. Error $\Delta^{14}C$ ‰ |
|---|---|---|---|---|---|---|---|---|---|
| All sites | -0.07 ± 0.15 | -0.007 ± 0.007 | 0.02 ± 0.017 | 414.6 ± 12.7 | 414.6 ± 18.3 | 9.8 ± 9.0 | 0.3 ± 8.0 | 0.2 ± 8.4 | 1.9 ± 0.05 |
| GAT | -0.07 ± 0.1 | -0.008 ± 0.007 | 0.021 ± 0.014 | 415.7 ± 12.7 | 416.2 ± 19.2 | 11.1 ± 7.2 | 2.1 ± 4.2 | 2.6 ± 4.8 | 1.9 ± 0.05 |
| HPB | -0.04 ± 0.05 | -0.005 ± 0.003 | 0.016 ± 0.008 | 414.0 ± 11.2 | 414.5 ± 16.7 | 10.4 ± 6.3 | 1.0 ± 6.3 | 1.5 ± 6.8 | 1.9 ± 0.04 |
| HTM | -0.07 ± 0.12 | -0.009 ± 0.009 | 0.016 ± 0.009 | 415.4 ± 12.3 | 415.5 ± 16.8 | 10.0 ± 5.7 | 1.0 ± 4.5 | 0.5 ± 4.8 | 1.9 ± 0.04 |
| JFJ | -0.03 ± 0.04 | -0.002 ± 0.002 | 0.01 ± 0.005 | 409.1 ± 5.0 | 409.0 ± 6.9 | 4.2 ± 2.1 | 5.5 ± 2.3 | 5.5 ± 2.9 | 2.0 ± 0.02 |
| KIT | -0.06 ± 0.06 | -0.005 ± 0.004 | 0.024 ± 0.012 | 427.1 ± 16.9 | 427.4 ± 26.9 | 17.7 ± 10.0 | -5.2 ± 10.5 | -4.7 ± 11.0 | 1.9 ± 0.05 |
| KRE | -0.05 ± 0.06 | -0.005 ± 0.004 | 0.014 ± 0.009 | 415.3 ± 12.6 | 415.0 ± 16.9 | 10.3 ± 6.0 | -4.0 ± 4.6 | -4.2 ± 4.9 | 1.9 ± 0.05 |
| LIN | -0.06 ± 0.09 | -0.007 ± 0.006 | 0.017 ± 0.011 | 420.9 ± 16.9 | 420.2 ± 25.2 | 15.2 ± 11.6 | -7.7 ± 9.4 | -8.2 ± 9.5 | 1.9 ± 0.05 |
| NOR | -0.07 ± 0.14 | -0.009 ± 0.009 | 0.011 ± 0.01 | 415.8 ± 10.7 | 415.5 ± 14.4 | 8.5 ± 4.8 | 4.9 ± 4.3 | 4.5 ± 5.1 | 1.9 ± 0.03 |
| OPE | -0.07 ± 0.08 | -0.006 ± 0.004 | 0.034 ± 0.021 | 416.7 ± 14.3 | 416.5 ± 21.5 | 13.3 ± 9.3 | -1.6 ± 6.8 | -1.2 ± 6.3 | 1.9 ± 0.04 |
| OXK | -0.06 ± 0.08 | -0.006 ± 0.004 | 0.02 ± 0.013 | 411.0 ± 7.3 | 410.8 ± 10.5 | 7.1 ± 3.0 | 1.8 ± 4.8 | 1.5 ± 5.8 | 1.9 ± 0.03 |
| PAL | -0.1 ± 0.13 | -0.011 ± 0.007 | 0.005 ± 0.004 | 412.3 ± 8.6 | 412.3 ± 10.8 | 6.0 ± 3.7 | 8.7 ± 4.2 | 8.9 ± 5.0 | 1.9 ± 0.03 |
| SAC | -0.08 ± 0.1 | -0.009 ± 0.007 | 0.04 ± 0.02 | 425.2 ± 23.0 | 425.6 ± 37.9 | 23.1 ± 20.0 | -13.1 ± 8.3 | -13.7 ± 8.8 | 1.9 ± 0.03 |
| STE | -0.08 ± 0.12 | -0.01 ± 0.007 | 0.021 ± 0.01 | 413.4 ± 10.0 | 413.7 ± 15.6 | 9.4 ± 7.2 | 0.4 ± 4.9 | -0.4 ± 6.2 | 1.9 ± 0.03 |
| SVB | -0.1 ± 0.16 | -0.011 ± 0.009 | 0.007 ± 0.006 | 412.5 ± 9.5 | 412.0 ± 12.4 | 7.1 ± 4.5 | 5.8 ± 3.0 | 5.5 ± 3.7 | 1.9 ± 0.03 |
| TRN | -0.08 ± 0.09 | -0.009 ± 0.007 | 0.041 ± 0.026 | 415.9 ± 13.7 | 415.7 ± 21.0 | 12.2 ± 10.7 | 2.8 ± 5.4 | 3.1 ± 6.1 | 1.9 ± 0.04 |
| BIR | -0.09 ± 0.1 | - | - | 410.7 ± 7.6 | 410.6 ± 10.3 | 6.1 ± 4.1 | - | - | - |
| CMN | -0.03 ± 0.05 | - | - | 408.4 ± 6.7 | 408.2 ± 8.8 | 5.1 ± 2.6 | - | - | - |
| HEL | -0.15 ± 0.25 | - | - | 414.1 ± 9.3 | 414.2 ± 16.7 | 11.1 ± 6.9 | - | - | - |
| IPR | -0.04 ± 0.05 | - | - | 428.8 ± 17.6 | 428.8 ± 26.0 | 16.8 ± 10.3 | - | - | - |
| JUE | -0.07 ± 0.08 | - | - | 417.6 ± 15.3 | 416.9 ± 24.8 | 15.2 ± 15.5 | - | - | - |
| LMP | -0.01 ± 0.27 | - | - | 410.5 ± 4.6 | 410.3 ± 6.5 | 4.5 ± 1.8 | - | - | - |
| LUT | -0.1 ± 0.14 | - | - | 416.8 ± 15.7 | 416.8 ± 24.9 | 14.4 ± 12.7 | - | - | - |
| PRS | -0.02 ± 0.04 | - | - | 408.9 ± 5.0 | 409.0 ± 6.7 | 4.0 ± 2.0 | - | - | - |
| PUI | -0.07 ± 0.12 | - | - | 410.9 ± 6.1 | 411.0 ± 8.1 | 5.1 ± 2.2 | - | - | - |
| PUY | -0.06 ± 0.08 | - | - | 409.4 ± 8.3 | 409.3 ± 11.5 | 6.5 ± 4.2 | - | - | - |
| RGL | -0.11 ± 0.13 | - | - | 409.6 ± 8.3 | 409.6 ± 11.1 | 6.9 ± 3.9 | - | - | - |
| SMR | -0.07 ± 0.13 | - | - | 414.2 ± 10.6 | 414.2 ± 13.9 | 7.9 ± 4.6 | - | - | - |
| SSL | -0.06 ± 0.06 | - | - | 410.2 ± 6.6 | 410.4 ± 9.7 | 6.4 ± 3.1 | - | - | - |
| TOH | -0.06 ± 0.09 | - | - | 414.7 ± 11.7 | 414.9 ± 16.4 | 9.8 ± 5.6 | - | - | - |
| UTO | -0.24 ± 0.45 | - | - | 414.2 ± 9.2 | 414.3 ± 14.5 | 9.4 ± 5.0 | - | - | - |
| WAO | -0.06 ± 0.07 | - | - | 419.5 ± 9.6 | 420.2 ± 19.5 | 14.0 ± 7.5 | - | - | - |
| WES | -0.08 ± 0.12 | - | - | 414.1 ± 10.3 | 414.2 ± 18.6 | 13.0 ± 6.9 | - | - | - |
| ZSF | -0.03 ± 0.04 | - | - | 409.1 ± 5.3 | 409.2 ± 7.4 | 4.7 ± 2.3 | - | - | - |

## Appendix B: Spatial clustering algorithm

The inversion solves for offsets to the prior fluxes at a variable spatial resolution: high (up to 0.25°) in the direct vicinity of observation sites, but lower in parts of the domain that are not well sampled by the observation network. To achieve this, the spatial domain of the inversion is divided into a set of clusters of grid cells, each defined by the following properties:

- cells: the list of grid cells included in the cluster.

- weight: the sum of a property carried by each grid cell. In our case, this property is the average sensitivity of the observation network to that grid cell.

- size: the number of grid cells in the cluster.

- mean_lat, mean_lon: the average (area-weighted) lat and lon of the grid cells in the cluster

- area: the total of all the grid cells included in the cluster.

- type: ocean, land, or mixed.

- continuity: whether it is possible to "walk" from any grid cell of the cluster to any other one or whether there are discontinuities (e.g. a "land" cluster separated in two parts by ocean grid cells).

The objective of the clustering algorithm is to divide the domain into a user-defined number of continuous clusters with roughly equal "weight". The "weight" of a single grid cell is, in our case, defined as the average value of the adjoint field in that grid cell for an adjoint simulation driven by model-data mismatches set proportional to the uncertainty of each observation. The clustering is performed iteratively as follows:

1. Initially, one single cluster is formed, comprising all grid cells of the domain. It is added to a pool of "dividable" clusters.

2. The "weight" of all clusters in that pool is calculated (i.e. the weight of the single initial cluster at the first iteration);

3. The cluster with the largest weight is then split into two even parts across its longest axis (i.e., in an eastern and western part, at the first iteration);

4. The resulting two new clusters are checked for continuity. If needed, they are further split into several continuous clusters;

5. If a cluster reaches the minimum size (1 grid cell), it is moved to a pool of "defined" clusters.

6. If the total number of clusters ("dividable" plus "defined") is lower than the target number of clusters, then repeat steps 2 to 6. Otherwise, exit.

Because of how the cluster weights are defined, clusters away from observation points end up being considerably larger, but they are in regions where the inversions would have applied very smooth flux adjustments, so there is no real drawback to this clustering.

*Author contributions.* All authors designed the experiments, CG and GM developed the code, SB provided the $\Delta^{14}CO_2$ data, and CG performed the simulations. CG prepared the paper, and GM, SB, and MS provided corrections and suggestions for improvements.

*Competing interests.* The authors declare that they have no conflict of interest.

*Acknowledgements.* We thank the Swedish Research Council for Sustainable Development FORMAS for funding the 14C-FFDAS project (Dnr 2018-01771). MS, GM, and CG acknowledge support from the EU projects AVENGERS (Grant Agreement (GA): 101081322) and CORSO (GA: 101082194) as well as from the three Swedish strategic research areas ModElling the Regional and Global Earth system (MERGE), the e-science collaboration (eSSENCE), and Biodiversity and Ecosystems in a Changing Climate (BECC). SB acknowledges the National Aeronautics and Space Administration NASA grant 80NSSC21K1708 and NASA/ESSIC cooperative agreement 80NSSC23M0011. The computations were enabled by resources provided by the National Academic Infrastructure for Supercomputing in Sweden (NAISS), the Swedish National Infrastructure for Computing (SNIC) at LUNARC, and NSC partially funded by the Swedish Research Council through grant agreements no. 2022-06725 and no. 2018-05973, and the Royal Physiographic Society of Lund through Endowments for the Natural Sciences, Medicine and Technology - Geoscience.

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
