# Peer review of "A $CO_2$ - $\Delta^{14}CO_2$ inversion setup for estimating European fossil $CO_2$ emissions"

_EGUsphere, 2023_

## Author Comment (AC1)

Lund, March 10th, 2024

**Carlos Gómez-Ortiz**
**Department of Physical Geography and Ecosystem Science**
**Lund University**
**Sweden**

**Editor and Referees assigned to Research article EGUSPHERE-2023-2215.**
**Atmospheric Chemistry and Physics (ACP)**

Dear Editor and Referees,

We thank both referees for their insightful comments and constructive suggestions. They have contributed to significantly improving the manuscript and its discussion. In the following, we address their comments point-by-point. We use text in italics to repeat the referees' comments, normal text for our response, and the marked-up text from the manuscript showing the changes applied.

**RC1**

*RC1: The manuscript delves into the pivotal task of independently estimating and verifying regional and national fossil CO2 emissions, employing the Lund University Modular Inversion Algorithm (LUMIA) for assimilating in situ observations over Europe. The study's foundation lies in the assimilation of data from the Integrated Carbon Observation System (ICOS) network, a crucial aspect that warrants attention. However, the paper falls short in clearly articulating the novel contributions it brings to the existing body of inverse modeling studies. The reliance on the ICOS network is evident, but the connection to prior studies and the specific advancements provided by this investigation are not well-defined.*

*A critical aspect to address is the lack of clarity on why earlier studies might have failed or why they were conceptually, or due to data limitations, unable to address the question posed in the title. The reviewer suggests considering aspects such as bio and oceanic recycling of dispersed 14C, which could have been potential challenges or gaps in previous research. Providing insights into these aspects would enhance the reader's understanding of the study's significance in addressing potential limitations or gaps in existing literature.*

We have extended the Introduction (L75-109 and 136-139 of the revised manuscript) and put the objective of our study in place with the existing body of inverse modeling studies focusing on the estimation of both fossil $CO_2$ emissions and terrestrial $CO_2$ fluxes using a multi-tracer ($CO_2$ and $\Delta^{14}CO_2$) approach. Clearly, the existing body of literature is very limited regarding such inversions on a continental scale, which is the specific objective of our study here. We do not claim that earlier studies have failed or could not address the problem of constraining fossil $CO_2$ emissions over Europe using $\Delta^{14}CO_2$ observations. In fact, to our knowledge, there is only one other existing study (Wang et al., 2018a) that addressed this specific problem. Our approach here is based on a very different modeling set-up than that of Wang et al. (2018) (e.g. transport model, resolution, $\Delta^{14}CO_2$ modeling approach), and hence contributes to the estimation of model uncertainty.

Besides, this is the first time LUMIA is used in a multi-tracer approach and this manuscript serves as a model description reference paper for future studies on some of the important questions raised by the referee such as the impact of the terrestrial disequilibrium on the inferred fluxes.

[revised manuscript text omitted]

*Despite these concerns, the paper effectively demonstrates LUMIA's capabilities in well-sampled regions, showcasing its potential for accurate estimation of fossil CO2 budgets. The challenges faced in regions with low sampling coverage are acknowledged, shedding light on the limitations of the applied methodology in certain contexts.*

*Furthermore, the study underscores the importance of a reliable prior estimate of terrestrial isotopic disequilibrium, emphasizing the need to minimize uncertainties for robust posterior fossil CO2 flux estimates. This aspect adds valuable insights to the methodology used in estimating and verifying fossil CO2 emissions.*

*In summary, while the study contributes valuable information regarding fossil CO2 emissions, addressing the critique by explicitly stating the novel results in relation to prior studies, highlighting potential limitations, and discussing alternative explanations, particularly related to the bio and oceanic recycling of dispersed 14C, would significantly strengthen the paper.*

As mentioned above we have extended the Introduction to put our study in context to previous studies and explained the novel aspect (LUMIA as a multi-tracer inversion system) of the manuscript. We also highlight the potential limitations and open questions of employing our system for estimating fossil $CO_2$ emissions. We believe this paper should serve as a model description reference and will address the open questions such as $\Delta^{14}CO_2$ sampling strategies, terrestrial disequilibrium, and $\Delta^{14}CO_2$ emissions from nuclear power plants in detail in a follow-up study. In addition to the adjustments made to the introduction in this regard, we updated the discussion to further comment on this regard.

*Specific points:*

*Prior to publication, a number of points need to be addressed, both in terms of content and methodological description. In detail:*

*Help the reader to make the paper more self-consistent. References to previous LUMIA papers and others (e.g. Chatterjee and Michalak, 2013; Rayner et al., 2019; Scholze et al., 2017) should not be overused as a substitute for a more comprehensive description of the inversion model. See below for specific points.*

*RC1: L120: What is a shifted delta? This should be defined in a mathematically sound way.*

We are not certain what exactly the referee means by 'shifted delta' because we do not use this terminology in the manuscript. We explained the meaning of capital delta (Δ) in lines 110 to 113 of the original manuscript. Nevertheless, we have rephrased the explanation of $\Delta^{14}CO_2$ and clarified the delta notation in L152-153 of the revised manuscript.

*RC1: There is further confusion in the transition from the rather traditional formulation of the cost function (6) (line 172) to the use of a space-time covariance matrix B (see also sloppy use of simple or bolded notation of B). Some specific details:*

We updated Equation 6 and the subsequent use of the matrix terms B, H, and R to straight bolded notation, consistent with other studies. The use of a space-time covariance matrix is completely standard in the research field, and our paper does not depart from the norm in that respect (see e.g. Broquet et al., (2011), Monteil & Scholze, (2021), Munassar et al., (2023))

$$J(x) = \frac{1}{2}(x - x_b)^T \underline{B}\mathbf{B}^{-1}(x - x_b) + \frac{1}{2}\left(\underline{Hx - y}\mathbf{H}x - y\right)^T \underline{R}\mathbf{R}^{-1}\left(\underline{Hx - y}\mathbf{H}x - y\right) \tag{6}$$

240    where $\underline{B}$ $\mathbf{B}$ is the prior uncertainty covariance matrix, and $\underline{R}$ $\mathbf{R}$ is the observational uncertainty covariance matrix, controlling the weight of each observation and target variable in the optimization. The iterative procedure searches for the value of $x$ that  minimize $J(x)$, i.e. the value of $x$ for which the gradient $(\nabla_x J)$ is equal to zero. The observation operator $H(x)$ can be expressed as the Jacobian matrix $\underline{Hx}$ $\mathbf{H}x$ that stores the sensitivity of each observation to each control vector element (Monteil and Scholze, 2021).

*RC1: L192-195: Confusing description of eqn (7): Adding a matrix TTXTH to a vector Fc0? Please rephrase and give more explanatory details. As it stands, this is of little use. Define T and X more precisely!*

There is indeed a small error: $\mathbf{F}_c$ with shape $(n_{mod}^t, n_{mod}^h)$ is a matrix and not a vector. We corrected this in L236 of the revised manuscript. The remainder of the text is correct.

$$\underline{F}\mathbf{F}_c = \underline{F}\mathbf{F}_c^0 + \mathbf{T}_T \mathbf{X}_x^c \mathbf{T}_H \tag{7}$$

260    where $\underline{F_c \text{ is the vector}}$ $\mathbf{F}_c$ is the matrix containing gridded emissions for the category $c$, with prior value $\underline{F_c^0}$ $\mathbf{F}_c^0$. The matrix

*RC1: L202: Why spatio-temporal for the diagonal matrix elements? This is just autocorrelation. Spatial correlations between two different locations are off-diagonal elements. Is this a block diagonal construction?*

The sentence refers to the spatiotemporal pattern of the uncertainties (i.e. the spatiotemporal distribution of the variances), not of their correlations. The text seems clear enough to us (and to the second referee), so we chose not to modify it.

*RC1: L203: Explain the temporal formulation together with the related segmentation of the vectors x / F (see also eq 6) for the cause of confusion, where x is used in a traditional space phase vector.*

We believe that what the referee asks is exactly what is provided in L211-216 of the original manuscript. Lines 202-210 describe in a generic way the process used to achieve a target's overall uncertainty, independently of what they are based upon.

*RC1: L207: Is the scaling controlled by some a posteriori technique? For systematic approaches see Talagrand, O. 1997 Assimilation of observations; an introduction. J. Meteorol. Soc. Japan, 75; Desroziers et al, 2005, QJRMS, Diagnosis of observation, background and analysis-error statistics in observation space.*

No, the scaling is set (prescribed) by us. The target uncertainties are given in Table 2 (as indicated in line 223 of the original manuscript).

*RC1: L214: This optimisation time step of one week should be explained in more detail above, along with the optimisation concept. Again, the implications for the construction of the state vector x are not sufficiently addressed to be understood.*

The optimization time step (now replaced in the text by 'optimization interval' to not be confused with the model time step) refers to the temporal resolution of the flux adjustments: the state vector contains weekly offsets to the prior emissions, whereas the prior emissions are hourly: the emissions within an optimization interval will be adjusted by the same value. This is already described in Section 2.3.1 (L185-191 of the original manuscript). There is no implication on the construction of the state vector (rather, the opposite: the way the state vector is constructed determines how the **B** matrix should be constructed).

We would like to point out that the overall approach is very standard in the field, similar to what has been used in other studies, both with the same inversion system (e.g. Monteil et al. (2020), Monteil & Scholze (2021), Munassar et al. (2023)) and with other inversion systems (e.g. (Basu et al., 2016) (TM5), Broquet et al. (2011, 2013) (PYVAR-CHIMERE)), therefore, it does not seem appropriate to re-describe this in details. We describe what makes the specificity of our implementation of that approach, but we think that we do not need to explain the fundamentals of inversions here and refer to other studies for a more fundamental understanding.

*RC1: Figure 3: Caption and subscript of the left panel show 1-hour integration backward in time. Is there evidence of hypersonic winds?*

The caption indicating a 1-hour sampling integration for $CO_2$ does not imply the presence of hypersonic winds. Instead, the 1-hour integration time refers to the period over which atmospheric data are collected and synthesized to represent the $CO_2$ sample at a specific moment. The plume or back trajectory displayed in the maps by the black to orange colors represents the sensitivity of the sampled atmospheric $CO_2$ to the surface fluxes over the 14 days before the sampling time (or the starting sampling time in the case of $\Delta^{14}CO_2$). This methodology is standard in atmospheric inversions for capturing the influence of regional fluxes on a sampling site and does not suggest unusual wind speeds.

We updated Figure 3 and removed the text "Integration: 1h" from the left panel to not generate further confusion. We added the text "Continuous samples" to the left panel and "Integrated samples" to the right panel, and updated the caption.

[Figure]

**Figure 3.**  Examples of so-called (pre-calculated) footprints for $CO_2$ (left) and $\Delta^{14}CO_2$ (right) at the Hyltemossa ICOS station. The maps show the sensitivity of the respective atmospheric tracer at the sampling site to the surface fluxes over the regional domain up to two weeks before the observation. The left panel displays the sensitivity of $CO_2$ at the indicated sampling time and shows influences by surface fluxes from the North Atlantic through Scandinavia, while the right  panel demonstrates the dispersed sensitivity of a 14-day integrated $\Delta^{14}CO_2$ sample across Northwestern Europe and the Baltic region. The two maps illustrate the distinct spatial integration  of  the two tracers over time.

*RC1: Lines 266-269: The statement is difficult to reconcile with the displayed extension in the graphics. Please clarify the meaning of 'm(?) 3'.*

As explained before, the purpose of the statement and the figure is to show and describe the simulated sensitivity of the two types of samples used in the study for a specific sampling time: continuous for $CO_2$ and integrated for $\Delta^{14}CO_2$. 'm(?) 3' is a typo and was removed from the text (see L313 of the revised manuscript).

*RC1: Lines 282-283: Do you conduct experiments with identical twins instead of OSSEs?*

In our study, we performed OSSEs. To clarify this, we modified the text in L330-337 of the revised manuscript.

355 is sampling the free troposphere.  For our OSSEs, we use the same transport model ( i.e. the pre-computed observation footprints from FLEXPART) to generate the synthetic observations and perform the inversions . Therefore, this data selection is not strictly necessary for this study, but we want to replicate the conditions of a real inversion.  Since we are using the same background concentration for the synthetic observations and the simulated prior and posterior observations (i.e. we are

360 assuming a perfect boundary condition), we simplify the calculation of it by computing a smoothed and detrended weekly (for $CO_2$) and monthly (for $\Delta^{14}CO_2$) average of the real observations (ICOS et al., 2023) for each sampling site. For sampling sites, for which there are for some reason no real observations for the year 2018 in the ICOS database (e.g. $\Delta^{14}CO_2$ measurements were not yet implemented or were not yet part of ICOS), we took the observations from the nearest year available to calculate the background.

OSSEs are the reference term in the field of study for this kind of sensitivity as shown in other studies such as Basu et al. (2016, 2020), Philip et al. (2019), Wang et al. (2018).

*RC1: Please declare sources for the values estimated along external information in L300-301.*

The text was modified in L357-365 of the revised manuscript to give more clarity.

385   flux uncertainties (See Section 2.3.1). Since our main purpose in this study is to demonstrate that our multi-tracer inversion system is capable of estimating both the fossil $CO_2$ emissions and natural $CO_2$ fluxes, we choose prior uncertainty values that are reasonable and consistent with other studies. The prior uncertainties are assigned as follows:  for $F_{ff}$, we use the difference between the annual budgets for the whole study domain from ODIAC (1.26 $PgC\ yr^{-1}$ ) (Oda and Maksyutov, 2020) and from an emissions product based on EDGARv4.3 (1.47 $PgC\ yr^{-1}$) (Gerbig and Koch, 2021b)

390   as a reference to define its uncertainty (Basu et al., 2016). We use 150% (0.3 $PgC\ yr^{-1}$) of the difference  as the base uncertainty for all the experiments, and we select two extra values to evaluate the impact of the prescribed uncertainty on the inversion: 50% of the difference (0.1 $PgC\ yr^{-1}$   and the exact difference of 0.21 $PgC\ yr^{-1}$  100%). For $F_{bio}$ we choose the 25% (0.37 $PgC\ yr^{-1}$) of the monthly prior

395   (Monteil and Scholze, 2021), and 30% (0.22 $PgC\ yr^{-1}$) of the annual budget for $F_{biodis}$ (Basu et al., 2020). We optimize all the

The same methodology was implemented by Basu et al. (2016), in which they defined the uncertainty for the fossil fuel fluxes as the spread among a series of emission inventories and products in the U.S. (CarbonTracker, VULCAN, and ODIAC). However, we need to point out that, while this is a reasonable value, we intend to demonstrate that our system works and can perform multi-tracer inversions using observations of $CO_2$ and $\Delta^{14}CO_2$.

*RC1: In L305-309, are there errors of representativity?*

In our inversion system, the observation error represents both the instrumental error and the model representation error. In the case of the $CO_2$ observations, the observation error is mainly composed of the error of representativity, since the instrumental error is very small (in the order of 0.1 ppm), and by calculating the moving standard deviation in a weekly window we calculate an error proportional to how rapidly $CO_2$ varies (for background sites it will be small while for polluted sites it will be larger). For $\Delta^{14}CO_2$, the opposite is true. The instrumental error is larger than the error of representativity, therefore, we pick a value of 0.8 ppm in $C\Delta^{14}C$ units (1.91±0.05‰ $\Delta^{14}CO_2$).

Note: There was confusion generated by the value used as the observation error of $\Delta^{14}CO_2$ and the unit conversion from ppm to ‰ that was pointed out by RC2 and was included in the manuscript.

We modify the text in L367-375 of the revised manuscript to clarify this.

(500 points). To set up the observation  error, which includes the instrumental and the representativity errors, we use different methods for the $CO_2$ and the $\Delta^{14}CO_2$. For $CO_2$, where the error of representativity is usually larger than the instrumental error, we apply a weekly moving standard deviation to each observation i.e. the prior  error of each

400 observation is equal to the standard deviation of the observations in a time window of $\pm 3.5$ days around that observation.  In this way, we account for the changes in the $CO_2$  concentrations according to the local site conditions. For instance, at a background station such as Jungfraujoch (JFJ) on the top of the Swiss Alpes, the observation error ranges from 0.9 to 29.2ppm (mean value of $9.3 \pm 4.0$ppm), while at polluted sites such as Saclay (SAC) just outside Paris the $CO_2$ concentrations change rapidly and the error ranges from 5.9 to 215.5ppm

405 (mean value of $55.8 \pm 40.7$ppm). For $\Delta^{14}CO_2$  on the other hand, the instrumental error is larger than the representativity error, we use a constant value of  0.8ppm in $C\Delta^{14}C$ units or $1.91 \pm 0.05$‰ in $\Delta^{14}CO_2$ units, calculated using Equation 8.

*RC1: The caption in Figure 4 is of little value, please add a significantly extended description to the 8 panels.*

We extended the description and updated the figure to enhance its value.

[Figure]

**Figure 4.** Synthetic  observations of $CO_2$ and $\Delta^{14}CO_2$ at the HTM  station over a one-year period. Panels a) to d) display $CO_2$ concentration variations due to different sources: a) fossil fuel, b) biosphere, c) ocean, and d) combined synthetic observations with random perturbation (blue dotted line) against the background concentrations (red dashed line). Panels e) to h) illustrate $\Delta^{14}CO_2$ variations: e) nuclear and fossil fuel, f) biospheric disequilibrium and biosphere, g) ocean disequilibrium and ocean, and h) total synthetic observations with random perturbation (blue dotted line) compared to the background (red dashed line). The blue solid and dashed lines represent the synthetic observations without and with random noise added, respectively.

*RC1: In Section 4.2.1: The subsection could benefit from additional analyses on the air mass pathways for other stations, following the scheme provided in Fig. 3 for a single station. The following footprints aim to explain the reasons for the varying performances across the individual regional domains that were analyzed.*

Figure 3 is a "snapshot" of the sensitivity for one specific hourly $CO_2$ and integrated $\Delta^{14}CO_2$ sample at one sampling station. Figure 2 is a more statistical representation of the overall sensitivity in the study. In regions where there is a high sensitivity (as shown in Figure 2 for Western/Central Europe where most yellow clusters are located), there is a better constraint of

the emissions as shown in Figure 6. An expanded analysis, as suggested, would require evaluating individual timesteps for each station to construct a detailed understanding of the air mass pathways. This would involve a substantial increase in computational resources and time, as each station's footprints need to be analyzed for each timestep to ascertain the variances in air mass influences. Moreover, the spatial and temporal resolution of the data, along with the model's inherent limitations in resolving complex atmospheric processes, may introduce additional uncertainties. It is not possible, and to our understanding also not needed, to provide the footprint for every observation and every site.

*RC1: In line 382, please define the term 'pixel-level'.*

We replaced the term "pixel-level" with "grid cell level" (L441) which is the correct term.

*RC1: Additionally, in line 386, please specify which samples were used to obtain the statistics. It would be clearer to state 'average values of...' and 'standard deviation' with respect to the underlying samples.*

We updated Equation 9 to make it consistent with the metrics used in the previous subsection (4.2.1 Retrieval of the monthly and regional time series).

We additionally updated Figure 7 to show the individual maps of the posterior RMSE for both experiments and both categories and updated the text to include the changes in the equation and the figure.

[Figure]

**Figure 7.** Spatial error of fossil $CO_2$ (a to d) and biosphere (e to h) for the ZBASE and ZCO2Only experiments. a) and c) show the prior RMSE for $F_{ff}$ and $F_{bio}$, respectively,  b) and f) show the posterior RMSE for ZBASE, c) and g) show the posterior RMSE for ZCO2Only, and d) and h) show the  RMSE reduction (see Equation 9) for fossil and biosphere. In Figures d) and h), positive values (in blue) show the pixels where ZBASE performs better than ZCO2Only (i.e. adding  $\Delta^{14}CO_2$ observations improves the posterior estimates), and negative values (in red) where ZCO2Only performs better than ZBASE. Crosses and diamonds represent stations that only measure $CO_2$ and those that additionally measure  $\Delta^{14}CO_2$, respectively.

*RC1: Lastly, please revise line 400. In the northern part of*

The sentence was changed to "[…] the northern part of Western/Central Europe, Denmark, and southern Sweden, as well as some areas in Eastern Europe." (see L456 of the revised manuscript).

*RC1: L475: the conclusion drawn that the inversion process has effectively adjusted the model outputs, bringing them closer to the true observations, is scientifically poor. It is recommended to provide more quantitative evaluation to support this claim.*

The whole section was completely modified following this comment and comments from RC2. We included the analysis of a polluted station (Saclay), along with JFJ (background), and added Figures 13 and 14, and Table 5 showing the performance metrics.

**4.3.3 The observational space**

[revised manuscript text omitted]

*RC1: In line 480, the text asks how the Chi2 validation is performed by designing B. It is unclear what is meant by 'designing B'. The text could be improved by providing more context and explanation for the question being asked.*

We do not understand the referee's comment. The term 'designing B' does not appear in this section, nor the whole document.

*RC1: The options for the validation method are Talagrand, O. 1997 Assimilation of observations and Desroziers et al, 2005, QJRMS, Diagnosis of observation, background and analysis-error statistics in observation space.*

We find his suggestions out of the scope of this study.

*RC1: "To improve numerical accuracy, it may be necessary to normalize the fluxes or implement regional scaling factors." A sound approach involves preconditioning techniques from minimization methods. Please provide your comments on this suggestion.*

The preconditioning technique was implemented since the initial development of LUMIA (see Monteil & Scholze (2021)). We realized that this was not mentioned anywhere in the text and we included it in section "2.3.1 Construction of the control vector (*x*)" (see L240-242 of the revised manuscript).

> 0.5°) to each optimized time-step $t_{opt}$ and cluster $p_{mod}$. To reduce the number of iterations and large matrix multiplications, the
>
> 265    optimization is performed on a preconditioned control vector $\omega = \mathbf{B}^{-1/2}(x - x_b)$. More information about the preconditioning can be found in Monteil and Scholze (2021).

This suggestion was completely removed from the discussion by the recommendation of the second referee.

**RC2**

*The main question of the paper by Gomez-Ortiz et al. is already in the title: "Can Δ14CO2 observations assist atmospheric inversions in constraining Europe's fossil CO2 emission budget?" The authors employ the Lund University Modular Inversion Algorithm (LUMIA) to estimate fossil CO2 emissions and natural fluxes by simultaneously inverting in-situ observations of CO2 and Δ14CO2 across Europe. They evaluate the system's performance through a series of Observing System Simulation Experiments (OSSEs). Their main result is that in regions with dense observation networks, like Western/Central Europe, LUMIA, with the inclusion of Δ14CO2 observations, can reconstruct the emissions' time series and the fossil and biogenic CO2 fluxes. However, in regions with lower observation coverage, such as Southern Europe and the British Isles, estimating fossil CO2 emissions is less successful.*

*The paper discusses a current and significant topic, showcasing the LUMIA system's capability to utilize both CO2 and 14CO2 data simultaneously. However, the manuscript's discussion of the results is often too descriptive and fails to address the underlying processes. As a result, the study requires major revisions before it can be published.*

*Overarching comments:*

*There are certain parts in the manuscript where the language needs to be more accurate and specific. For instance, the national emission data that is reported to the UNFCCC cannot be compared with the emission inventories that are distributed spatially and temporally. However, both are referred to as "inventories" in the text. For experienced readers, the context might be*

*clear, but for new readers, the language needs to be more precise and differentiated in many places.*

We updated the manuscript to use the correct language.

*The manuscript contains many estimates of posterior CO2 emissions at regional and national levels, but it fails to mention the uncertainties associated with these estimates. To address this issue, an ensemble approach can be used that takes into account different realizations of synthetic observations and various prior uncertainties.*

We performed a Monte Carlo ensemble of 100 members to calculate the posterior uncertainties and updated Figures 8 and 10 and the corresponding results and discussion.

[Figure]

**Figure 8.**  True, prior and posterior annual budgets of fossil (a-b), biosphere (c-d) and total $CO_2$ (e-f) for the study domain, the sub-regions (right), and some of the largest European countries by area (left). The white bars show the true  annual budgets based on  EDGAR  and LPJ-GUESS flux products. The  black bars  represent  the  prior value,  0 PgC. The blue  and  green bars show the posterior  budgets of ZBASE  and ZCO2Only  respectively. The  error bars represent the prior  and posterior uncertainty calculated with a Monte Carlo ensemble of 100 members.

[Figure]

**Figure 10.** Total annual fossil $CO_2$ emissions for the study domain, Western/Central Europe, Eastern Europe, Northern Europe, Germany, and Poland. The white bars show the true emissions based on the EDGAR emission database. The red bars show the prior fluxes based on the ODIAC emission data product. The blue, green and tan bars show the posterior fossil $CO_2$ emissions for the BASE0.1, BASE0.21, and BASE0.3  experiments, respectively. The error bars represent the prior and posterior uncertainty calculated with a Monte Carlo ensemble of 100 members.

*The OSSE assumes a constant nuclear 14C contamination. However, as the authors write in their summary, this does not reflect reality. An OSSE study on 14CO2 in Europe is predestined to analyse the influences of variable nuclear contamination. The authors should calculate an additional scenario for this purpose.*

We mentioned this in the 'Discussion' and 'Conclusions and future perspectives' sections in the original manuscript. We feel that adding a scenario for this purpose is beyond the scope of this manuscript (which serves as a model description reference paper). A more detailed analysis of the influence of a range of influencing variables (including nuclear contamination) will be investigated in a follow-up study.

*Below are general remarks on specific sections. Note that these comments do not cover all minor issues such as grammatical errors, missing words, or imprecise wording. After addressing the general remarks, a second review should take care of these smaller issues.*

*Section 1:*

*RC2: The literature cited lacks clarity and omits fundamental publications, particularly regarding the basic 14C cycle, or is citing them only indirectly.*

The Introduction section was thoroughly updated to satisfy the concerns of both referees regarding existing literature in the field of using radiocarbon observations for estimating fossil $CO_2$ emissions (L75-109 and 136-139 of the revised manuscript).

[revised manuscript text omitted]

*Section 2.1:*

*RC2: The regional model is presented, but not a word is said about the boundary conditions and how these were realised.*

We added an explanation of the boundary condition calculation to Section 3.3 (L331-337 of the revised manuscript) and commented on the perfect transport and perfect boundary conditions in the discussion (L630-652 of the revised manuscript).

not strictly necessary for this study, but we want to replicate the conditions of a real inversion.  Since we are using the same background concentration for the synthetic observations and the simulated prior and posterior observations (i.e. we are
360   assuming a perfect boundary condition), we simplify the calculation of it by computing a smoothed and detrended weekly (for $CO_2$) and monthly (for $\Delta^{14}CO_2$) average of the real observations (ICOS et al., 2023) for each sampling site. For sampling sites, for which there are for some reason no real observations for the year 2018 in the ICOS database (e.g. $\Delta^{14}CO_2$ measurements were not yet implemented or were not yet part of ICOS), we took the observations from the nearest year available to calculate the background.

Discussion

The Observing System Simulation Experiment (OSSE) framework used in this study assumes a perfect realization of atmospheric transport and mixing processes by employing the same transport model across the simulations. This assumption simplifies the complex nature of atmospheric dynamics and is a common approach to limit the scope of variability in such studies. However, it is crucial to acknowledge that this simplification overlooks one of the largest sources of uncertainty in
730   atmospheric inverse modeling: the accurate representation of atmospheric transport and mixing processes. The variability and uncertainty in atmospheric transport can significantly impact the estimation of greenhouse gas sources and sinks. As demonstrated by Schuh et al. (2019), inconsistencies in transport simulations can introduce systematic biases in surface flux estimations, which can be as substantial as 1.7 PgC year$^{-1}$ for large zonal bands. In a study by Munassar et al. (2023), in which multiple combinations of global and regional models were tested using two different inversion frameworks (LUMIA
735   and CarboScope-Regional (CSR)), they found that using a different regional transport (FLEXPART and STILT (Stochastic Time-Inverted Lagrangian Transport)) model can cause differences in the posterior NEE annual budget of 0.51 PgC year$^{-1}$. This highlights the sensitivity of inversion-derived emission estimates to the accuracy of the transport model used and emphasizes the critical role that transport uncertainty plays across global flux inversion systems.

Furthermore, the assumption of perfect boundary conditions in the model presents another significant simplification. Boundary
740   conditions in atmospheric modeling can greatly influence the concentration gradients and flux estimates, and their mischaracterization can propagate errors throughout the model domain. Coming back to the study by Munassar et al. (2023), the use of a different global transport model (TM3 and TM5) for the estimation of the boundary condition can cause discrepancies in the posterior annual budget as large as 0.23 PgC year$^{-1}$. Errors in these aspects of the transport model could lead to skewed emission estimates. Given these considerations, the presented results should be interpreted with caution, understanding that the true
745   uncertainty in atmospheric inverse modeling is likely understated in these OSSEs. It underscores the need for more comprehensive approaches that account for transport model uncertainties, such as employing ensemble modeling techniques that incorporate multiple transport models and boundary conditions to better capture the inherent uncertainties in atmospheric dynamics (Locatelli et al., 2015; Aleksankina et al., 2018).

*RC2: Eq. 3: What assumptions are made here concerning the d13C?*

Since the nuclear emissions have a resolution of 1 year, we assumed $\delta^{13}C$ as the global atmospheric average value reported by NOAA, as used in studies such as Basu et al. (2016). We included this explanation in the revised manuscript in L176-177.

*Section 2.3.2:*

*RC2: Please begin this section by pointing out that this construction of the B matrix is about determining their spatiotemporal structure and that the absolute magnitude of the uncertainty is afterwards scaled with the reported uncertainties.*

The text was modified as suggested in L244-245 of the revised manuscript.

**2.3.2 Construction of the prior error covariance matrix (B)**

Our matrix B is constructed such that we first determine the spatio-temporal structure of the uncertainties, which is then scaled to match the reported uncertainties. Since we are optimizing for offsets, the prior control vector $x_b$ contains only zeros (so

*Section 3:*

*RC2: In the introduction of the OSSE, you should point out that this OSSE uses the same transport model, and thus, a perfect realisation of the atmospheric transport and mixing processes is assumed. This ignores one of the largest sources of uncertainty existing for inversions.*

We included this in the introduction of Section 3 as suggested by the referee (L331-337 of the revised manuscript), and commented on this in the discussion (L630-652 of the revised manuscript).

*Section 3.3.*

*RC2: Where do the synthetic background concentrations for CO2 and Δ14CO2 come from? Was TM5 used for the background? I can't find anything about this in the manuscript.*
As mentioned in a previous answer to the referee, we added the explanation on the background concentration calculation in Section 3.3.

*L274: There are no gaps during sampling that can be attributed to the calibration.*

We agree with the referee and removed this from the text (see L321-322 of the revised manuscript).

In this way, we account for the sampling gaps and the differences in integration times commonly produced due to  maintenance, and general operational eventualities. For stations with the number of observations, $N_{obs}$, equal to zero in Table

*L285: CΔ14C -> Δ14C as you also show the 14C in Fig 4 and not the C14C. How large was the random perturbation which was added to the data?*

We updated CΔ$^{14}$C with Δ$^{14}$CO$_2$ and added a sentence explaining the calculation of the random perturbation added to the synthetic observations (see L341-343 of the revised manuscript).

site, observation time and tracer. To weaken the assumption of a perfect transport and boundary condition, we add a random perturbation to the synthetic observations . This random perturbation is equal

370 to $y^* = y + \varepsilon \times \xi$, where $y$ is the synthetic observation, $\varepsilon$ is the observation error (both the instrumental and representativity errors, see Section 3.3.1 below), and $\xi$ is a standard normal random vector. In this way, the added perturbation is based on the observation error. Figure 4 shows the synthetic $CO_2$  and $\Delta^{14}CO_2$ observation time-series and the components

*Section 3.3.1:*

*What is the motivation behind the definition of the prior uncertainties?*

*For Fbio and Fbiodis, this is not motivated in detail. What is the rationale for setting the bio error as 10% of negative Fbio flows? Likewise, for the 30% for the Fbiodis?*

We updated the text in L357-365 of the revised manuscript to better motivate the choice of our prior uncertainties. We also modified the explanation of the $F_{bio}$ uncertainty to be more accurate. As described in the manuscript, our aim with this study is to demonstrate the capabilities of the multi-tracer LUMIA system. Therefore, we focused on selecting uncertainty values that are reasonable and consistent with other studies.

385 flux uncertainties (See Section 2.3.1). Since our main purpose in this study is to demonstrate that our multi-tracer inversion system is capable of estimating both the fossil $CO_2$ emissions and natural $CO_2$ fluxes, we choose prior uncertainty values that are reasonable and consistent with other studies. The prior uncertainties are assigned as follows:  for $F_{ff}$, we use the difference between the annual budgets for the whole study domain from ODIAC (1.26 PgC yr$^{-1}$ ) (Oda and Maksyutov, 2020) and from an emissions product based on EDGARv4.3 (1.47 PgC yr$^{-1}$) (Gerbig and Koch, 2021b)

390 as a reference to define its uncertainty (Basu et al., 2016). We use 150% (0.3 PgC yr$^{-1}$) of the difference  as the base uncertainty for all the experiments, and we select two extra values to evaluate the impact of the prescribed uncertainty on the inversion: 50% of the difference (0.1 PgC yr$^{-1}$   and the exact difference of 0.21 PgC yr$^{-1}$  100%). For $F_{bio}$ we choose the 25% (0.37 PgC yr$^{-1}$) of the monthly prior

395 (Monteil and Scholze, 2021), and 30% (0.22 PgC yr$^{-1}$) of the annual budget for $F_{biodis}$ (Basu et al., 2020). We optimize all the

*The definition of the prior Fff uncertainty as the difference between EDGAR and ODIAC, where EDGAR is used as truth and ODIAC as prior, is clearer. However, it FORCES the inversion in the BASE scenario to fully utilise the prior uncertainty budget to arrive at the "truth". In the 01Base sensitivity run, it is then even more "expensive" for the inversion algorithm to return to the truth. Some discussion on this would be welcome.*

Following the referee's suggestion, we commented on this in the discussion in L539-600 of the revised manuscript.

The  BASE experiments, in which we use realistic prior fluxes, show that the posterior fossil $CO_2$ emissions are not very sensitive to the prescribed prior uncertainty in regions with a dense sampling network

685 , even when using a low prior $F_{ff}$ uncertainty in which case it is more difficult for the inversion algorithm to recover the true fluxes. As we have observed in previous studies using LUMIA (Monteil et al., 2020; Monteil and Scholze, 2021), the cost of fitting the observations dominates the total cost function value. In this sense, the relative value of the prior uncertainty of $F_{bio}$ against $F_{ff}$

690 is going to significantly impact the spatio-temporal distribution of flux adjustments, but the total uncertainty of the fluxes is of lesser importance since the model has enough freedom to adjust the data. In other words, the error structure and how is it set up for the different flux categories, is going to have more impact than the total prior uncertainty. Both Basu et al. (2016) and Wang et al. (2018) highlight the importance of a regional horizontal correlation and

*Uncertainties of observations:*

*Defining the uncertainty of the observations as the standard deviation of the observations over a 7-day window is incorrect. Over such a long period, the standard deviation of the concentration is dominated by the variable transport and mixing processes in the atmosphere. This has nothing to do with the uncertainty of the measurements. In the manuscript (p.14 l.307), this leads to uncertainties of the CO2 measurements varying from 1 to 215ppm! (See also Table A1.)*

*The selected constant measurement uncertainty of 1.5‰, however, is too optimistic and should be replaced by 2‰. How does the (0.8 ppm ‰) error in Table 2 result?*

*With regard to the 14C measurement error, it would also be extremely interesting for an OSSE to illuminate the difference between 1.5 and 2 ‰ measurement accuracy.*

Indeed, our aim with this definition of the $CO_2$ observation error is to represent the variability of the transport and mixing processes in the atmosphere, which can be larger in polluted sites in contrast with background sites. In our inversion system, the observation error represents both the instrumental error and the model representation error. In the case of the $CO_2$ observations, the observation error is mainly composed of the error of representativity, since the instrumental error is very small (in the order of 0.1 ppm), and by calculating the moving standard deviation in a weekly window we calculate an error proportional to how rapidly $CO_2$ varies.

For $\Delta^{14}CO_2$, in which the instrumental error is larger than the error of representativity, we selected a fixed value of 0.8 ppm $C\Delta^{14}C$ which translates to a mean value of 1.91±0.05‰ $\Delta^{14}CO_2$ using Equation 8 (closer to the 2‰ error suggested by the referee), and not 1.5‰ as previously stated in the original manuscript. Nevertheless, we run new inversions using an observation error of 0.9 ppm $C\Delta^{14}C$ (2.15±0.05‰ $\Delta^{14}CO_2$) and 1.0 ppm $C\Delta^{14}C$ (2.38±0.06‰ $\Delta^{14}CO_2$), the last one to show the impact of an approximately 0.5‰ in the observation error, as suggested by the referee. There are small differences in the results, but we consider them not to be significant enough and do not change the discussion and conclusions of the manuscript. Replicating Figures 5, 6, 9, and 11 of the manuscript:

Fossil fuel emissions

[Figure]

Figure 5.

Biosphere (NEE) fluxes

[Figure]

Figure 6.

[Figure]

Figure 9.

Figure 11.

We modify the text in L367-375 of the revised manuscript to clarify this.

(500 points). To set up the observation  error, which includes the instrumental and the representativity errors, we use different methods for the $CO_2$ and the $\Delta^{14}CO_2$. For $CO_2$, where the error of representativity is usually larger than the instrumental error, we apply a weekly moving standard deviation to each observation i.e. the prior  error of each observation is equal to the standard deviation of the observations in a time window of $\pm 3.5$ days around that observation.  In this way, we account for the changes in the $CO_2$ concentrations according to the local site conditions. For instance, at a background station such as Jungfraujoch (JFJ) on the top of the Swiss Alps, the observation error ranges from 0.9 to 29.2ppm (mean value of $9.3 \pm 4.0$ppm), while at polluted sites such as Saclay (SAC) just outside Paris the $CO_2$ concentrations change rapidly and the error ranges from 5.9 to 215.5ppm (mean value of $55.8 \pm 40.7$ppm). For $\Delta^{14}CO_2$  on the other hand, the instrumental error is larger than the representativity error, we use a constant value of  0.8ppm in C$\Delta^{14}$C units or $1.91 \pm 0.05$‰ in $\Delta^{14}CO_2$ units, calculated using Equation 8.

*Section 4.2.1*

*This section is long and difficult to read... much is obvious and should be tried to be presented in a shorter and more concise way. For example, the numbers are all given in Table 4 and, therefore, do not need to be included in the text.*

We updated the whole subsection following the referee's recommendation.

**4.2.1 Retrieval of the monthly and regional time series**

445  In general, there is a closer agreement between the truth and the posterior time series for the ZBASE and ZCO2Only experiments across all regions for the biosphere fluxes ($F_{bio}$) (Figure 6)  in contrast to the fossil $CO_2$ emissions ($F_{ff}$) (Figure 5). In the study domain, the  inclusion of $\Delta^{14}CO_2$ observations in the ZBASE experiment yields better performance than ZCO2Only for both flux categories.

450  Specifically, ZBASE exhibits closer alignment to the posterior  with a lower RMSE (see Table 4) indicating a better fit of the seasonality for $F_{ff}$. Similarly, the posterior biosphere fluxes more closely follow the true time series  than the fossil $CO_2$ emissions in both experiments, with  ZBASE

455 outperforming ZCO2Only  in terms of RMSE and BIAS values

460    The regional analysis reflects the influence of the coverage by sampling stations on the inversion outcomes. Western/Central Europe, benefiting from the highest number of stations (18 out of 33 stations considered in this study, 10 of them measuring both tracers),

465     shows the best alignment between the posterior and true time series for $F_{ff}$, especially in the ZBASE experiment  (Figure 5b), while ZCO2Only shows pronounced RMSE and BIAS values (Table 4). Conversely, regions like Eastern

470    Europe (one station measuring both tracers) and the British Isles  (two stations measuring only CO₂of sampling stations. ), despite their lower station coverage, exhibit posterior ZBASE $F_{ff}$ time series that closely approximate the truth, ,

475    with Eastern Europe showing consistent performance throughout the year (panels d and f in Figure 5). However, the posterior ZBASE biosphere fluxes in these  regions do not align as closely with the true values as observed in e.g. Western/Central Europe (panels d and f in Figure 6). In Eastern Europe, the posterior ZBASE shows big differences with the truth during May, June (maximum difference of 0.42 TgC day⁻¹), and later in September, while ZCO2Only shows a better fit during these months

480    but a positive bias the rest of the year (Figure 6d). In contrast, the posterior biosphere flux from the ZCO2Only experiment shows a better fit to the truth than the ZBASE one in the British Isles (Table 4).

       Lastly, Southern and Northern Europe show similar results despite their differences: Northern Europe has better coverage of sampling stations, and its annual truth fossil CO₂ emissions are lower (an average of 0.20 TgC day⁻¹ against 0.59 TgC day⁻¹).

485    In both regions, the posterior $F_{ff}$ of the two experiments is far from the truth (Figures 5c and 5e), while the posterior $F_{bio}$ of both regions and experiments is close to each other, with Northern Europe showing a better fit to the truth than Southern Europe, in which the posterior shows a more pronounced positive bias along the year (Figures 6c and 6e).

**Section 4.2.2:**

*L380: I thought the errors in the Z simulations were larger than indicated in Table 2?*

As we mentioned in the caption of Table 2 and at the beginning of Section 3.3.1, the same uncertainty and error correlation values were used across all the inversions (experiments) in the study.

*To Fig 7:*

*RC2: From Eq. 9 I understand that it is a relative RMSE reduction. If this is true, then why are the units in Fig 7.d g/(m2day)?*

We updated Equation 9 to make it consistent with the metrics used in the previous subsection (4.2.1 Retrieval of the monthly and regional time series) and the units shown in Figure 7.

$$RMSE_{\text{reduction}} = ((RMSE_{\text{ZCO2Only}}^{apos} - RMSE_{\text{ZBASE}}^{apos}) - \mu)/\sigma \tag{9}$$

 Here, positive values

*RC2: The multi-pole structure in the biospheric RMSE reduction in Fig. 7d is striking. However, this is not discussed in the text. In Fig 7c one can see that the prior is spatially relatively smooth. However, since the results of ZBASE and ZCO2 are only mixed in the RMSE_reduction (7d), it is not possible to recognise whether this multipole structure arises from an inversion or from the interaction of the two. In any case, the multipole structure should be discussed more, also regarding a possible overfitting. The formation of the multipole structure cannot be due to the station distribution alone, as there are no stations in South-East Europe, and similar RSME_reductions are achieved in Central-West Europe.*

We updated Figure 7 to show the individual maps of the posterior RMSE for both experiments and both categories and updated the text to comment on the multipole structure. In general terms, the regions where the biosphere fluxes are poorly constrained show higher RMSE values conforming dipoles in the map. We include a comment on this in Section 4.2.2 (see L453-454 of the revised manuscript) and the Discussion (L607-609).

[Figure]

**Figure 7.** Spatial error of fossil $CO_2$ (a to d) and biosphere (e to h) for the ZBASE and ZCO2Only experiments. a) and e) show the prior RMSE for $F_{ff}$ and $F_{bio}$, respectively,  b) and f) show the posterior RMSE for ZBASE, c) and g) show the posterior RMSE for ZCO2Only, and d) and h) show the  RMSE reduction (see Equation 9) for fossil and biosphere. In Figures d) and h), positive values (in blue) show the pixels where ZBASE performs better than ZCO2Only (i.e. adding  $\Delta^{14}CO_2$ observations improves the posterior estimates), and negative values (in red) where ZCO2Only performs better than ZBASE. Crosses and diamonds represent stations that only measure $CO_2$ and those that additionally measure  $\Delta^{14}CO_2$, respectively.

Section 4.2.2

510   observations (Figure 7d). For the biosphere fluxes,  the posterior RMSE maps (Figures 7f and 7g) show the regions that are poorly constrained due to the absence of observations such as the southern part of the domain and the Baltic States.

Discussion

the scale that is relevant given the model setup, as we demonstrate through the time series and annual budget results. Such

700   spatial misattribution is illustrated in the spatial RMSE reduction results for the biosphere fluxes. We can clearly identify the formation of dipoles (clusters of larger RMSE values) in regions with no observations such as the southern part of the study domain and the Baltic States indicating that these areas are underconstrained.

*RC2: The first sentence of the caption of Fig7D refers to 8 images... but only 4 are shown.*

We updated the figure and the corresponding caption in the revised manuscript as shown in the previous answer.

*RC2: L389: …show for each location…?*

We removed the whole sentence since we found it rather confusing.

CO₂ observations (ZCO2Only).

 For fossil fuel, we find  larger

*Section: 4.2.3*

*RC2: All data in Figure 8 are presented without any uncertainty information. A suitable measure for determining uncertainty should be considered and added here.*

We performed a Monte Carlo simulation ensemble of 100 members to calculate the posterior uncertainty. We added this uncertainty to Figures 8 and 10. We also removed the bar "Ref. (ODIAC)" from Figure 8 to not cause any confusion, and updated the caption accordingly.

[Figure]

**Figure 8.**  True, prior and posterior annual budgets of fossil (a-b), biosphere (c-d) and total CO₂ (e-f) for the study domain, the sub-regions (right), and some of the largest European countries by area (left). The white bars show the true  annual budgets based on  EDGAR  and LPJ-GUESS flux products. The  black bars  represent   prior value,  0 PgC. The blue  and  green bars show the posterior  budgets of ZBASE  and ZCO2Only  experiments, respectively. The  error bars represent the prior  and posterior uncertainty calculated with a Monte Carlo ensemble of 100 members.

[Figure]

**Figure 10.** Total annual fossil $CO_2$ emissions for the study domain, Western/Central Europe, Eastern Europe, Northern Europe, Germany, and Poland. The white bars show the true emissions based on the EDGAR emission database. The red bars show the prior fluxes based on the ODIAC emission data product. The blue, green and tan bars show the posterior fossil $CO_2$ emissions for the BASE0.1, BASE0.21, and BASE0.3  experiments, respectively. The error bars represent the prior and posterior uncertainty calculated with a Monte Carlo ensemble of 100 members.

*RC2: What is the reason for the significant underestimation of the total CO2 fluxes in the study domain in both the ZBASE and the ZCO2only simulation (Fig 8e)? Is this an indication that the uncertainties in the prior fluxes are too small?*

The main reason is under-sampling in the whole study domain. As mentioned in the manuscript (and as can be seen in Figure 8) for Western/Central Europe and its countries (Germany, France, and Benelux), where there is a better network coverage (and therefore more samples) the inversion is able to estimate posterior values close to the true values, while in Southern Europe (and Spain) with a sparse sampling network this is not the case.

Section 4.3.1

*RC2: L437: The summer deviation in Western/Central and Eastern Europe are nearly of the same magnitude. The analysis of the annual cycles should go into a little more depth. Is it due to an incorrect BIO prior? Or is an incorrect assumption on the 14C signature of the heterotrophic respiration? Is it due to the stronger dilution of the fossil emissions and, therefore, the smaller signal-to-noise ratio of the 14C measurements? The analysis needs to go into more depth here.*

The prior terrestrial isotopic disequilibrium flux is on purpose incorrect with the aim of showing the impact that it can have on the estimation of fossil $CO_2$ emissions. We commented on this in the discussion in L611-618 of the revised manuscript.

(Figure 11). The prior terrestrial isotopic disequilibrium flux in our experiments is on purpose incorrect with the aim of showing

705   the impact that it can have in the estimation of fossil $CO_2$ emissions. As shown in Figure 11, the maximum difference between the prior and the true $F_{biodis}$ is of the same order of magnitude for Western/Central Europe (2.1 TgC day$^{-1}$) and Eastern Europe (1.3 TgC day$^{-1}$) in July. For $F_{ff}$, however, the difference between the prior and truth is about one order of magnitude larger for Western/Central Europe compared to Eastern Europe (0.03 vs 0.005 TgC day$^{-1}$). This larger difference causes a stronger dilution of the fossil emissions in Eastern Europe, and therefore essentially lowers the signal-to-noise ratio of the $\Delta^{14}CO_2$

710   measurements, and added to the lower network coverage compared to Western/Central Europe, a poorer constrain of the fossil $CO_2$ emissions. According to Turnbull et al. (2009), one of the main contributors to atmospheric $\Delta^{14}CO_2$ is heterotrophic

*RC2: L444: Make a reference to Fig. 10.*

The reference was included in the text (see L504 of the revised manuscript).

*RC2: L447: "… with BASE0.3 having the highest recovery of 92%.". This sentence is slightly misleading. Even in this scenario, the improvement in the DIFFERENCE of the Prior is only around 50%. The figure of 92% probably refers to the total emission, of which approx. 80% will certainly already be "recovered" by the prior.*

We agree with the referee. In the text, we are referring to the recovery of the difference between prior and truth, while the percentage corresponds to the recovery of the total budget. We correct this in L505-507 of the revised manuscript.

> by all three experiments, with a recovery ranging from 30% for BASE0.1 to 45% for BASE0.3 . In Western/Central Europe, the three experiments recover 96% of the truth (around 71% of the difference between true
> 565  and prior), similar to Germany, where the recovery ranges from 94% for BASE0.1 to 97% for BASE0.3 (68% to 82% of the difference). As we find in the time series (Figure 9d), the prior annual budget is very close to the truth both in Eastern

*Section 4.3.2*

*This section lacks the desired discussion of the effects. The summer overestimation of the Fbiodis flux is probably reflected in the summer maximum of the fossil flux. However, this is not discussed. This is also clearly shown in the BASEnoBD control experiment. The fact that Eastern Europe does not improve in this control experiment should not come as a surprise given the assumed station distribution.*
*At this point, the fundamental question arises as to how the third unknown, Fbiodis (in addition to FffCO2 and FbioCO2), can be robustly derived from the two observed variables CO2 and 14C?*

These suggestions, together with the ones for section 4.3., were added to the discussion in L611-618 of the revised manuscript

*Section 4.3.3*

*Why was JFJ selected as a representative station? JFJ certainly has by far the lowest ffCO2 contribution in Central/Western Europe. Also, the fact that the prior correlation for JFJ is only 0.61, whereas it is 0.92 for all stations, shows that JFJ is not a representative station.*

We kept JFJ and added Saclay (SAC) to the analysis to show the parallel between a background and a polluted station. This section was rewritten completely and we added Figures 13 and 14 and Table 5 to complement the results analysis.

**4.3.3 The observational space**

[revised manuscript text omitted]

*Fig 10e: The Synth Obs cannot be seen in the way they are plotted. Please change the line style to allow every piece of information to be seen.*

We updated the figure to improve the visualization as shown in the answer above.

*The random perturbation of the synth CO2 also appears to be too large in this plot, at least for measurement errors (see comments above). If the uncertainties of a real transport model error*

*should also be represented by the large uncertainties, then a fundamental revision of section 3.3 on error determination is required.*

As mentioned in a comment before, the observation error is the aggregate of the measurement or instrumental error and the error of representativity.

*The conclusions drawn at the end of section 4.3.3 are fairly trivial. They merely show that the inversion optimisation algorithm works as it should, but this does not imply that the results are correct.*

We acknowledged this comment when rewriting this section.

*Unfortunately, the authors do not address the interesting mismatch of the prior 14C observations. Fig 12c shows a pronounced overestimation in 14C. The JFJ plot suggests that this overestimation occurs in summer/autumn and, therefore, likely originates from Fbiodis. All of this reflects to the previous results, but the discussion was not very in-depth. Fig. 12f, suggests that the Fbiodis fluxes are strongly overestimated. Here it would be interesting to know whether this is due to the randomly altered 14C signature of heterotrophic respiration or to the very different fluxes of VRPM and ORCHIDEE.*

We added Figure 14 to show the influence of each category on the total prior $\Delta^{14}CO_2$ content. The main reason for this overestimation indeed originated from the prior $F_{biodis}$, followed by the prior $F_{nuc}$, which we found to have a large impact on stations surrounded by nuclear facilities such as SAC.

*Section 5: Discussion*

*The discussion section is more like a summary and a comparison with previous literature. Unfortunately, there is no real in-depth discussion of the results.*

We updated the Discussion section including the referee's concern about the assumption of perfect transport, perfect boundary condition, prior and posterior uncertainties, spatial distribution, and the impact of the prior terrestrial isotopic disequilibrium product.

*L508: Fig10 -> Fig9, Fig9 -> Fig 10 , Fig 11-> ?*

We updated the references to the figures.

*L520: How will such a scaling workaround solve the problem of small signals? This approach does not change the measurement uncertainty or the observational signal-to-noise ratio. The authors mention that this approach might be problematic due to noise. Thus, I recommend not making this suggestion at all.*

We removed this suggestion from the manuscript and updated the text in L586-592 of the revised manuscript.

sions in an OSSE setup given the current $CO_2$ and $\Delta^{14}CO_2$ observation networks.  We think that a more likely explanation is the difference in the magnitude of the fossil $CO_2$ emissions in  this region against the natural fluxes  . This can be seen by the differences in the seasonal amplitude of the fluxes   , which includes the fluxes transported  . In Western/Central Europe $F_{bio}$ and $F_{ff}$ are of a similar order of magnitude (2.81 TgC day$^{-1}$ for $F_{bio}$ and 0.6 TgC day$^{-1}$ for $F_{ff}$) (see Figures 5 and 6). In contrast, in Northern Europe, there is a tenfold difference in the seasonal amplitude of the two fluxes: 2.44 TgC day$^{-1}$ for $F_{bio}$ and 0.06 TgC day$^{-1}$ for $F_{ff}$.

In addition, the prior uncertainty for $F_{ff}$ (0.002 PgC year$^{-1}$) in this region is much lower compared to $F_{bio}$ (0.12 PgC year$^{-1}$) in Northern Europe.

*L524: The realistic… -> The more realistic…*

We modified this in the text.

*Appendix A:*

*Table A1: The numbers for the CO2 Obs Error are unrealistic (see above).*

We commented on this in the answers above.

---

## Author Response (AR2)

Lund, June 3$^{rd}$, 2024

**Carlos Gómez-Ortiz**
**Department of Physical Geography and Ecosystem Science**
**Lund University**
**Sweden**

**Jens-Uwe Grooß**
**Editor assigned to Research article EGUSPHERE-2023-2215.**
**Atmospheric Chemistry and Physics (ACP)**

Dear Editor,

We appreciate the detailed feedback provided by the referee on our manuscript titled "Can $\Delta^{14}CO_2$ observations help atmospheric inversions constrain the fossil $CO_2$ emission budget of Europe?". While their comments have been instrumental in guiding our revisions, we would like to address two key points where we believe there may have been misunderstandings or misinterpretations of our revised manuscript:

1.  The referee has repeatedly requested a mathematical description and justification for Appendix B. This appendix describes the procedure used to define the reduced grid, represented in Figure 2b. In our setup, the tracer transport is computed at a spatial resolution of 0.5°, but the emissions are resolved on a coarser grid towards the edges of the domain, where there are fewer observations to constrain the emissions. This is clearly represented in Figure 2 and described in Section 2.3.1 (lines 228 to 242 of the revised manuscript). While we welcome critical comments on this choice, it is important to note that it is a setting rather than a derivation. For practical reasons, it is defined through an iterative algorithm, but it is fundamentally an arbitrary choice, similar to the definition of the domain extent or the selection of a specific resolution for the transport model. As such, it is not feasible to provide a mathematical description for this more than the actual code referenced in the answer below.
2.  The referee has criticized the lack of methodological description regarding the construction of the error covariance matrix. However, this information is clearly presented in Section 2.3.2 since the original version of the manuscript. Moreover, our inversion setup follows standard practices. While there may be aspects of this methodology that can be improved, our study's focus is not on these specific improvements. Therefore, we believe it is appropriate to refer to other studies for detailed methodologies on this topic.

Below, we provide a detailed response (in regular font) to each of the referee's comments *(in italics)*, indicating how we have addressed them in the revised manuscript. We hope this clarifies any misunderstandings and demonstrates our commitment to meeting the high standards of the journal.

**Referee's comments**

*The revised manuscript "Can $\Delta^{14}CO_2$ observations help atmospheric inversions constrain the fossil $CO_2$ emission budget of Europe?" by Carlos Gómez-Ortiz, Guillaume Monteil, Sourish Basu,*

*and Marko Scholze, provides a clear improvement of the original manuscript. This applies mainly for the part of the paper, which presents the results. The presentation of the methodology has made little improvements, however. The reviewers' guide on the clearness of the method emphasizes some essential questions about the methodological aspects. I cite from the reviewers' guide on the clearness of the method:*
*• Are mathematical formulae, symbols, abbreviations, and units correctly defined and used?*
*• Are the scientific methods and assumptions valid and clearly outlined?*
*• Are the results sufficient to support the interpretations and conclusions?*
*• Is the description of experiments and calculations sufficiently complete and precise to allow for their reproduction by fellow scientists (traceability of results).*

*The authors must explain the objectives of this paper more clearly. Simply stating that the aim is "to explore the interest of using these $CO_2$ and $\Delta^{14}CO_2$ observations to constrain the fossil $CO_2$ emissions in Europe" appears not to be a sufficiently strong motivation for performing this study and publishing it in ACP.*

We believe our manuscript fits very well to the scope of ACP, in particular to the subject area 'Gases' and to the research activity 'Atmospheric Modelling and Data Analysis'. This submission is in line with previous articles submitted to and published by ACP such as: "Potential of 14C-based versus $\Delta$CO-based $\Delta ffCO_2$ observations to estimate urban fossil fuel $CO_2$ ($ffCO_2$) emissions" (https://doi.org/10.5194/egusphere-2023-1239), "Atmospheric radiocarbon measurements to quantify $CO_2$ emissions in the UK from 2014 to 2015" (https://doi.org/10.5194/acp-19-14057-2019), and "Potential of European $^{14}CO_2$ observation network to estimate the fossil fuel $CO_2$ emissions via atmospheric inversions" (https://doi.org/10.5194/acp-18-4229-2018). Therefore, we consider ACP as suitable and relevant for publishing our manuscript.

To further clarify the objective and the relevance of our study, we have expanded this section of the Introduction as follows:

"In this work, we present the new capabilities of the Lund University Modular Inversion Algorithm (LUMIA) system (Monteil and Scholze, 2021) to perform simultaneous inversions of atmospheric $CO_2$ and $\Delta^{14}CO_2$ observations as a first attempt to develop a model capable of supporting the monitoring and verification of fossil $CO_2$ emissions across Europe. Such emissions monitoring and verification support capacities are essential for assessing compliance with international agreements, such as the Paris Agreement (UNFCCC, 2015), and for guiding policy decisions aimed at reducing carbon emissions as outlined by Janssen-Maenhout et al. (2020). We perform Observing System Simulation Experiments (OSSEs), recreating the current state of the ICOS network and its sampling strategy, and using different flux products (as priors and true values) to demonstrate the performance of the inversion scheme and show its capabilities. We begin by assessing the impact of oceanic fluxes on the total $CO_2$ and $\Delta^{14}CO_2$ concentrations. Then, we evaluate the impact of adding $\Delta^{14}CO_2$ observations on the estimation of fossil $CO_2$ emissions by comparing the model's ability to recover true fluxes starting from a prior flux set to zero. Finally, with a more realistic setup, i.e., prior, we evaluate the impact of the prescribed fossil $CO_2$ flux uncertainty and the impact of the terrestrial isotopic disequilibrium product."

*The reader is entitled to a more detailed reference to the underlying LUMIA model in the model description subsection 2.1. There is no such reference here. For example, there is no outline of the Lagrangian approach embedded in the TM5 global model, let alone further reasonable specifications, suitable to acknowledge a vetted methodology.*

There is no mention of a "Lagrangian approach embedded in the TM5 global model" in our manuscript. The only reference to the TM5 model appears in the discussion. We agree that the title of Section 2.1 could have been clearer and have slightly modified it. However, it is intentional that the transport model is not described in detail here, as the approach remains consistent whether the transport model is Lagrangian, Eulerian, global, or regional. The underlying transport model used in our study is FLEXPART, a widely used mesoscale Lagrangian transport model. This is described (and referenced) in Section 3.2, and a pointer to Section 3.2 is given in Section 2.1 (line 163). The specific implementation has been extensively described in Monteil & Scholze (2021), and it is standard practice to refer to the original model development paper rather than repeating the information.

To address the reviewer's concerns and improve clarity, we have modified the initial paragraph of Section 2.1 as follows:

"We are using the LUMIA (Lund University Modular Inversion Algorithm) system as described by Monteil and Scholze (2021), modifying the way background concentrations ($y^b$ in Equation 1) are calculated by computing a smoothed and detrended average of real observations from the ICOS network for each sampling site. Originally, the LUMIA system was developed to optimize regional Net Ecosystem Exchange (NEE) fluxes over Europe using in situ $CO_2$ observations from the ICOS (Integrated Carbon Observation System) Atmosphere network. In this study, we have extended LUMIA to additionally assimilate $\Delta^{14}CO_2$ observations from the same network and optimize multiple flux categories. This extension introduces a new step in the mass balance of the atmospheric transport, as follows:

$$y_{CO_2} = y^b_{CO_2} + \sum_c H(F_c)$$

$$y_{C\Delta^{14}C} = y^b_{C\Delta^{14}C} + \sum_c H(\Delta_c F_c)$$

"

*This point was not addressed in my line 120 remark. Rather, only my "shifted delta" item was questioned, which simply means that the delta sign, typically located ahead of the flux F, has been displaced behind F (in the difference parentheses). In the writing of formulae, the authors do not differentiate between scalars, vectors, and operators by notation (normal vs. bold face, small vs, capital letters), which should be standard in mathematically oriented publications.*

The radiocarbon signature $\Delta$ is a vector, and its position on either side of the flux term does not affect the result of the multiplication. However, to ensure consistency in the formula, we have rewritten Equation 2b by moving the disequilibrium subtraction in front of $F$ as follows:

$$\sum_c H(\boldsymbol{F}_c) = H(\boldsymbol{\Delta}_{\text{ff}}\boldsymbol{F}_{\text{ff}}) + H\big(\boldsymbol{\Delta}_{\text{atm}}(\boldsymbol{F}_{\text{bio}} + \boldsymbol{F}_{\text{oce}})\big) + H\big((\boldsymbol{\Delta}_{\text{bio}} - \boldsymbol{\Delta}_{\text{atm}})\boldsymbol{F}_{\text{bio2atm}}\big)$$
$$+ H\big((\boldsymbol{\Delta}_{\text{oce}} - \boldsymbol{\Delta}_{\text{atm}})\boldsymbol{F}_{\text{oce2atm}}\big) + H(\boldsymbol{\Delta}_{\text{nuc}}\boldsymbol{F}_{\text{nuc}})$$

All the manuscript's equations were carefully revised and adjusted when needed using the appropriate notation following Table 1 from Fundamentals of data assimilation (Rayner et al., 2019).

*Original L172 : In examining the space-time covariance matrix as understood by the authors, I consulted the Broquet et al. (2011) paper, as referenced in the reply. However, I could not discern any indication of the use of this spatio-temporal matrix formulation, as claimed. I would be interested in understanding the design of such a space-time matrix, which is used in data assimilation by the rarely applied Physical Space Statistical Analysis System approach (PSAS). Please provide precise information on the matrix entries Bij and the methodology used to obtain them. The repeated references to other studies, such as Basu et al., are insufficient and do not comply with the standards set by ACP and other quality journals (see above).*

The construction of the matrix **B** is thoroughly detailed in Section 2.3.2, titled "Construction of the prior error covariance matrix (**B**)." We believe this section provides a clear, step-by-step explanation. Specifically, the entries of the prior error covariance matrix **B** are described by the equation found in Section 2.3.2 (L251 of the revised manuscript, originally L204).

*Original L192…: The additional description, including matrix dimensions, has been noted. However, I was unable to reconcile the clustering approach detailed in Appendix B with equation (7).*

We understand the referee's concern. However, it is important to clarify that Equation 7 and Appendix B address different aspects of the methodology (see also our general response to the review):
  - Equation 7 describes the transition between the model grid and the optimization grid. This is a step within the inversion process.
  - Appendix B explains how the optimization grid is defined, specifically how matrix $\mathbf{T}_h$ in Equation 7 is constructed. This is part of the experiment design and serves as an input to the inversion, rather than a step within it.

*It is evident that Appx B is purely descriptive, devoid of any formulas or derivations that could potentially relate to (7). It appears that the authors have derived a sensitivity analysis through the use of adjoint (backward in time) integration, which has enabled them to identify clusters of emission cells.*

We would like to clarify that we have not identified clusters of emissions. Instead, we have used a clustering approach to define the extent (in space) of the emission offsets optimized by the inversions. These "clusters" are based on the adjoint sensitivity of the network to the emissions (dE/dy), which is entirely independent of the emissions themselves. In other words, the grid is denser where there are more observations, as clearly illustrated in Figure 2. However, the specific approach used to define this reduced grid is not very relevant from the perspective of the inversion, therefore it is described in an appendix and not in the main text.

*In order to highlight the level of mathematical rigor that is evident in the authors' approach, it is worth noting the following literature, which features methods for achieving resembling objectives. Berliner, L. M., Lu, Z., and Snyder, C.: Statistical design for Adaptive Weather Observations, J. Atmos. Sci., 56, 2536–2552, 1998. ** Bishop, C. H. and Toth, Z.: Ensemble Transformation and Adaptive Observations, J. Atmos. Sci., 56, 1748–1765, 1998. ** Buizza, R. and Palmer, T. N.: The Singular-Vector Structure of the Atmospheric Global Circulation, J. Atmos. Sci., 52, 1434–1456, 1993. ** Goris, N. and Elbern, H.: Singular vector-based targeted observations of chemical constituents: description and first application of the EURAD-IM-SVA v1.0, Geosci. Model Dev., 8, 3929–3945, https://doi.org/10.5194/gmd-8-3929-2015, 2015.*

Thank you for highlighting these references. However, the objective in our case is more straightforward than the reviewer seems to think. Our goal is to construct a grid that has a denser resolution where observational coverage is better. This allows us to work with a smaller control vector and avoids the impracticality of solving for emission offsets at a very high resolution where there are no observations. The fine-scale structure of the emissions is maintained regardless.

*Original L202: In response to the authors' initial request, I am unable to comprehend the rationale provided. A more rigorous mathematical approach would be beneficial. It is recommended that the authors provide a clear and detailed outline of the mathematical methods employed, in accordance with the criteria set out by the journal.*

We do not understand the referee's comment here. The mathematical method is totally standard, and is entirely described in this section. Lines 202-208 describe how each element $\mathbf{B}_{ij}$ of the covariance matrix is constructed, in a generic way (i.e. with covariances based on the product of variances and distance-based correlation functions), while lines 209-217 describe how the elements of that product (the variances and correlations) are computed, for each category. All the equations that are used in the code are already present in the paper.

*Original L282: It is unclear whether the authors have fully grasped the distinction between OSSEs and identical twin experiments. To illustrate, if meteorological simulations (wind transport) are conducted using the same model configuration for observational modelling and inversion procedures, the latter is provided. In such a case, the assimilation system may be considered to "err on the optimistic side", as outlined in the textbook by Daley (1991): Atmospheric Data Analysis, due to tacitly assuming the transport simulation as perfect. The authors are encouraged to subsume their approach in terms of this aspect.*

*Original L282: I don't know whether the authors have fully understood the difference between OSSEs and identical twin experiments. For example, if the meteo simulations (wind transport) are performed by the same model setup for observation modelling and the inversion procedure, then the latter is given, in which case the assimilation system "errs on the optimistic side" (see textbook Daley, 1991, Atmospheric data analysis).*

We acknowledge that, in strict meteorological terminology, our approach aligns more closely with identical twin experiments rather than OSSEs. As the referee correctly point out, a 'real' OSSE would typically involve using a different transport model for generating pseudo-observations to avoid the assumption of perfect transport simulation. However, it is important to note that within

the atmospheric inversion community, the term "OSSE" is commonly used to describe experiments like ours, even when the same model configuration is employed for both observational modeling and inversion procedures. This usage is prevalent in the literature and aligns with the established terminology in our field.

To illustrate this, we refer to studies where identical twin experiments are referred to as OSSEs, such as in Philip et al. (2019), Byrne et al. (2019), Liu et al. (2014), Chen et al. (2023), and Wang et al. (2014). These references demonstrate the conventional use of the term "OSSE" in the context of atmospheric inversions. Therefore, while we understand and acknowledge the meteorological distinction, we rather call it "perfect transport OSSEs" in our manuscript to remain consistent with the common terminology in atmospheric inversion studies.

*Original L480: It would appear that the authors' response indicates a fundamental point of confusion. Firstly, it is unclear whether the Chi-squared value is set to approach ½ or 1. In practice, it is more likely to be set to ½, as exemplified by Talagrand. The successful approximation to the target value is contingent upon a consistent design of R and B, which may not be readily apparent even through educated guesses of these matrices. It is of the utmost importance to "tune" the covariances and validate the minimisation process in order to draw valid conclusions from a consistently designed OSSE. For further details, please refer to Desroziers et al. (2005). Furthermore, the remarks made in relation to original line 282 are equally applicable here.*
*I am unable to understand the explanation of the authors to my initial request. Substatially more mathematical rigour will help. The authors are strongly recommended to clearly outline the mathematical methods, following the journals' criteria mentioned above.*

*Original L480: Apparently, the authors' answer indicates a fundamental point of confusion: Firstly, is the Chi2 value set to ideally approaching at ½ or at 1 (mostly the former, e.g. Talagrand)? The successful approximation to the target value requires (only as necessary, not sufficient condition) a consistent design of R and B, not necessarily given even by educated guesses of these matrices. "Tuning" the covariances and validation of the minimization is essential to extract conclusions from a consistently designed OSSEs. See Desroziers et al. 2005 for details. In addition, the remarks to original line 282 apply here as well.*

We calculate the reduced chi-square statistic as a diagnostic to ensure that we have improved upon the initial state, which is clearly supported by the change from the prior to posterior values (1.77 to 1.06), and as a way to guarantee that we are not under or overfitting the model's optimization. We calculated the $\chi_\nu^2$ as:

$$\chi_\nu^2 = \frac{1}{\nu} \sum_{i=1}^{N} \left( \frac{y_i^{so} - y_i^{b,a}}{\epsilon_i} \right)^2$$

Where $y_i^{so}$ is the synthetic observation $i$, $y_i^{b,a}$ is either the prior (b) or the posterior (a) concentration $i$, $\epsilon_i$ is the error of the synthetic observation $i$, N is the number of observations, and $\nu$ are the degrees of freedom calculated as $\nu = N - p$, being p the number of fitted parameters in the model. Since p is difficult to calculate due to the different time and space clusters, we keep the number of observations as the degrees of freedom ($\nu = N$). We added this explanation to the manuscript to make it clearer.

We acknowledge the importance of a consistent design of the covariance matrices **R** and **B**, as we discussed in our methodology, however, we would like to emphasize that our current study represents a first demonstrator of the inversion scheme rather than an operational system. The level of fine-tuning expected for an operational system, as highlighted in the referee's comments, is indeed a goal for future work. At this stage, our primary aim is to demonstrate the feasibility and potential of our approach, which naturally involves some initial assumptions and approximations such as the perfect transport and boundary conditions and it does not make sense to fine tune on the current approach.

*Appendix B:*
*The authors seem reluctant to respond positively to suggestions for a more comprehensive and rigorous presentation of the mathematical foundations. In particular, Appendix B is inadequate in its purely verbal presentation. I find an implementation hardly reproducible.*

As stated above, there is no mathematical foundation for the clustering process. It is just a practical way to construct the "reduced grid" represented in Figure 2b. The code is provided for reproducibility (along with the entire code used in this project), the role of Appendix B is to explain to an interested reader how that code works. However, it is not a central part of the methodology (therefore it is in an Appendix). The clustering algorithm can be found in the code available as supplementary material (https://doi.org/10.5281/zenodo.8426217) on lumia/Tools/optimization_tools.py or in the GitHub repository (https://github.com/lumia-dev/lumia/blob/lumia_multitracer/lumia/Tools/optimization_tools.py).

**References**

Byrne, B., Jones, D. B. A., Strong, K., Polavarapu, S. M., Harper, A. B., Baker, D. F., & Maksyutov, S. (2019). On what scales can GOSAT flux inversions constrain anomalies in terrestrial ecosystems? *Atmospheric Chemistry and Physics*, *19*(20), 13017–13035. https://doi.org/10.5194/ACP-19-13017-2019

Chen, H. W., Zhang, F., Lauvaux, T., Scholze, M., Davis, K. J., & Alley, R. B. (2023). Regional CO2 Inversion Through Ensemble-Based Simultaneous State and Parameter Estimation: TRACE Framework and Controlled Experiments. *Journal of Advances in Modeling Earth Systems*, *15*(3), e2022MS003208. https://doi.org/10.1029/2022MS003208

Liu, J., Bowman, K. W., Lee, M., Henze, D. K., Bousserez, N., Brix, H., Collatz, G. J., Menemenlis, D., Ott, L., Pawson, S., Jones, D., & Nassar, R. (2014). Carbon monitoring system flux estimation and attribution: impact of ACOS-GOSAT XCO2 sampling on the inference of terrestrial biospheric sources and sinks. *Tellus, Series B: Chemical and Physical Meteorology*, *66*(1). https://doi.org/10.3402/TELLUSB.V66.22486

Monteil, G., & Scholze, M. (2021). Regional CO2 inversions with LUMIA, the Lund University Modular Inversion Algorithm, v1.0. *Geosci. Model Dev.*, *14*(6), 3383–3406. https://doi.org/10.5194/gmd-14-3383-2021

Philip, S., Johnson, M. S., Potter, C., Genovesse, V., Baker, D. F., Haynes, K. D., Henze, D. K., Liu, J., & Poulter, B. (2019). Prior biosphere model impact on global terrestrial CO2 fluxes estimated

from OCO-2 retrievals. *Atmospheric Chemistry and Physics*, *19*(20), 13267–13287. https://doi.org/10.5194/ACP-19-13267-2019

Rayner, P. J., Michalak, A. M., & Chevallier, F. (2019). Fundamentals of data assimilation applied to biogeochemistry. *Atmos. Chem. Phys.*, *19*(22), 13911–13932. https://doi.org/10.5194/acp-19-13911-2019

Rödenbeck, C., Gerbig, C., Trusilova, K., & Heimann, M. (2009). A two-step scheme for high-resolution regional atmospheric trace gas inversions based on independent models. *Atmos. Chem. Phys.*, *9*(14), 5331–5342. https://doi.org/10.5194/acp-9-5331-2009

Wang, J. S., Kawa, S. R., Eluszkiewicz, J., Baker, D. F., Mountain, M., Henderson, J., Nehrkorn, T., & Zaccheo, T. S. (2014). A regional CO2 observing system simulation experiment for the ASCENDS satellite mission. *Atmospheric Chemistry and Physics*, *14*(23), 12897–12914. https://doi.org/10.5194/ACP-14-12897-2014

---

## Author Response (AR3)

Lund, July 19th, 2024

**Carlos Gómez-Ortiz**
**Department of Physical Geography and Ecosystem Science**
**Lund University**
**Sweden**

**Jens-Uwe Grooß**
**Editor assigned to Research article EGUSPHERE-2023-2215.**
**Atmospheric Chemistry and Physics (ACP)**

Dear Editor,

Here we address the latest review of our manuscript titled "Can Δ14CO2 observations help atmospheric inversions constrain the fossil CO2 emission budget of Europe?"

We sincerely appreciate feedback on our manuscript and are committed to improve it further. But in this round of review, we are partly facing again questions that have already been answered and supported by relevant bibliography in previous rounds as well as new concerns that have not been mentioned before. While we have made modifications based on the latest comments to clarify the manuscript for the referee, we consider these new comments rather to be minor revisions. However, we are concerned that this process may become an endless loop of revisions, as there seems to be a fundamental misunderstanding of the concept of our study by the referee. This may be caused by preconceived conceptions of their field of research from the assessment of our manuscript (examples for this are the confusion about the grid, the usage of the term 'Observation Simulation System Experiment', and the comment on basic linear algebra as explained in more detail below). We have also noticed that the referee has progressively lowered the rating of the manuscript in terms of scientific significance, scientific quality, and presentation quality, despite acknowledging improvements made in each round of review. Therefore, we suggest that a direct contact with the referee may help to clarify their concerns.

We are puzzled about the referee's question on basic linear algebra concepts, such as "However, it is unclear how $B^{-1}$ and x multiply in (6),". As mentioned above we assume that this is caused by a fundamental misunderstanding of our manuscript. This misunderstanding appears to arise from an earlier stage of the manuscript, leading to questions about basic concepts. We cannot adequately address the referee's concerns without knowing the specific source of this confusion. It seems the referee believes our study involves a more complex methodology than it actually does. Our manuscript describes a classical sensitivity experiment in a variational inverse modeling application, using standard optimization algorithms common in the field and a general methodology described in many publications and also more specifically already elsewhere (e.g. Monteil and Scholze, 2021). This is not a model development manuscript and does not employ any unique optimization techniques.

Below, we provide a detailed response (in regular font) to each of the referee's comments (in italics), indicating how we have addressed them in the revised manuscript. We hope this

clarifies any misunderstandings and demonstrates our commitment to meeting the high standards of the journal.

**Referee's comments**

*The updated manuscript contains a number of improvements. However, I also see a number of points where the authors are reluctant to follow my recommendations.*
*I summarise the main points below:*
*comments on the original communication to the editor:*
*1. "…but it is fundamentally an arbitrary choice, similar to the definition of the domain extent or the selection of a specific resolution for the transport model. As such, it is not feasible to provide a mathematical description for this more than the actual code referenced in the answer below."*
*As such, it is not feasible to provide a mathematical description for it beyond the actual code referenced in the response below".*
*The grid design choices made by the authors may well be reasonable, but they are not arbitrary per se, as the model is expected to produce valid results consistent with the objective of the paper.*

Our sentence in the rebuttal directly addressed the referee's request for a "mathematical foundation for Appendix B" (as noted in the last line of their previous review). Appendix B outlines the procedure for defining the reduced grid on which the control vector is based. It is important to note that this is distinct from the transport model grid, which was mentioned in the manuscript (lines 241-245). The choice for the control vector grid is indeed arbitrary, based on the authors' best judgment rather than a formal derivation. We justified this choice in lines 234-240 of the manuscript, a justification to which the referee has not previously objected.

*I understand that a formal study of model optimisation is beyond the scope of the paper. However, scientific scrutiny deserves at least that the model grid design (extension, resolution) and assumed emission patterns are based on a discussion of typical advection timescales and minimum error estimates for emission source assumptions, summarising the underlying assumptions and assuring the reader of the limits of possible modelling errors. I do not understand why this moderate request for more concern and awareness of possible design problems is apparently seen as an imposition by the authors. One sentence should suffice.*

The referee's comment suggests to us that they misunderstand our grid design. All previous questions were about the optimization grid shown in Figure 2, which is not at all related to advection timescales. Our transport calculations are done at a 0.5° spatial and hourly temporal resolution, and emissions are provided at the same resolution. The optimization grid is used only to solve the inverse problem (i.e., the control vector), and advection timescales are not relevant.

This new request for justification of resolution and domain definition was not raised before. The domain choice has been commented since the first version of the manuscript at the beginning of Section 2.1. Regional transport model (Section 2.2. Observations, L189-190 of the last version) and supported by publications such as Monteil et al., 2020; and Thompson et al., 2020 (cited on the manuscript) and other ACP and Copernicus publications such as McGrath et al., 2023; Munassar et al., 2023; and Petrescu et al., 2021. We have already explained and justified our

choices for the optimization grid in the manuscript. We think that our current explanation is sufficient and appropriate for the scope of this manuscript (as it was in the previous publications mentioned before).

*2. The algorithmic description is still a mystery to me. In some exemplary detail: The authors have (L222): "H is the observation operator, which includes the transport model…". Since, according to eq. (6), H (including the time propagating transport model) acts on the vector x to be optimised, the latter presumably according to the Ide formulation.*

Indeed, our formulation was somewhat confusing and now we have changed our equations 1 to 7 to account for that. In equations 1-4 we have introduced the regional transport model operator $K$. Equations 5-7 have been reformulated to clarify the linearity of the observation operator $H$.

*3. The H M(tangent linear model)(t_i, t_0) acts on x(t=0), which is not a multi-time step state vector but rather an observation operator H. The background term, with the inverse space-time covariance matrix B (as previously stated in 2.3.2), would be in compliance with this if the generalized observation operator H were also applied. However, this is absent, and there is no indication of how the time propagation is managed for this 1. rhs term of (6).*

We adjusted the cost function (Equation (6)) to write it in the common form used by other authors, and removed the sentence in L219-220 of the latest version "that are available over a range of times, also known assimilation window" which might be misunderstood as that we do sequential assimilation, but otherwise, we do not understand the referee's concern. We do not really understand the comment of the reviewer. There is no tangent linear model in our case, since **H** is linear. **H** acts on $x$ in its entirety, not on $x(t)$. It's not clear to us what the referee means by "time propagation": there is a time propagation within **H** (i.e. the observations at time $t = t1$ are influenced by control variables $x(t \leq t1)$, but there is no time propagation of the solution (i.e. $x(t = t1)$ doesn't depend on $x(t < t1)$).

*Furthermore, the presentation of the covariance formulation is not sufficiently clear.*
*L 265: cov(x1, x2) = σx1 σx2 exp(−(d(p1,p2)/Lh)2exp(−|t2−t1|/Lt)*
*This expression should be indexed properly, that is, with time and location indices running independently, and both by different index variables with associated limits. This would ensure that the reader can identify the entry-wise construction and associated x.*

The equation is strictly correct, and exactly how it's used in the code. The covariance between two elements $x_1$ and $x_1$ of the control vector is the product of their variances ($\sigma x_1$ and $\sigma x_2$), and of two decorrelation functions, decreasing exponentially with the spatial and temporal distance between the two points. There is zero ambiguity in the Equation. Nevertheless, we replaced the sub-indexes 1 and 2 in the equation by $i$ and $j$ to put it in a more general way. The indices refer neither to time nor to location, they refer to the position of the variables in the control vector. $x_i$ has coordinates $(p_i, t_i)$, and $x_j$ has coordinates $(p_j, t_j)$.

The Equation at L265 gives exactly the entry-wise construction of the covariance matrix. The only step missing in this equation is for the case when $x_i$ and $x_j$ belong to different emission categories, in which case the correlation is 0. But this is very clearly specified just above, at L252: "we assume no correlations between different categories and different tracers".

*However, it is unclear how B^-1 and x multiply in (6).*

1. $x$ has $n = n_t^{opt} \times n_p^{opt} \times n_{cat}$
2. $\mathbf{B}$ has $(n, n)$ elements
3. The product $x^T \mathbf{B}^{-1}$ results in a $n$ dimension vector

*Section 2.3.2 (Covariance construction-related issues)*
*In their response, the authors assert that: "It is our contention that this section provides a lucid, step-by-step explication. In particular, the entries of the prior error covariance matrix B are defined by the equation presented in Section 2.3.2 (L251 of the revised manuscript, originally L204). I must respectfully disagree. I endeavoured to comprehend the material, but upon reaching a point where I believed I had succeeded, I encountered inconsistencies in another location. It is unclear why a section is not devoted to defining the subspace section of vector x (sub)segment wise.*

We are puzzled why the referee is misquoting us here. Their paraphrasing suggests a very different tone than we actually used in our previous response. The correct quote is: "We believe this section provides a clear, step-by-step explanation. Specifically, the entries of the prior error covariance matrix $\mathbf{B}$ are described by the equation found in Section 2.3.2 (L251 of the revised manuscript, originally L204)."

The vector $x$ can be divided in one section for each tracer/category. The approach for each of these sections is the same and is described in L253 ("the sections of $\mathbf{B}$ specific to each tracer/category ...") to L261. The part specific to the subsections is described just after, from L264 to L270. There is no ambiguity in our description of the covariance matrix.

*Firstly, it is assumed that the term "offset" refers to what is commonly referred to in data assimilation as an "increment," which is likely an analysis increment rather than an observation increment.*

No. The term "offset" refers to the fact that the control vector contains "offsets" to the prior emissions, as opposed to e.g. scaling factors, which are another commonly encountered approach. It refers to the physical quantities that the control vector represents, whereas "increment" refers to the change of the values of the control vector between two iterations of the inversion (regardless of what physical values this "increment" corresponds to).

The term "analysis" used by the referee is commonly used in the context of sequential inversions, such as those employed for weather forecasting, which estimate the atmospheric state through a series of forecast and analysis steps. However, this term is not adapted to atmospheric flux inversions, where the terms "prior" and "posterior" are to be used instead. Beyond the naming convention, this also highlights that there is no time propagation of the solution.

*L250: : "The matrix Xc x is the portion of the control vector x that contains offsets for the category c, reshaped as a (ntopt, npopt) matrix, with ntopt and npopt the number of optimized (weekly) intervals and grid-cell clusters, respectively."*
*In what sense may a matrix X, comprising portions of a vector, be defined in terms of vector calculus? It is possible that this is the source of the misunderstanding.*

- $\mathbf{T}_T$ is a $(n_t^{opt}, n_p^{opt})$ matrix.
- $x_c$ is a $(n_t^{opt} \times n_p^{opt})$ vector, reshaped as a $(n_t^{opt}, n_p^{opt})$ matrix $\mathbf{X}_c$.
- $\mathbf{T}_H$ is a $(n_t^{opt}, n_p^{opt})$ matrix.

The product $d\mathbf{F}_c = \mathbf{T}_T \mathbf{X}_c \mathbf{T}_H$ gives a $(n_t^{opt}, n_p^{opt})$ matrix (i.e. 1 year of hourly emission offsets at a 0.5° resolution, for one category $c$). The corresponding emissions are just $\mathbf{F}_c = \mathbf{F}_c^0 + d\mathbf{F}_c$, with $\mathbf{F}_c^0$ the prior emissions for that category (still at a 0.5° and hourly resolution).

The adjoint is simply $\mathbf{X}_c^{adj} = \mathbf{T}_H^T d\mathbf{F}_c^{adj} \mathbf{T}_T^T$, which aggregates a $(n_t^{opt}, n_p^{opt})$ adjoint emission array (for category $c$) onto a $(n_t^{mod}, n_p^{mod})$. The $\mathbf{T}_T$ and $\mathbf{T}_H$ matrices are category specific. These are basic aggregation/rebinning operations.

The rationale behind these equations is:
- The transport is computed hourly (with hourly emissions), but the inversions solves for weekly offsets (i.e. the emissions within a week will all be increased or decreased by the same absolute amount). An operator is needed to project these weekly offsets onto the hourly emissions.
- Likewise, in the spatial dimension, the offsets are defined on a reduced grid (Figure 2), whereas the emissions themselves are defined on a 0.5° resolution, the aforementioned operator also needs to be extended to project the emissions from the reduced grid to the model grid.

We have modified Eq. 7 and the section related to it such that we do not use $\mathbf{X}$ in the description. We think that our revised version is easier to follow.

*While for partial differential equations, discretized physical domains are transferred to vectors and back to harness vector calculus, the rationale behind the procedure addressed here is unclear.*

There are no partial differential equations involved in this procedure, neither are they mentioned in the text or the equations. The rationale is just that at the transport model resolution, there are 42 million spatiotemporal grid cells (24 hours * 365 days * 80 lon * 60 lat), for each emission category, which drops to ~130000 per category with our reduced grid, which is computationally more favorable. Also, the density of the observation network is in any case way too low to robustly resolve the emissions at the model resolution.

Note, once again, that this reduced resolution concerns only the offsets optimized by the inversions. The base emissions, onto which these offsets are added, are still provided and transported at high resolution.

*In addition to the ACP paper writing guidelines, which have been partly replicated in my previous review, it is important to ensure that mathematical formulae, symbols, abbreviations and units are correctly defined and used. • Are the scientific methods and assumptions valid and clearly outlined? It is not possible to discern an enhanced derivation from the generic variational formula (6) in comparison to eq. (7), and furthermore, the verbal description of the covariance matrix B in 2.3.2 is not sufficiently clear.*

We answer this in previous questions above. Regarding the verbal description of **B**, we disagree. The description is perfectly clear and unambiguous. As mentioned before, we suspect that there is a more fundamental misunderstanding, which lead the referee to misinterpret this section. We don't know precisely how to solve this misunderstanding without knowing its origin, but it is not in this section.

*Please refer to Section 3. The confusion surrounding the use of the term "OSSE" and its relation to identical twin (IT) experiments*
*Sect 3. The confusion surrounding the use of the acronym "OSSE" in relation to identical twin (IT) experiments: Although the authors acknowledge that their approach is correctly classified as an IT, their justification by customary use in their atmospheric inversion community is not valid. The authors might not make false use of a terminology with precise distinctions between (OSSE – IT) from data assimilation without any necessity, neglecting precision.*
*The authors may not be employing the correct terminology with regard to the precise distinctions between OSSE and IT in the context of data assimilation, without any necessity for such precision.*

We have already addressed this question in the past and we are surprised by the invalidation of our terminology, especially given the established practices in our research community. Numerous papers within our community (also published in ACP and already cited in previous responses) support the usage of the terminology we have employed. It is important to recognize that different research communities may develop distinct terminologies and methodologies, even if the underlying principles are similar. This divergence in terminology can cause confusion to the referee, but it is consistent with the practices in our field of atmospheric inverse modeling.

Our intention was not to neglect precision but to adhere to the accepted conventions within our community. We hope this explanation clarifies our approach and the reasoning behind our terminology.

We have also clearly stated in the manuscript that we are performing a 'perfect transport Observing System Simulation Experiment' (L 282).

**Note:** Given the questions raised by the referee about the rigorousness of the mathematical formulations, in the attached document we provide a detailed derivation of all the equations in the methodological section of the manuscript. However, we consider such level of detail is too detailed for the paper.

**References**

McGrath, M. J., Petrescu, A. M. R., Peylin, P., Andrew, R. M., Matthews, B., Dentener, F., Balkovič, J., Bastrikov, V., Becker, M., Broquet, G., Ciais, P., Fortems-Cheiney, A., Ganzenmüller, R., Grassi, G., Harris, I., Jones, M., Knauer, J., Kuhnert, M., Monteil, G., … Walther, S. (2023). The consolidated European synthesis of CO2 emissions and removals for the European Union and United Kingdom: 1990-2020. *Earth System Science Data*, *15*(10), 4295–4370. https://doi.org/10.5194/ESSD-15-4295-2023

Monteil, G., Broquet, G., Scholze, M., Lang, M., Karstens, U., Gerbig, C., Koch, F.-T., Smith, N. E., Thompson, R. L., Luijkx, I. T., White, E., Meesters, A., Ciais, P., Ganesan, A. L., Manning, A., Mischurow, M., Peters, W., Peylin, P., Tarniewicz, J., … Walton, E. M. (2020). The regional European atmospheric transport inversion comparison, EUROCOM: first results on European-wide terrestrial carbon fluxes for the period 2006–2015. *Atmos. Chem. Phys.*, *20*(20), 12063–12091. https://doi.org/10.5194/acp-20-12063-2020

Monteil, G., & Scholze, M. (2021). Regional CO2 inversions with LUMIA, the Lund University Modular Inversion Algorithm, v1.0. *Geosci. Model Dev.*, *14*(6), 3383–3406. https://doi.org/10.5194/gmd-14-3383-2021

Munassar, S., Monteil, G., Scholze, M., Karstens, U., Rödenbeck, C., Koch, F. T., Totsche, K. U., & Gerbig, C. (2023). Why do inverse models disagree? A case study with two European CO2 inversions. *Atmospheric Chemistry and Physics*, *23*(4), 2813–2828. https://doi.org/10.5194/ACP-23-2813-2023

Petrescu, A. M. R., McGrath, M. J., Andrew, R. M., Peylin, P., Peters, G. P., Ciais, P., Broquet, G., Tubiello, F. N., Gerbig, C., Pongratz, J., Janssens-Maenhout, G., Grassi, G., Nabuurs, G. J., Regnier, P., Lauerwald, R., Kuhnert, M., Balkovič, J., Schelhaas, M. J., van der Gon, H. A. C. D., … Dolman, A. J. (2021). The consolidated European synthesis of CO2emissions and removals for the European Union and United Kingdom: 1990-2018. *Earth System Science Data*, *13*(5), 2363–2406. https://doi.org/10.5194/ESSD-13-2363-2021

Thompson, R. L., Broquet, G., Gerbig, C., Koch, T., Lang, M., Monteil, G., Munassar, S., Nickless, A., Scholze, M., Ramonet, M., Karstens, U., van Schaik, E., Wu, Z., & Rödenbeck, C. (2020). Changes in net ecosystem exchange over Europe during the 2018 drought based on atmospheric observations. *Philosophical Transactions of the Royal Society B: Biological Sciences*, *375*(1810), 20190512. https://doi.org/10.1098/rstb.2019.0512

**matrix formulation of the transport**

The model estimate $y^m$ for the observations $y$ is given by

$$y^m = \mathcal{H}(x, E, y^{bg})$$

where:

- $E$ represents the emissions of $CO_2$ and $C\Delta14C$. For $CO_2$ those correspond to the $F$ terms, defined in Section 2.1 of the paper, while for some of the $C\Delta14C$ emission categories, the definitions are slightly more complicated, for instance, the $C\Delta14C$ fossil fuel emissions are given by $E_{ff}^{C\Delta} = \Delta_{ff}F_{ff}$ (see Equation 4b of the paper for the definition of all the categories). The emissions are provided at a 0.5°, hourly resolution. Therefore $E$ can be seen as an array of dimensions ($ncat_{mod}$, $nt_{mod}$, $nlat$, $nlon$). Note that there is no "tracer" dimension: some emission categories refer to $CO_2$, while some others refer to $C\Delta14C$.
- $y^{bg}$ represents the background concentrations (i.e. the influence of emissions further away in time and/or space). Note that "background" here should be understood from a modelling perspective (it is a form of boundary condition), and not in a data assimilation sense.
- $x$ is the control vector, which is adjusted by the inversion. It contains a correction to the emissions E, in the form of offsets. $x$ is provided at a lower resolution than $E$, and has size $n_{opt} = ncat_{opt} \times nt_{opt} \times np_{opt}$, with $n_{opt}$ the number of optimised emission categories (2 or 3 depending on the simulations), $nt_{opt}$ the number of weekly optimisation time steps, and $np_{opt}$ the number of optimised "clusters" (or patches, regions, super-cells, ...), as shown in Figure 2b of the paper.
- $\mathcal{H}$ is the observation operator, which links the control vector $x$, containing the adjusted by the inversion (i.e. in the optimisation space), to the corresponding values in the observation space. It handles essentially three operations:
    1. project the coarse resolution emission offsets $x$ onto the model grid
    2. compute the regional transport of these emissions, i.e. their influence on observed values
    3. combine it with the background concentrations $y^{bg}$

In our case, the atmospheric transport is linear, meaning that the previous equation can be re-written as

$$y^m = y^{bg} + \mathbf{K}(\mathbf{e} + \mathbf{Px})$$

with:

- $\mathbf{P}$ the ($n_{mod}$, $n_{opt}$) matrix, projecting $\mathbf{x}$ onto the emission space.
- $\mathbf{e}$ the vectorized form of $E$ (i.e. $\mathbf{e}$ is a $nmod = ncat_{mod} \times nt_{mod} \times nlat \times nlon$ vector)
- $\mathbf{K}$ is the transport matrix, projecting the emissions onto the observations space. $\mathbf{K}$ has shape ($nobs$, $nmod$)

Note that with this formulation, when $\mathbf{x}(:)$ is 0, (i.e. when $\mathbf{x} = \mathbf{x_b}$), the equation simplified to $y_{apri}^m = y^{bg} + \mathbf{Ke}$: The original, fine-resolution structure (0.5°, hourly) of the emissions is always

present and transported, even when the control vector is at a much lower resolution.

Classically, inversions seek to minimise a cost function $J(\mathbf{x})$ that balances the fit to the prior $\mathbf{x_b}$ with the fit to observations $y$:

$$J(\mathbf{x}) = \frac{1}{2}(\mathbf{x} - \mathbf{x_b})^T \mathbf{B}^{-1}(\mathbf{x} - \mathbf{x_b}) + \frac{1}{2}(\mathbf{y_m} - \mathbf{y})^T \mathbf{R}^{-1}(\mathbf{y_m} - \mathbf{y})$$

We then define $\delta y = y - y_{apri}^m$. Since $y_{apri}^m = y^{bg} + \mathbf{K}\mathbf{e}$, we can replace $y^m - y$ in the equation above by simply $\mathbf{K}\mathbf{x} - \delta y$:

$$J(\mathbf{x}) = \frac{1}{2}(\mathbf{x} - \mathbf{x_b})^T \mathbf{B}^{-1}(\mathbf{x} - \mathbf{x_b}) + \frac{1}{2}(\mathbf{KPx} - \delta\mathbf{y})^T \mathbf{R}^{-1}(\mathbf{KPx} - \delta\mathbf{y})$$

This means that our observation operator $\mathbf{H} = \mathbf{KP}$ is fully linear, and has an adjoint $\mathbf{H^T} = \mathbf{P^T K^T}$.

The transport operator $\mathbf{K}$ is computed using the Lagrangian transport model FLEXPART, and consists essentially of one FLEXPART footprint for each observation. The operator $\mathbf{P}$ is simply regridding the offsets $\mathbf{x}$ onto the model grid. This is done through a series of matrix operations:

$$\mathbf{e}^{apos} = \mathbf{e}^{apri} + \delta\mathbf{e}$$

with:

$$\delta\mathbf{e_c} = (\mathbf{T_H} \otimes \mathbf{T_T})\mathbf{x_c}$$

where $\mathbf{x_c}$ is the portion of the control vector $\mathbf{x}$ containing the offsets to the emissions in category $c$. $\mathbf{T_T}$ is a $(nt_{mod}, nt_{opt})$ matrix such that $\mathbf{T_t}(i,j)$ contains the fraction (between 0 and 1) of the optimisation time step $j$ that falls within the model time step $i$ (so, in our case, it typically contains either 0 (when the model interval $i$ is outside the optimisation interval $j$), or 1 / 168 (i.e. 1 / 7 / 24), when the model interval $i$ is within the optimization interval $j$ (since the model is hourly and the optimisation is weekly). Likewise $\mathbf{T_H}$ is a $(np_{mod}, np_{opt})$ matrix (with $np_{mod} = nlat \times nlon$) such that $\mathbf{T_H}(i,j)$ contains the fraction of the area of the optimisation cluster $j$ that falls within the model grid cell $i$. The values range between 0 and 1 since the optimisation "grid" is irregular (Figure 2b).

The formula above is not practical ($\mathbf{T_H} \otimes \mathbf{T_T}$ is very large!), but it can be reformulated as

$$\delta e_c = vec(\mathbf{T_T X_c T_H^T})$$

, where $\mathbf{X_c}$ is $\mathbf{x_c}$ reshaped as a $(nt_{opt}, np_{opt})$ matrix, and $vec$ is the operator reshaping the result of that operation from a $(nt_{mod}, np_{mod})$ matrix to a $nt_{mod} \times np_{mod}$ vector.

The operation is conducted in a similar way for each category. Mathematically, this can be formulated

$$\delta\mathbf{e} = (\mathbf{T_c} \otimes \mathbf{T_H} \otimes \mathbf{T_T})\mathbf{x_c}$$

with $\mathbf{T_C}$ a $(ncat_{mod}, ncat_{opt})$ matrix such that $\mathbf{T_c}(i,j)$ is 1 if the category $i$ is optimised, and 0 if it is not.

In practice, the calculations are performed one observation at a time, and one category at a time:

$$y_{apos}^i = y^{bg} + \sum_c \mathbf{K}^i \mathbf{e_c} + \sum_{c_{opt}} \mathbf{K^i T_t X_{c_{opt}} T_h}$$

, where $c$ is the list of categories relevant for the tracer ($CO_2$ or $C\Delta14C$) corresponding to the observation $y^i$, and $c_{opt}$ is the list of optimised categories relevant for that tracer (i.e. *ff* and *bio* for $CO_2$, plus *biodis* for $C\Delta14C$, in some simulations). Each row of $\mathbf{K}$ corresponds to one FLEXPART footprint, out of which only the non-zero components are stored (but, mathematically, these footprints are still defined over the entire length of the simulation). Furthermore, $\mathbf{K}$ is in fact identical for all categories (so its real shape is only ($nobs$, $nt_{mod} \times nlat \times nlon$)).

---

## Author Response (AR4)

Lund, October 5th, 2024

**Carlos Gómez-Ortiz**
**Department of Physical Geography and Ecosystem Science**
**Lund University**
**Sweden**

**Jens-Uwe Grooß**
**Editor assigned to Research article EGUSPHERE-2023-2215.**
**Atmospheric Chemistry and Physics (ACP)**

Dear Editor,

Here we address the latest review of our manuscript titled "Can Δ14CO2 observations help atmospheric inversions constrain the fossil CO2 emission budget of Europe?"

We sincerely appreciate feedback on our manuscript and are committed to improve it further.

Below, we provide a detailed response (in regular font) to each of the referee's comments (in italics), indicating how we have addressed them in the revised manuscript. We hope this clarifies any misunderstandings and demonstrates our commitment to meeting the high standards of the journal.

**Referee's comments**

*This manuscript describes the implementation of a dual-tracer approach (CO2 mixing ratios and radiocarbon isotope ratios, Δ14CO2) in an atmospheric inversion framework, LUMIA, for the co-optimization of fossil emissions and land-biosphere fluxes of CO2.*

*While overall the results appear sound, there are a few points in the description of the methodology that should be clarified before final publication.*

*Moreover, I am not convinced that this study fully answers the question in the title: "Can Δ14CO2 observations help atmospheric inversions constrain the fossil CO2 emission budget of Europe?" and wonder if a different title would better reflect the scope of the study. The OSSEs carried-out in this study, although fine for testing the implementation of the dual tracer approach, are a bit limited in terms of answering this question. Specifically, the prior error in the OSSE inversions is known, as it is determined from the difference between the true and prior flux datasets. This is not representative of the reality, when the prior error is unknown. Also, the transport in the OSSEs is perfect, which is also not reflecting the reality. Furthermore, to fully answer the question in the title, would entail investigating which of the sampling strategies, that is, i) hourly integrated samples every 3-days, ii) 2-weekly integrated samples, or iii) both, would provide the best constraint on fossil fuel CO2 emissions. This is not to say that these aspects must be covered for the manuscript to be accepted for publication, only that the title should perhaps better reflect the scope of the present study.*

We agree with the reviewer that the title might be misleading and suggests results beyond the scope of our methodology and described study here. We propose the new title: "A $CO_2$ - $\Delta^{14}CO_2$ inversion setup for estimating European fossil $CO_2$ emissions".

*Specific comments*

*L14-15: This sentence is unclear, needs more context, do the authors refer to the posterior biosphere fluxes which are retrieved with bias, or something else?*

Indeed, we refer to the posterior fluxes. These lines were modified as follows to give more context:

"In all experiments, regions with low sampling coverage, such as Southern Europe and the British Isles, show poorly resolved posterior fossil $CO_2$ emissions. Although the posterior biosphere fluxes in these regions follow the seasonal patterns of the true fluxes, a significant bias remains, making it impossible to close the total $CO_2$ budget."

*L46-47: While it is correct that inverse modelling systems that only constrain land-biosphere fluxes assume that the fossil CO2 emissions are well-known, it does not follow that "this is to avoid any bias the fossil CO2 flux might introduce to the estimates of terrestrial fluxes". Rather the opposite, a fossil CO2 flux estimate that is biased but assumed not to be will introduce errors in the terrestrial fluxes. Furthermore, even in systems constraining only biosphere fluxes, the uncertainty of fossil CO2 emission can be (and should be) accounted for in the observation space.*

Indeed, this sentence was misleading and we agree with the reviewer that prescribing wrong fossil CO2 emissions would lead to a bias in the inferred land-biosphere fluxes. Hence, we removed the sentence.

*L102: Please change "CO2 concentration" to "CO2 mixing ratios" (or "mole fractions") as it is the volume mixing ratio (or equivalently mole fraction) that is reported, not the concentration. Please change this elsewhere in the manuscript as well.*

We changed the term concentration by mixing ratios in the whole manuscript.

*Eq. 1a and 1b: Please use standard notation. In these equations presumably y_co2 and yb_co2 are scalars and Fc is a vector representing 2D space?*

Indeed, $y$ and $y^b$ in Eq. 1a and 1b are scalars. We modified the equation and the subsequent mentions in the text.

*L150: For completeness please also describe what is y_c_delta_14C. Is this the mixing ratio of 14C-CO2?*

Yes, it refers to the mixing ratio of $CO_2 \times \Delta^{14}CO_2$. To add clarity, we reformulated this paragraph as follows:

"where y is the assumed $CO_2$ and $C\Delta^{14}C$ mixing ratio, yb is the modeled $CO_2$ and $C\Delta^{14}C$ background mixing ratio (i.e., the boundary condition) (see Section 3.3). Since the values of $\Delta^{14}CO_2$ in ‰ (permil) units are not additive (as it represents the change of the $^{14}C{:}^{12}C$ atmospheric ratio relative to an absolute standard of $^{14}C$ from 1950 (Stuiver and Polach, 1977)), we convert all $\Delta^{14}CO_2$ values to values of $CO_2 \times \Delta^{14}CO_2$ (or $C\Delta^{14}C$ for simplification) (Basu et al., 2016). In terms of units, for mixing ratios this would be $C\Delta^{14}C$ ppm ‰, and for fluxes PgC ‰ $yr^{-1}$. Since ‰ only means multiplication by 1000, we drop that factor from $\Delta^{14}C$ into the quantity $C\Delta^{14}C$, expressing it in ppm for mole fractions and PgC $yr^{-1}$ for fluxes to maintain the same order of magnitude and units for $CO_2$ and $C\Delta^{14}C$. For example, a sample with a $CO_2$ mole fraction of 400 ppm and a $\Delta^{14}C$ value of 45 ‰ would have $C\Delta^{14}C$ = 18 ppm. Expressed in this way, $C\Delta^{14}C$ becomes additive and can be transported by a model. {…}"

*L150: Here the authors state that y^b is the "modelled background", whereas in L142, they state that y^b is "calculated by computing a smoothed and detrended average of real observations". Please clarify which of these is it?*

We changed the word "modelled" by "assumed" in L150 to make it consistent with the sentence in L142. Since we are doing perfect transport OSSEs, we are using the same background for calculating the synthetic observations and performing the inversions, focusing only in the regional component.

*L153: I think in Eq. 1b it should rather be the fraction of 14C in F_c and not the isotopic signature, which represents the ratio of 14C in the sample relative to the reference, and y_c_delta_14C would be the mixing ratio of 14C-CO2.*

We agree with the referee. We modified these lines as follows:

"In Eq. 1b, the term $\Delta_c$ represents the fraction of $^{14}C$ in the accompanying flux category $F_c$ (Tans et al., 1979; Turnbull et al., 2016)."

*L155: Similar to the above comment, to calculate the mixing ratio of 14C-CO2 one would need to multiply CO2 mixing ratio by the fraction of 14C-CO2, not delta_14C. Or unless the authors use the assumption that 14C << 12C and thus the ratio 14C/12C is approximately equal to 14C/(12C + 14C) in which case this should be explicitly stated.*

We modified this paragraph to add clarity as answered above. We use $\Delta^{14}CO_2$ as defined by Stuiver & Polach (1977), since this is how ICOS samples are reported. An approximation of this definition is:

$$\Delta^{14}C(\text{‰}) = \left( \frac{^{14}C/^{12}C_{sample}}{^{14}C/^{12}C_{standard}} - 1 \right) \times 1000$$

*L155: "ppm" is a unit of mixing ratio not concentration.*

We changed the term concentration by mixing ratios in the whole manuscript.

*L157: Again, if the fraction of 14C/12C is used rather that delta_14C, which I think it should be, then the units of delta_c\*F_c will be PgC/yr. The unit of PgC_permil/yr does not correspond with y_c_delta_14C, which is ppm.*

Indeed, we modeled $C\Delta^{14}C$ in units of ppm for mixing ratios and e.g., $PgC\ yr^{-1}$ for fluxes. We have modified this paragraph as answered above.

*L170: Since fossil CO2 does not contain any 14C it does not contribute to a change in the mixing ratio of 14C-CO2, i.e., has no effect on y_c_delta_14C (in Eq.1b).*

We respectfully disagree with the referee. Although fossil $CO_2$ does not contain any $^{14}C$, it does contribute to a change in the $\Delta^{14}C$ of atmospheric $CO_2$ by diluting the amount of $^{14}C$ in the atmosphere. This dilution leads to a reduction of the $C\Delta^{14}C$ mixing ratio and, consequently, the $\Delta^{14}CO_2$ isotopic ratio. This process is the basis of the Suess effect (Suess 1955; Tans, De Jong, and Mook 1979), and it is the fundamental reason for using $\Delta^{14}CO_2$ as a tracer to separate the fossil and the natural components in atmospheric $CO_2$ observations (Turnbull et al. 2009; Turnbull, Graven, and Krakauer 2016).

*L204: Why was the 2-week integrated sampling strategy for delta_14C chosen, rather than the 1-hour integrated sample every 3-days? Surely, the 1-hour samples would better help resolve the fossil fuel signal, since the transport and source regions could change significantly over the course of 2 weeks.*

The ICOS Atmosphere network has been collecting 2-week integrated samples of $\Delta^{14}CO_2$ since 2016 at 12 stations across Europe thus it is important to us to evaluate the potential use of the available data. Nevertheless, we are aware of the limitations of the 2-week samples and we recently submitted a new manuscript to ACP exploring different sampling strategies.

*L240: Instead of "grid points" do the authors rather mean "grid cells"?*

We changed this line to "grid cells".

*Eq.7: What is the matrix operation indicated by $\otimes$ ? I read it to mean the Kronecker product, in which case T_H $\otimes$ T_T would have dimensions (n^p_mod\*n^t_mod, n^p_opt\*n^p_mod), and then x_c would need to be a vector of n^p_opt\*n^p_mod. Please confirm if this is correct? It would help if the dimensions of H and x_c were also given.*

The reviewer is correct that this is a Kronecker product, but it results is a $\left(n_{mod}^{p} * n_{mod}^{t}, n_{opt}^{p} * n_{opt}^{t}\right)$ matrix. We have added the dimensions of $\boldsymbol{x}_c$ and $\mathbf{H}$ to the sentence following the equation, to lift any source of doubt on the reader side:

"where $\mathbf{H}$ is the observation operator with dimensions $\left(n_{obs}, n_{p_{opt}} * n_{t_{opt}}\right)$, and $\boldsymbol{x}_c$ with dimensions $\left(n_{p_{opt}}, n_{t_{opt}}\right)$ is the portion of the control vector $\boldsymbol{x}$ that contains offsets for the optimized categories c. The matrices $T_T\left(n_{t_{mod}}, n_{t_{opt}}\right)$ and $T_H\left(n_{p_{mod}}, n_{p_{opt}}\right)$ contain the relative contribution of each model time step $t_{mod}$ (1 hour) and of each grid cell $p_{mod}$ (0.5° × 0.5°) to each optimized time step $t_{opt}$ and cluster $p_{mod}$, with $n_{t_{opt}}$ and $n_{p_{opt}}$ the number of optimized intervals (weekly) and grid cell clusters, respectively."

*L259: The definitions of L_h and L_t should be included here.*

We added the following sentence to the end of this line:

"$L_h$ and $L_t$ represent the horizontal and temporal correlation lengths, respectively."

*L381-382: The time window over which the standard deviation is calculated (7-days), which is used as a proxy for the observation error, is very long. This would imply that the authors do not have much confidence in the model's ability to represent synoptic variability in the mixing ratios. There is no discussion of why this long time window was chosen or the evaluation of this choice, e.g., how well does the model capture the variability of tracers for which the fluxes are likely better known (Radon or SF6)?*

This is a valid point, and to test this we repeated the ZBASE experiment with two additional time windows: half a week, and one day. We found the standard deviation to not be too sensitive to the window width (in average at each site) and also, there is no significant impact on the posterior results (Fig. 5 and 6 of this document). In inversions against real observations, we would fine tune the observation uncertainties based in part of the quality of the prior fit to the data, but we cannot do this here since we do not assimilate real data. Furthermore, it is common in inversions to "inflate" the uncertainties, to compensate for the fact that the observation uncertainties are treated as independent (i.e. the "R" matrix is diagonal), which isn't accurate. Our observation error values are on the same order of magnitude as what is typically used in LUMIA CO2 inversions. The table below reports for instance the values used in Munassar et al. (2023). Note that these are weekly aggregated uncertainties. For comparison we have calculated (average) weekly aggregated uncertainties in our case.

| Site | Averaged observation errors per site (ppm) | | | | |
| --- | --- | --- | --- | --- | --- |
| | 1 week | Half week | 1 day | Weekly uncertainty Munassar et al. | Weekly uncertainty as Munassar et al. |
| BIR | 6.76 | 6.37 | 5.62 | 2.5 | 1.1 |
| CMN | 5.9 | 5.58 | 4.96 | 1.5 | 1.0 |
| GAT | 13.27 | 12.38 | 10.87 | 1.5 | 2.2 |
| HPB | 12.19 | 11.35 | 10.06 | 1.5 | 2.1 |
| HTM | 11.8 | 11.06 | 9.74 | 1.5 | 2.0 |
| IPR | 19.94 | 18.97 | 16.72 | 1.5 | 3.4 |
| JFJ | 4.63 | 4.36 | 3.9 | 1.5 | 0.8 |
| KRE | 12 | 11.19 | 9.73 | 1.5 | 2.0 |
| LIN | 18.09 | 16.8 | 14.95 | 2.5 | 3.1 |
| LMP | 5.09 | 4.83 | 4.35 | 1.5 | 0.9 |
| LUT | 17.28 | 16.16 | 14.18 | 2.5 | 2.9 |

| NOR | 10.03 | 9.34 | 8.19 | 1.5 | 1.7 |
|------|-------|-------|-------|-----|-----|
| OPE | 15.82 | 14.65 | 12.94 | 1.5 | 2.7 |
| PAL | 6.78 | 6.26 | 5.57 | 2.5 | 1.1 |
| PUI | 5.77 | 5.47 | 4.82 | 1.5 | 1.0 |
| PUY | 7.31 | 6.83 | 6.12 | 1.5 | 1.2 |
| RGL | 7.87 | 7.32 | 6.59 | 1.5 | 1.3 |
| SAC | 27.9 | 25.88 | 23.27 | 2.5 | 4.7 |
| SMR | 9.14 | 8.51 | 7.52 | 1.5 | 1.5 |
| SSL | 7.16 | 6.78 | 6.06 | 1.5 | 1.2 |
| SVB | 7.65 | 7.12 | 6.41 | 1.5 | 1.3 |
| TRN | 14.66 | 13.52 | 12.03 | 1.5 | 2.5 |
| UTO | 10.52 | 9.91 | 8.85 | 1.5 | 1.8 |
| WAO | 14.73 | 13.62 | 12.26 | 1.5 | 2.5 |

[Figure]

Figure 5. Fossil $CO_2$ emissions.

[Figure]

Figure 6. Biosphere (NEE) fluxes.

*L392-393: I think the authors should specify that the prior fluxes for F_ff and F_bio can have similar distributions to the "true" fluxes, otherwise it's not clear if by "similar" the authors mean similar to each other or what similar to what?*

We reformulate this sentence as follows:

"The reason for using prior fluxes set to zero is that the flux products for both categories can have spatial and temporal distributions similar to their respective true values, making it easier for the model to retrieve the true fluxes."

*L402-403: I do not follow how the potentially large error in F_biodis can be accounted for by using the true value in the inversion and not optimizing it?*

We modified the description of the experiment as follows to add clarity:

"In the final inversion, BASENoBD, we prescribe $F_{biodis}$ (i.e., the true value in this context) instead of optimizing it. The terrestrial disequilibrium term ($F_{biodis}$) is challenging to estimate due to the large uncertainties associated with heterotrophic respiration fluxes and the age of respired carbon (Basu et al., 2016). These uncertainties can vary significantly depending on the vegetation model or methodology used. We compare the posterior $F_{ff}$ of this experiment with the one of the BASE experiment (in which $F_{biodis}$ is optimized), to evaluate the impact of the prior $F_{biodis}$ product on the posterior $F_{ff}$. By keeping $F_{biodis}$ fixed in BASENoBD, we can assess how much of the error in the posterior $F_{ff}$ of BASE comes from the additional optimization of $F_{biodis}$."

*L425: It is not clear what is being compared here, the ZBASE and ZCO2ONLY inversions are closer in agreement to the truth compared to what? The prior?*

The sentence is intended to highlight the comparison between the agreement of the truth and the posterior for $F_{bio}$ relative to $F_{ff}$ in the ZBASE and ZCO2Only experiments. To clarify this, we revised the sentence as follows:

"In general, there is a closer agreement between the posterior and the truth for the biosphere fluxes ($F_{bio}$) than for the fossil $CO_2$ emissions ($F_{ff}$) in both the ZBASE and ZCO2Only experiments. This means that the model performs better at recovering $F_{bio}$ from the observations compared to $F_{ff}$, as shown in Figure 6 for $F_{bio}$ and Figure 5 for $F_{ff}$."

*L428: "ZBASE exhibits a closer alignment to the posterior" – do the authors rather mean that the posterior of ZBASE agrees better with the truth?*

Yes, we mean that the posterior $F_{ff}$ of ZBASE agrees better with the truth than the one of ZCO2Only. We modified this sentence as follows:

"Specifically, the posterior $F_{ff}$ ZBASE exhibits closer alignment to the truth than ZCO2Only with a lower RMSE (see Table 4), indicating a better fit of the seasonality for $F_{ff}$."

*Figure 9: I think the authors should discuss why in July (especially in Eastern Europe) there is this strong departure from the prior and from the true emissions? What is driving this?*

We do discuss this in L630-L637. We added at the end of these lines a new sentence (highlighted):

"As shown in Figure 11, the maximum difference between the prior and the true Fbiodis is of the same order of magnitude for Western/Central Europe (2.1 TgC day$^{-1}$) and Eastern Europe (1.3 TgC day−1) in July. For $F_{ff}$, however, the difference between the prior and truth is about one order of magnitude larger for Western/Central Europe compared to Eastern Europe (0.03 vs 0.005 TgC day$^{-1}$). This larger difference causes a stronger dilution of the fossil emissions in Eastern Europe, and therefore essentially lowers the signal-to-noise ratio of the $\Delta^{14}CO_2$ measurements, and added to the lower network coverage compared to Western/Central Europe, a poorer constrain of the fossil $CO_2$ emissions. As seen also in Figure 9, this is particularly evident in Eastern Europe during the summer months, where the fossil $CO_2$ signal is further convoluted by the large biospheric uptake, making it more difficult to accurately constrain fossil emissions in this region."

*Discussion:*

*How do the diurnal cycles of biosphere CO2 fluxes differ between LPJ-GUESS and VPRM? The diurnal cycle is not optimized in LUMIA (weekly means only are optimized) thus I was wondering how sensitive is the inversion to differences in the diurnal cycle – or are the uncertainties in the observation space so large that this does not have much of an impact?*

Thank you for raising this important point. We agree that the diurnal cycles of biospheric $CO_2$ fluxes are an interesting aspect to explore, particularly when comparing LPJ-GUESS and VPRM. However, in the context of our LUMIA implementation, we focus on weekly means, and the diurnal cycle is not directly optimized in the inversion process. The foreground part of the

observations is sensitive to fluxes aggregated over a few days, which naturally attenuates the impact of diurnal cycles in the observations.

Moreover, since we use afternoon-only data, the inversion is not designed to resolve the full daily cycle of $CO_2$ fluxes. While differences in the diurnal cycles between LPJ-GUESS and VPRM might exist, we don't expect these differences to significantly impact the results of the inversion. The uncertainties in the observation space, combined with the aggregation over days, likely minimize the sensitivity to variations in the diurnal cycle.

We also consider that this aspect is slightly beyond the scope of this paper. Testing this in a pure $CO_2$ inversion might provide valuable insights. However, in practice, we rely on the daily cycles provided by vegetation models, as the "true" daily cycle remains uncertain. Thus, optimizing the diurnal cycle is not a primary focus of the inversion setup at this stage.

We appreciate your insightful question and think it could be a great direction for future research in a more dedicated inversion setup.

*Technical comments*

*L603: should be: "constrained in their inversions" (its -> their)*

We fixed this in the text.

**References**

Munassar, Saqr, Guillaume Monteil, Marko Scholze, Ute Karstens, Christian Rödenbeck, Frank-Thomas Koch, Kai U. Totsche, and Christoph Gerbig. 2023. 'Why Do Inverse Models Disagree? A Case Study with Two European CO $_2$ Inversions'. *Atmospheric Chemistry and Physics* 23(4):2813–28. doi: 10.5194/acp-23-2813-2023.

Stuiver, Minze, and Henry A. Polach. 1977. 'Discussion Reporting of $^{14}$ C Data'. *Radiocarbon* 19(3):355–63. doi: 10.1017/S0033822200003672.

Suess, Hans E. 1955. 'Radiocarbon Concentration in Modern Wood'. *Science* 122(3166):415–17. doi: 10.1126/science.122.3166.415.b.

Tans, P. P., A. F. M. De Jong, and W. G. Mook. 1979. 'Natural Atmospheric 14C Variation and the Suess Effect'. *Nature* 280(5725):826–28. doi: 10.1038/280826a0.

Turnbull, J. C., H. Graven, and N. Y. Krakauer. 2016. 'Radiocarbon in the Atmosphere'. Pp. 83–137 in *Radiocarbon and Climate Change: Mechanisms, Applications and Laboratory Techniques*, edited by E. A. G. Schuur, E. Druffel, and S. E. Trumbore. Cham: Springer International Publishing.

Turnbull, Jocelyn, Peter Rayner, John Miller, Tobias Naegler, Philippe Ciais, and Anne Cozic. 2009. 'On the Use of 14CO2 as a Tracer for Fossil Fuel CO2: Quantifying Uncertainties Using an Atmospheric Transport Model'. *Journal of Geophysical Research: Atmospheres* 114(D22). doi: https://doi.org/10.1029/2009JD012308.

---

## Author Response (AR5)

Lund, November 7th, 2024

**Carlos Gómez-Ortiz**
**Department of Physical Geography and Ecosystem Science**
**Lund University**
**Sweden**

**Jens-Uwe Grooß**
**Editor assigned to Research article EGUSPHERE-2023-2215.**
**Atmospheric Chemistry and Physics (ACP)**

Dear Editor,

Here we address the latest review of our manuscript titled "A $CO_2$ - $\Delta^{14}CO_2$ inversion setup for estimating European fossil $CO_2$ emissions"

We sincerely appreciate feedback on our manuscript and are committed to improve it further.

Below, we provide a detailed response (in regular font) to each of the referee's comments (in italics), indicating how we have addressed them in the revised manuscript. We hope this clarifies any misunderstandings and demonstrates our commitment to meeting the high standards of the journal.

**Referee's comments**

*L151: In line 142, the authors state "the background mixing ratios are calculated by computing a smoothed and detrended average of real observations" but in line 151, they still state that the background is modelled. The authors state that they changed this sentence according to the same comment I made in the first review, but they changed "modelled" to "assumed" for the mixing ratio "y" but not the background "y^b". If I understand correctly the sentence should be e.g.: "y is the modelled mixing ratio, and y^b is the background mixing ratio derived from smoothing real observations…"*

We agree with the reviewer, we made a mistake when modifying the sentence in the last rebuttal. We have modified the sentence in L151 as follows:

"where $y_{CO_2}$ and $y_{C\Delta^{14}C}$ represent the modeled mixing ratios of $CO_2$ and $C\Delta^{14}C$, respectively, and $y_{CO_2}^b$ and $y_{C\Delta^{14}C}^b$ denote their background mixing ratios (i.e., the boundary condition), derived from smoothed real observations (see Section 3.3)."

*L151 and Eq.1b: It would be clearer to simply state that y is the modelled mixing ratio, and y^b is the modelled background mixing ratio. And then I suggest changing the name of the variable y_C_delta^14C to simply e.g. y_14CO2 since this is not a delta value any more since it represents the 14CO2 mixing ratio. Then below, the authors should state that the y_14CO2 mixing ratio is calculated from the delta14C value and that mixing ratio is modelled (and not delta14C) because this is additive.*

*L170: There appears to still be some confusion about the meaning of the mixing ratio of 14CO2. Mixing ratios are calculated with respect to the volume (volume mixing ratio) or mass (mass mixing ratio) of air. Since fossil fuel emissions contain no 14CO2 then these emissions will not affect the mixing ratio of 14CO2 (as far as these emissions do not affect the total mass of air). They will only affect the mixing ratio of CO2. So although fossil emissions affect the ratio of 14CO2 to 12CO2, and thus delta 14CO2, they do not affect the mixing ratio of 14CO2 as this is relative to air (not CO2).*

Here we address the points raised related to L151, Eq. 1b and L170. The referee is correct that emissions of fossil $CO_2$ do not affect $^{14}CO_2$ mole fractions, which is precisely why it is not useful to convert measurements of $\Delta^{14}CO_2$ into $^{14}CO_2$ mole fractions for an inversion. Emissions of fossil $CO_2$ show up as strong depletions in $\Delta^{14}CO_2$ space, but $\Delta^{14}CO_2$ cannot be transported because it is not additive. For instance, $\Delta^{14}CO_2$ cannot be summed across grid cells to construct a "global total $\Delta^{14}CO_2$". Although equation (1b) of Miller et al. (2012) is convenient for expressing the different forcings on atmospheric $\Delta^{14}CO_2$, it is not useful for transport modeling.

Instead, equations (1a) and (1b) of Miller et al. (2012) can be combined to derive equation (1b) of Basu et al., (2016), which serves as the basis for Equations 1-4 of our manuscript. In this formulation, the quantity $C\times\Delta_{atm}$ or $CO_2\times\Delta^{14}CO_2$ *is* additive and therefore *can* be transported. Note that $CO_2\times\Delta^{14}CO_2$ is *not* the mole fraction of $^{14}CO_2$, as per the definition of $\Delta^{14}CO_2$ (Stuiver, 1980; Stuiver and Polach, 1977). It is simply a made-up tracer whose emissions can be calculated given emissions of $CO_2$, $^{14}CO_2$ and $\Delta^{14}CO_2$ source signatures, and whose atmospheric observations can be derived from measurements of $CO_2$ and $\Delta^{14}CO_2$. The formulation of the tracer $CO_2\times\Delta^{14}CO_2$ lends itself to mass balance equations that can be coded up in an atmospheric inverse model.

*Eq. 7: The authors state that the matrices and vectors have the following dimensions:*

*x_c : (n_popt, n_topt) but since x_c is a vector presumably the authors mean (n_popt\*n_topt)*
*H: (n_obs, n_popt\*n_topt)*
*T_H⊗T_T (n_pmod\*n_tmod, n_popt\*n_topt)*

*but then the dimensions of T_H⊗T_T do not conform with those of H, which has n_popt\*n_topt number of columns. And since H is the Jacobian matrix corresponding to the optimized state vector, x_c, why is there any need for T_H⊗T_T because the mapping from the original to the optimized resolution appears to be already taken into account with H.*

We agree with the referee. Indeed, the Kronecker product $\mathbf{T}_H\otimes\mathbf{T}_T$ is the part of $\mathbf{H}$ used to map the fluxes from the modeling space to the optimization space. We replaced $\mathbf{H}$ by $\mathbf{K}$ (the transport operator in Equations 1, 2, and 4, with dimensions $(n_{obs}, n_{p_{mod}} * n_{p_{opt}})$), such that $\mathbf{H} = \mathbf{K}(\mathbf{T}_H \otimes \mathbf{T}_T)$.

We have removed the Kronecker product from the equation and revised the explanatory paragraph as follows:

"Equation 7 can be rewritten as:

$$\delta_y = \sum_c \mathbf{H} x_c$$

where $\mathbf{H}$ is the Jacobian matrix of the observation operator with dimensions $(n_{obs}, n_{p_{opt}} * n_{t_{opt}})$, and $x_c$, with dimensions $(n_{p_{opt}} * n_{t_{opt}})$, represents the portion of the control vector $x$ that contains offsets for the optimized categories $c$. Thus, $x_c$ is built from the relative contribution of each model time step $t_{mod}$ (1 hour) and of each grid cell $p_{mod}$ (0.5°×0.5°) to each optimized time step $t_{opt}$ and cluster $p_{mod}$. Here, $n_{t_{opt}}$ and $n_{p_{opt}}$ represent the number of optimized intervals (weekly) and grid cell clusters (e.g. 2500 for biosphere), respectively."

**References**

Basu, S., Miller, J.B., Lehman, S., 2016. Atmos. Chem. Phys. 16, 5665–5683.

Miller, J.B., Lehman, S.J., Montzka, S.A., Sweeney, C., Miller, B.R., Karion, A., Wolak, C., Dlugokencky, E.J., Southon, J., Turnbull, J.C., Tans, P.P., 2012. Journal of Geophysical Research: Atmospheres 117.

Stuiver, M., 1980. Radiocarbon 22, 964–966.

Stuiver, M., Polach, H.A., 1977. Radiocarbon 19, 355–363.

---

## Author Response (AR6)

Lund, November 13th, 2024

**Carlos Gómez-Ortiz**
**Department of Physical Geography and Ecosystem Science**
**Lund University**
**Sweden**

**Jens-Uwe Grooß**
**Editor assigned to Research article EGUSPHERE-2023-2215.**
**Atmospheric Chemistry and Physics (ACP)**

Dear Editor,

We are submitting our revised manuscript titled titled "A $CO_2$ - $\Delta^{14}CO_2$ inversion setup for estimating European fossil $CO_2$ emissions." Below, we outline the latest changes made prior to submission for publication:

1. We have updated the affiliation of the second author, Guillaume Monteil, to include:

"Barcelona Supercomputing Center, Barcelona, Spain"

2. We have added a missing reference on line 322:

(Hersbach et al., 2018)

Hersbach, H., Bell, B., Berrisford, P., Biavati, G., Horányi, A., Muñoz Sabater, J., Nicolas, J., Peubey, C., Radu, R., Rozum, I., Schepers, D., Simmons, A., Soci, C., Dee, D., and Thépaut, J.-N.: ERA5 hourly data on single levels from 1959 to present, Copernicus Climate Change Service (C3S) Climate Data Store (CDS), https://doi.org/10.24381/cds.adbb2d47, 2018.

Thank you for your attention to our submission.

Best regards,
Carlos Gómez-Ortiz